# UniCA: Unified Covariate Adaptation for Time Series Foundation Model

**Lu Han**[*1,2], **Yu Liu**[*3], **Lan Li**[1,2], **Qiwen Deng**[3], **Jian Jiang**[3], **Yinbo Sun**[3], **Zhe Yu**[3],
**Binfeng Wang**[3], **Xingyu Lu**[3], **Lintao Ma**[†3], **Han-Jia Ye**[†1,2], **De-Chuan Zhan**[1,2]

[1]School of Artificial Intelligence, Nanjing University, China
[2]National Key Laboratory for Novel Software Technology, Nanjing University, China
[3] Ant Group, Hangzhou, China
{hanlu, yehj, zhandc}@lamda.nju.edu.cn,
{nuoman.ly, lintao.mlt}@antgroup.com,yzae2623@gmail.com

## ABSTRACT

Time Series Foundation Models (TSFMs) have achieved remarkable success through large-scale pretraining. However, their design primarily targets real-valued series, limiting their ability to handle general forecasting tasks involving diverse and often *heterogeneous covariates*—such as categorical variables and multimodal data (e.g., images, text)—which are typically task-specific and difficult to leverage during pretraining. To address this gap, we propose Unified Covariate Adaptation (UniCA), a framework to bridge TSFMs with general covariate-aware forecasting. UniCA first performs covariate homogenization to transform heterogeneous covariates into high-level homogeneous series representations and then fuses them via a unified attention-based fusion mechanism. UniCA is compatible and universal for adaptation with both homogeneous and heterogeneous covariates, incorporating extra covariate information while preserving the generalization ability of TSFMs. Extensive experiments on multiple unimodal and multimodal covariate-aware forecasting benchmarks demonstrate the superiority of UniCA, highlighting the promise of covariate-aware TSFM adaptation in real-world forecasting scenarios. Code: https://github.com/hanlu-nju/UniCA.

## 1 INTRODUCTION

Time series forecasting is essential in a wide range of domains, including environmental monitoring Gruca et al. (2022), traffic management Kadiyala & Kumar (2014), energy systems Kardakos et al. (2013), communication networks Peng et al. (2013), and healthcare Morid et al. (2023). Accurate forecasting supports critical decisions in planning, policymaking, and operations. Traditional statistical models such as ARIMA and Exponential Smoothing Box et al. (2015) have been widely used for their simplicity and effectiveness in specific settings. With the advancement of deep learning, models based on Recurrent Neural Networks (RNNs) Hochreiter & Schmidhuber

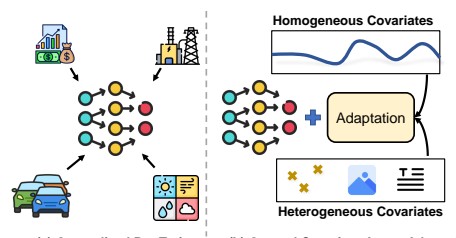

(a) Generalized Pre-Train    (b) General Covariate-Aware Adaptation

Figure 1: TSFMs are pretrained on time series from diverse domains. However, many tasks contain homo/heterogeneous covariates that are hard to use in pre-training. Adaptation methods to handle these covariates are important in these tasks.

(1997); Cho et al. (2014); Rangapuram et al. (2018) and Convolutional Neural Networks (CNNs) Bai et al. (2018); Franceschi et al. (2019) have enabled more expressive modeling of complex temporal

---

[*]Equal Contribution
[†]Corresponding Author

dynamics. Transformer-based architectures Zhou et al. (2021); Nie et al. (2023); Liu et al. (2024b) further advanced the field by capturing long-range dependencies and achieving strong performance, particularly in long-horizon multivariate forecasting. Inspired by the success of foundation models in NLP and vision Devlin et al. (2019); Brown et al. (2020); Radford et al. (2021); Kirillov et al. (2023), recent Time Series Foundation Models (TSFMs) Das et al. (2024); Woo et al. (2024); Ansari et al. (2024); Goswami et al. (2024) have shown strong generalization capability by pretraining on large-scale time series. They learn transferable temporal representations and deliver impressive performance even in zero-shot scenarios.

However, most state-of-the-art TSFMs (Ansari et al., 2024; Das et al., 2024; Goswami et al., 2024) are fundamentally designed for univariate forecasting, processing each time series in isolation. This architectural choice renders them unable to leverage the critical relationships between a target series and its exogenous covariates, limiting their applicability in many real-world scenarios. Some TSFMs adopt covariate-aware (Woo et al., 2024) strategies during pretraining; yet, models trained in this manner often fail to achieve stable and superior performance across diverse tasks (Aksu et al., 2024). More fundamentally, the standard TSFM pretraining paradigm imposes a key limitation: it can only effectively leverage homogeneous covariates (e.g., real-valued time series similar to the target variable). This restricts their ability to handle heterogeneous covariates, which are increasingly common in practical scenarios. Heterogeneous covariates typically includes structured categorical variables (e.g., item IDs, calendar features) and multimodal inputs (e.g., images, texts) (Ma et al., 2024; Liu et al., 2024a). The diversity and task-specific nature make their integration into existing TSFM pipelines non-trivial.

While prior work has addressed unimodal covariate-aware forecasting or multi-modal forecasting through specialized model architectures Salinas et al. (2020); Lim et al. (2019); Das et al. (2023); Jin et al. (2024); Ma et al. (2024); Liu et al. (2024a), these methods are often biased to the task-specific data, lack the generalization ability, and underperform compared to large-scale pretrained TSFM models (Aksu et al., 2024). Therefore, a key challenge remains: *How can we adapt powerful pretrained TSFMs to general covariate-aware forecasting, including homogeneous and heterogeneous covariates, without losing the generalization ability obtained from pretraining?*

In this paper, we address this challenge by proposing a unified and effective adaptation method named **Unified Covariate Adaptation (UniCA)**. The core idea is to perform *covariate homogenization* that transforms heterogeneous covariates into a unified, homogeneous temporal representation, representing the high-level feature changing over time. This transformation allows us to solve the general covariate forecasting with a unified framework in the time series modality. In addition, we design an attention-based fusion mechanism with pre-fusion and post-fusion components that incorporate covariate information before and after the TSFM backbone, respectively, with the parameters of the TSFMs unchanged. The adaptation modules fully utilize the encoding and temporal extraction power of the TSFMs, incorporating the covariates' information while retaining the forecasting ability obtained during their pretraining process. Extensive experiments across a wide range of benchmarks, including traditional single-modal covariate datasets and challenging multimodal datasets, demonstrate the effectiveness and flexibility of UniCA. Our results show that by properly adapting covariate information into the series space, TSFMs can significantly outperform specialized models, thus opening new possibilities for general-purpose time series forecasting in covariate-rich environments. Our main contributions are summarized as follows:

- We formalize the problem of adapting Time Series Foundation Models (TSFMs) to general covariate-aware forecasting scenarios, where the heterogeneous covariates, like categorical or multi-modal covariates, can not be directly utilized by TSFMs.

- We propose Unified Covariate Adaptation (UniCA), a novel framework featuring: (a) *covariate homogenization* to transform diverse covariates into a unified temporal representation, and (b) a dual attention-based fusion mechanism to integrate covariate representation with a frozen TSFM backbone.

- We conduct comprehensive experiments across single-modal and multimodal covariate datasets, demonstrating that UniCA enables TSFMs to achieve superior performance compared to task-specific baselines. The effectiveness of covariate homogenization on TSFMs and specialized methods also proves that it is a simple way to integrate heterogeneous covariates.

## 2 RELATED WORK

**Time Series Foundation Models (TSFMs).** Recent efforts (Ansari et al., 2024; Woo et al., 2024; Goswami et al., 2024; Das et al., 2024; Ekambaram et al., 2024) have developed large-scale pretrained models for time series, enabling zero-shot forecasting or fine-tuning across tasks. Moirai (Woo et al., 2024), MOMENT (Goswami et al., 2024), and TimesFM (Das et al., 2024) adopt a patch-based transformer architecture, while TTM (Ekambaram et al., 2024) builds on TSMixer (Ekambaram et al., 2023), which uses MLPs to mix temporal and feature dimensions. In contrast, Chronos (Ansari et al., 2024) tokenizes time series into discrete vocabularies and trains language models directly on these sequences. TSFMs are trained based on either channel-independence (Han et al., 2024; Goswami et al., 2024; Das et al., 2024; Ansari et al., 2024) that ignores covariates, or the equivariant attention mechanism (Woo et al., 2024) that requires covariates to be homogeneous. The adaptation to heterogeneous covariates scenarios is an unsolved challenge.

**Forecasting with Covariates.** Covariates–help in the forecasting of the target series, but that we are not interested in forecasting–play a crucial role in capturing external signals in forecasting tasks. Classical models like ARIMA (Box & Jenkins, 1968) incorporate them via extra coefficients, while deep models such as DeepAR (Salinas et al., 2020) and TFT (Lim et al., 2019) integrate them as inputs or through specialized encoders. NBEATSx (Olivares et al., 2021) concatenates covariates with the main series for fixed-size input. TTM-CM (Ekambaram et al., 2024) introduces a fine-tuning approach based on channel mixing. (Chen & Zhao, 2024) introduces MiTSformer to handle mixed time series by recovering latent continuous representations from discrete variables to mitigate heterogeneity. Among TSFMs, only Moirai (Woo et al., 2024) natively supports covariates by flattening series and covariates into a joint sequence, using variate IDs for differentiation. None of the existing methods can handle both homogeneous and heterogeneous, especially multi-modal covariates, while our method is meant to solve this challenge.

**Multimodal Time Series Forecasting.** Most multimodal forecasting studies focus on textual enhancement. A line of work seeks to utilize the powerful temporal encoding ability of LLM to improve the forecaster (Zhou et al., 2023; Gruver et al., 2023). While these methods provide the possibility for multimodal forecasting, they usually handle static textual data. Another line combines numerical time series with dynamic textual data, *e.g.* news (Dheenadayalan et al., 2022; Wang et al., 2024a) or weather reports (Obst et al., 2019). Time-MMD (Liu et al., 2024a) introduces a multimodal dataset and a model that processes time and text modalities independently and merges them via linear fusion. Towards image covariates, FusionSF (Ma et al., 2024) proposed the MMSP dataset and a method meant especially for the satellite scenario. Our approach proposed a new way that converts information from other modalities into series and handles them uniformly in time series modality.

**Adapting TSFMs to handle Covariates.** Standard foundation model adaptation, often employing parameter-efficient methods like adapters (Houlsby et al., 2019) and LoRA (Hu et al., 2022), typically assumes consistency between pretraining and downstream task input/output structures. Adapting TSFMs to handle covariates is more complex. While methods like TimesFM (Das et al., 2024), which uses an auxiliary regressor for residual correction, and ChronosX (Pineda-Arango et al., 2025), which injects covariates through linear transformations but limits its application to only point-wise TSFM. All the current adaptation methods struggle with heterogeneous covariates, highlighting the need for more flexible adaptation strategies, which our proposed method provides.

## 3 PROBLEM FORMULATION

**General Covariates-Aware Time Series Forecasting.** In covariates-aware time series forecasting, the objective is to predict $Y_{T+1:T+H} \in \mathbb{R}^{H \times 1}$ by utilizing both past observations of the target $Y_{1:T} \in \mathbb{R}^{T \times 1}$ and the external covariates, as well as considering their temporal relationships. The model takes into account both static covariates $S$ and dynamic covariates $C_{1:T+H}$ [1] to make predictions about the future. Formally, we can express the prediction problem as:

$$\hat{Y}_{T+1:T+H} = f(Y_{1:T}, C_{1:T+H}, S),$$

---

[1]For notational simplicity, we denote both future-known and future-unknown covariates as $C_{1:T+H}$. For future-unknown covariates, the values in the interval $[T+1:T+H]$ are unobserved at prediction time $T$.

where the static covariates $S$ remain unchanged within a series. The dynamic covariates $C_{1:T+H}$ may provide extra information about the past or future state. Both $S$ and $C_{1:T+H}$ may contain *homogeneous and heterogeneous* covariates.

**Heterogeneous Covariates.** In traditional time series analysis tasks, exogenous covariates typically share the same form as the target series, often represented as real-valued numerical variables. This homogeneity allows exogenous covariates and targets to be processed under a unified modeling framework without the need for modality-specific designs (Woo et al., 2024; Liu et al., 2024b). Such covariates are referred to as *homogeneous covariates*. However, with the advancement of data collection, covariates in modern forecasting scenarios exhibit increasingly diverse forms. In many practical applications, covariates are no longer restricted to simple real-valued signals but may involve a wide range of data types. Among these, a particularly significant challenge for TSFMs comes from *heterogeneous covariates*. These covariates can be broadly categorized into two major types: (1) **Categorical covariates**: Discrete attributes such as item identifiers, store locations, event types, or temporal markers. These variables are not inherently numerical and require embedding techniques or specialized handling to be incorporated into forecasting models. (2) **Multimodal covariates**: High-dimensional, complex data modalities such as images, text descriptions. The emergence of heterogeneous covariates poses fundamental challenges to existing TSFM architectures. Unlike homogeneous covariates, which can be directly integrated, heterogeneous covariates demand modality-specific preprocessing, feature extraction, and fusion strategies.

**Covariate-Aware Adaptation.** Time Series Foundation Models (TSFMs) are designed to model the temporal dependencies within a given series. These models are trained on diverse time series from different domains, which makes it hard to incorporate covariates across different domains. *covariate adaptation* involves modifying the model architecture to integrate the covariate information while fully utilizing the temporal encoding ability. The mathematical formulation of this task can be expressed as follows: Given a TSFM $f_{fm}$, the objective is to construct a new forecaster based on the trained foundation model:

$$\tilde{Y}_{T+1:T+H} = g_{ada} \circ f_{fm}(Y_{1:T}, C_{1:T+H}, S), \tag{1}$$

where $g_{ada}$ is the adaptation module, and $g_{ada} \circ f_{fm}$ is the composition model after adaptation. In contrast to training covariate-aware deep learning models from scratch, covariate adaptation involves three distinct challenges:

- **Compatibility**: The adaptation module should be compatible with pretrained TSFMs without requiring extensive full-model retraining or architecture redesign.
- **Universality**: It should be able to handle both homogeneous and heterogeneous covariates.
- **Generalization Preservation**: It should leverage the temporal encoding capabilities learned during pretraining while preserving the generalization ability of the foundation model.

In response to these challenges, we propose **Unified Covariate Adaptation (UniCA)**.

## 4 METHODOLOGY

**Unified Covariate Adaptation (UniCA)** is a general framework that enables Time Series Foundation Models (TSFMs) to effectively incorporate heterogeneous covariate information without disrupting their pretrained temporal modeling capabilities. At a high level, UniCA follows two key principles (1) **Covariate Homogenization**: We transform categorical and multimodal covariates into dense continuous series representations, thus reducing the heterogeneity gap between covariates and target time series. (2) **Modular Fusion**: We decompose the TSFM architecture into interpretable stages and insert an attention-based fusion module to inject covariate information at appropriate locations without interfering with the model's pretrained dynamics.

### 4.1 COVARIATE HOMOGENIZATION

To address covariate heterogeneity, UniCA introduces a homogenization process that converts all covariates into a unified homogeneous space. Specifically, categorical covariates are processed using embedding layers that map discrete tokens into continuous vectors. Multimodal covariates-such as images or texts-are initially fed through modality-specific encoders (e.g., convolutional neural networks for images, pretrained transformers for text) to obtain dense feature representations $H^{(het)}$.

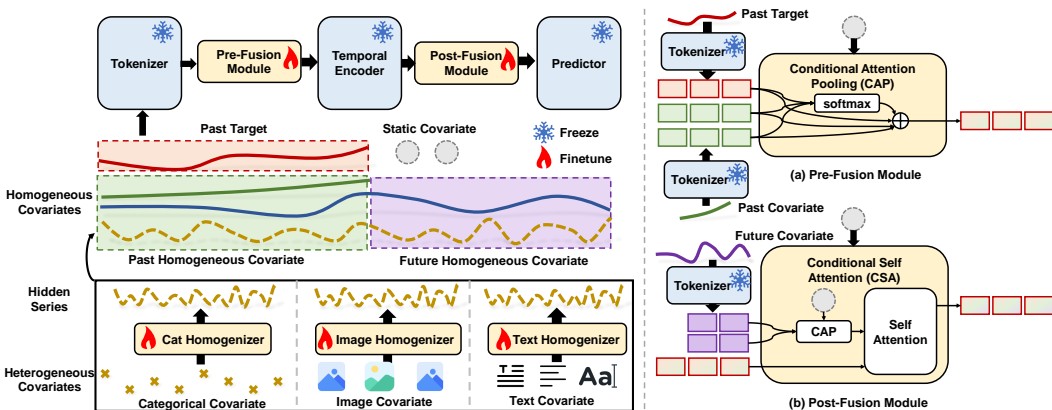

Figure 2: Overview of **Unified Covariate Adapter (UniCA)**. UniCA consists of two key pipelines (1) **Covariate Homogenization**: We use a converter to transform heterogeneous covariates into dense continuous series representations, thus reducing the heterogeneity gap between covariates and target time series. (2) **Modular Fusion**: We decompose the TSFM architecture into interpretable stages and insert Pre-Fusion and Post-Fusion modules to inject covariate information at appropriate locations without interfering with the model's pretrained dynamics.

Similar to connectors used in multimodal learning (Liu et al., 2023), we use a **Covariate Homogenizer (CH)**, a simple linear layer, to transform $H^{(het)}$ into latent homogeneous covariates $C^{(het)}$. These covariates encapsulate the temporal dynamics of high-level features derived from heterogeneous covariates:

$$C^{(het)}_{1:T+H} = \mathrm{CH}(H^{(het)}_{1:T+H}), \tag{2}$$

where $C^{(het)}_{1:T+H} \in \mathbb{R}^{(T+H) \times d^{het}}$, with $d^{het}$ being a tunable hyperparameter. Finally, all homogeneous covariates-whether hidden or observed are aligned along the temporal dimension and concatenated to produce a cohesive set of homogeneous series covariates $C_{1:T+H} \leftarrow [C_{1:T+H}, C^{(het)}_{1:T+H}]$, enabling their integration into a unified covariate fusion framework. In the following part, we assume the unified covariates representation $C_{1:T+H} \in \mathbb{R}^{(T+H) \times M}$, where $M$ is the total number of homogeneous covariates, including the observed homogeneous and the homogenized heterogeneous covariates. This homogenization process ensures the **universality of UniCA**.

## 4.2 COVARIATE FUSION MODULE

**Decomposition of TSFM.** To better incorporate covariate information into the TSFM and fully leverage its capabilities, we first decompose the TSFM architecture according to the functionality:

- **Tokenizer:** $Z = \mathcal{T}(Y_{1:T})$**:** This module transforms raw time series inputs $Y$ into a sequence of tokens $Z \in \mathbb{R}^{P \times d}$, where $d$ is the token dimension, and $P$ is the number of tokens along the temporal dimension and varies between patch-based (Das et al., 2024; Liu et al., 2024c) and point-based (Ansari et al., 2024; Hoo et al., 2025; Shi et al., 2024) methods. The tokenizer is responsible for generating suitable representations for the temporal encoder, acting as the connection between the raw representation and the main part of the model.

- **Temporal Encoder $H = \mathcal{E}(Z)$:** Subsequently, the encoder processes the tokenized sequence $Z$ to extract high-level temporal patterns and dependencies. The most popular encoder is Transformer. This stage leverages the pre-trained temporal encoding capabilities of the TSFM.

- **Predictor $\hat{Y}_{T+1:T+H} = \mathcal{P}(H)$:** Finally, the predictor utilizes the encoded representations $H$ to generate forecasts $\hat{Y}$ for the future horizon $T + 1$ to $T + H$. For decoder-only architectures (Brown et al., 2020; Das et al., 2024), we regard the linear output layer as the predictor.

This modular decomposition is applicable to the vast majority of TSFMs, ensuring the **compatibility of UniCA** and enabling a clean separation of responsibilities and facilitating the integration of covariate information without disrupting the core temporal processing. Based on this, we propose attention-based pre/post fusion modules to incorporate past and future covariates into the TSFMs.

**Pre-Fusion Module.** Prior to the encoding stage, the pre-fusion module integrates past covariate information with the historical target values. This module enriches the tokens with historical external factors, allowing the encoder to capture the joint dynamics between the time series and its past covariates. Inspired and simplified from Lim et al. (2019), we use a *Conditional Attention Pooling(CAP)* mechanism to fuse the past information while maintaining interpretability. Concretely, given past target $Y_{1:T}$, past covariates $C_{1:T}$ and the static feature $S$, $C_{1:T}$ and $S$ are first converted to embeddings by the tokenizer of the TSFM and a newly initialized embedding layer $\rho$:

$$E_{C_{1:T}} = \mathcal{T}(C_{1:T}), \quad E_S = \rho(S), \tag{3}$$

where $E_{C_{1:T}} \in \mathbb{R}^{P \times M \times d}$, $E_S \in \mathbb{R}^{N \times d}$ ($E_S = 0$ if no static covariates provided) is the representation of dynamic and static covariates. Then:

$$\begin{aligned} Z_{C_{1:T}} &= \mathrm{CAP}(E_{C_{1:T}} \mid E_S) := \mathrm{softmax}(A)V, \\ \text{where} \quad A &= \mathrm{GRN}(\mathrm{flat}(E_{C_{1:T}}), E_S) \text{ and } V = \mathrm{GRN}(E_{C_{1:T}}). \end{aligned} \tag{4}$$

GRN is Gated Residual Network (a residual MLP) used in Lim et al. (2019), $\mathrm{flat}(\cdot)$ flattens the last two dimension of $E_{C_{1:T}}$, $A \in \mathbb{R}^{P \times 1 \times M}$ is the attention affinity on each feature, $V \in \mathbb{R}^{P \times M \times d}$. Then, a Gated Linear Unit (GLU) Dauphin et al. (2017) is used to further trade off the influence of covariates $Z_c$:

$$\tilde{Z} = Z + \mathrm{GLU}(Z_{C_{1:T}}). \tag{5}$$

This fused representation is then forwarded to the temporal encoder to produce $\tilde{H} \in \mathbb{R}^{P \times d}$:

$$\tilde{H} = \mathcal{E}(\tilde{Z}). \tag{6}$$

**Post-Fusion Module.** The future-known covariates $C_{T+1:T+H}$ provide direct insight into future conditions, making them particularly valuable for forecasting. Therefore, we choose to use a post-fusion module to incorporate future covariate information into the encoded representations $H$ after the temporal extraction process. This step is crucial when future exogenous factors are expected to influence the forecast. We first tokenize the future-known covariates:

$$E_{C_{T+1:T+H}} = \mathcal{T}(C_{T+1:T+H}), \tag{7}$$

where $E_{C_{T+1:T+H}}$ represents the tokenzied future covariates. We then apply the conditional attention pooling mechanism to selectively aggregate the most relevant aspects of these future covariates at each time step. Formally,

$$Z_{C_{T+1:T+H}} = \mathrm{CAP}(E_{C_{T+1:T+H}} | E_S). \tag{8}$$

Once the most relevant future information is selected and fused, we integrate it with the past sequence by feeding both into a self-attention layer. This step enables the model to learn contextual dependencies between past and future covariates, allowing for an enriched representation that better captures the interplay between historical and forward-looking information. Mathematically, the final fused representation is obtained as:

$$[\hat{H}, \hat{Z}_{C_{T+1:T+H}}] = \mathrm{SelfAttn}([\tilde{H}, Z_{C_{T+1:T+H}}]). \tag{9}$$

Then we predict the future target $\hat{Y}$ with the predictor $\mathcal{P}$:

$$\hat{Y}_{T+1:T+H} = \mathcal{P}(\hat{H}). \tag{10}$$

Similar to adapters (Houlsby et al., 2019) in LLM, our UniCA is a plug-in module that keeps the pretrained model parameters unchanged, thus **preserving generalization** capabilities of TSFMs.

### 4.3 LOSS FUNCTION

A key design principle of UniCA is its seamless compatibility with diverse TSFMs. We train the UniCA adaptation modules using the same loss function the foundation model was originally pre-trained with. This aligns the adaptation process with the TSFM's inherent objective. Specifically, we employ the quantile loss (Wen et al., 2017; Lim et al., 2019) for Chronos and TimesFM, the Huber loss (Huber, 1992) for Time-MoE, and Negative Log Likelihood (NLL) for Moirai. For training stability across series of varying scales, we normalize each target instance by its historical mean and standard deviation, following the instance normalization approach in Kim et al. (2021).

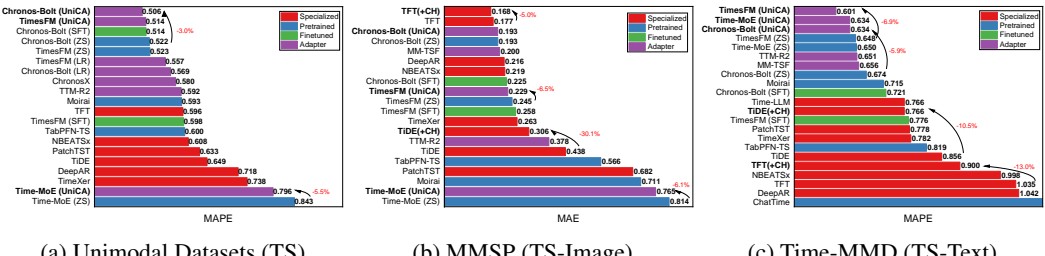

| (a) Unimodal Datasets (TS) | (b) MMSP (TS-Image) | (c) Time-MMD (TS-Text) |

Figure 3: Forecasting performance on general covariate-aware forecasting datasets, including 12 unimodal datasets and multi-modal datasets MMSP and Time-MMD. Results are reported as MAPE averaged over sub-datasets for both unimodal and Time-MMD datasets. For the MMSP dataset, MAE is used instead, as near-zero target values render MAPE unstable.

## 5 EXPERIMENTS

**Metrics.** Following the evaluation in (Aksu et al., 2024; Zhou et al., 2021), we consider four metrics to evaluate the performance of forecasters: Mean Absolute Percentage Error (MAPE), Mean Square Error (MSE), Mean Absolute Error (MAE) for point forecasting ability, and Continuous Ranked Probability Score (CRPS) for probabilistic forecasting, which is implemented as the mean Weighted Quantile Loss (WQL) (Park et al., 2022). In all experiments, the WQL is computed on quantile levels $\{0.1, 0.2, \ldots, 0.9\}$. For methods generating sample forecasts, we compute the quantiles based on 256 samples, whereas quantile forecasting methods are trained on the same quantile levels we use for evaluation. Following the practice in Woo et al. (2024) to reduce the dataset bias, we normalize each result by dividing the result of the **Naive** method (Hyndman & Athanasopoulos, 2018), where all forecasts have the value of the last observation.

**Compared Methods.** To comprehensively evaluate the effectiveness of our proposed UniCA framework, we compare it against a broad set of baseline methods spanning four major categories: (a) **Specialized Models**: These models are trained from scratch for specific forecasting tasks. We include two representative subtypes: (i) *univariate methods*, which include **PatchTST** (Nie et al., 2023) and (ii) *covariate-aware methods*, which includes **DeepAR** (Salinas et al., 2020), **TFT** (Lim et al., 2019), **TiDE** (Das et al., 2023), **N-BEATSx** (Olivares et al., 2021), **TimeXer** (Wang et al., 2024b). (b) **Pretrained TSFM (ZS)**. They are evaluated in a *zero-shot* manner without task-specific fine-tuning. We select three popular TSFMs – **Chronos-Bolt** (Ansari et al., 2024),**TimesFM** (Das et al., 2024), **Time-MoE** (Shi et al., 2024). (c) **Fine-tuned TSFM (SFT)**: Full-parameter fine-tuning on downstream datasets. (d) **Adapter-based Models**: These methods introduce additional modules attached to the TSFM to inject covariate information, allowing adaptation with fewer trainable parameters. We compare the **Linear Regression (LR) adaptation** proposed in (Ansari et al., 2024; Das et al., 2024), which regresses the ground truth against covariates and the residuals are fed to TSFMs, **TTM-R2** (Ekambaram et al., 2024), **ChronosX**[2] (Pineda-Arango et al., 2025). For **multi-modal** experiments, we include **FusionSF** (Ma et al., 2024) and **MM-TSF**(Chronos-Bolt as TS predictor) (Liu et al., 2024a) for TS-image task (MMSP); **Time-LLM** (Jin et al., 2024), **MM-TSF** and **ChatTime** (Wang et al., 2025) for TS-text task (Time-MMD).

**Implementation Details.** We adopt default context length for each TSFM, *e.g.* 2048 for Chronos-Bolt and 4096 for Time-MoE, while prediction lengths are dataset-specific (Appendix A). All time series data is pre-processed such as normalization, as detailed in Appendix C.3. In all experiments, learning rates were selected from $\{10^{-3}, 10^{-4}, 10^{-5}, 10^{-6}\}$ based on validation performance. The homogenizer CH is implemented as a simple linear model. For each heterogeneous covariate, we select the number of projected hidden series $d^{het}$ in $\{1, 2, 4, 8, 16\}$. For image covariates, we use a simple 4-layer CNN (Krizhevsky et al., 2012) because the satellite images with dimension $64 \times 64 \times 4$ are not regular images. For text covariates, we use GIST (Solatorio, 2024) as the encoder. A comprehensive list of hyperparameters is presented in Appendix C.7.

---

[2]This method is especially designed for Chronos-T5 model. Results obtained from our own implementation, as official code for ChronosX was not available at the time of this work.

## 5.1 UNI-MODAL COVARIATE AWARE FORECASTING

**Datasets.** We evaluate our method on 12 publicly available datasets commonly employed in covariate-aware forecasting research (Lim et al., 2019; Das et al., 2023; Oreshkin et al., 2019; Pineda-Arango et al., 2025; Aksu et al., 2024; Olivares et al., 2021). To create the test sets, we employ two distinct strategies based on the number of subseries in each dataset. For datasets with a relatively large number of subseries, *i.e.*, we reserve the final "prediction length" points of each subseries as the test set (Lim et al., 2019). For datasets with fewer subseries, we partition 10% of the data as the test set and apply a sliding window approach for evaluation, with a step size of 1 (Zhou et al., 2021). We also spare validation sets with the same points as the test set. Detailed descriptions of the datasets can be found in Appendix A.

**Main results.** The results in Figure 3a highlight the effectiveness of UniCA in uni-modal covariate-aware forecasting. UniCA consistently outperforms zero-shot TSFMs, achieving optimal performance among adapter methods (0.506 MAPE for Chronos-Bolt). While standard finetuning shows minimal gains or degradation, UniCA delivers substantial improvements, confirming its ability to *preserve generalization*. UniCA also surpasses specialized methods (0.596-0.738 MAPE), demonstrating *universality* across architectures. These results validate UniCA's design goals of *compatibility*, *universality*, and *generalization preservation*.

## 5.2 MULTI-MODAL COVARIATE-AWARE FORECASTING

**Datasets.** We evaluated UniCA on tasks involving multi-modal covariates, specifically images and text. For image-based covariates, we utilized the Multimodal Solar Power (MMSP) dataset from (Ma et al., 2024). For text-based covariates, we used the Time-MMD (Liu et al., 2024a) dataset. Details are in the Appendix A. Traditional covariate models use no multi-modal information.

**Main results.** On the Time-MMD dataset (figure 3c), TimesFM (UniCA) ranks among the top performers, significantly outperforming most specialized and pretrained baselines. In the MMSP benchmark (figure 3b), TFT variants achieve the best results, while our UniCA-enhanced models show consistent improvements over their base versions. *Notably, UniCA provides substantial gains for TimesFM (6.5% reduction in error) and Chronos-Bolt (5.9% reduction), confirming its effectiveness in multi-modal covariates modeling*. This suggests that multi-modal covariates can hinder forecasting performance if not handled appropriately, but our UniCA framework can robustly control the information flow through its attention-based fusion module and covariate homogenization, leading to consistent performance improvements across different foundation models.

**Homogenizer on Specialized Methods.** To evaluate the generalizability of the Covariate Homogenizer (CH), we integrate it into two representative covariate-based forecasting models to support multi-modal forecasting: TFT and TiDE. The models augmented with CH consistently outperform their vanilla counterparts. For instance, TFT+CH achieves a notable 5% drop on MMSP and 13.0% on Time-MMD. Similarly, TiDE+CH demonstrates a substantial improvement, with the MAE reduced by 30.1% on MMSP and 10.5% on Time-MMD. *These results highlight that CH provides a simple yet effective way to integrate multimodal information.*

## 5.3 ANALYSIS

**Efficiency.** The homogenizer of the UniCA uses a linear layer. The pre-/post- fusion module computes the covariate weights and pooling with the weights, introducing complexity only linear to the number of covariates and the model's dimension. All the components are lightweight. Figure 5a shows that UniCA introduces little computation or storage burden for the TSFMs.

**Effectiveness of Covariate Adaptation.** Figure 4 highlights the effectiveness of the adapter-based covariate integration strategy in leveraging the generalization capabilities of pre-trained TSFM models. In figure 4a, we observe that the Adapted variant—using our proposed UniCA adapter—consistently outperforms the Pretrained zero-shot model (Moirai(ZS)) across all metrics. This indicates that *the diverse reliance of covariates on different datasets is difficult to learn with a pretrained model*. Adaptation with UniCA provides better performance. However, the pretrained knowledge should not be fully discarded. In figure 4b, compared to fully finetuned backbones, the Frozen + Adapter setup achieves better performance, particularly in terms of MAPE and CRPS. These findings validate our design intuition: adapter-based covariate incorporation serves as

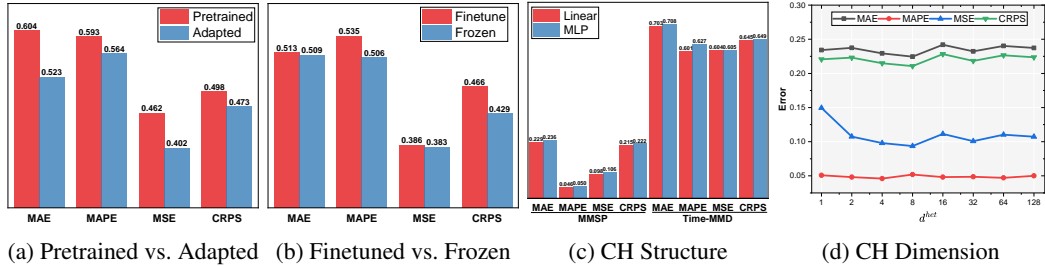

(a) Pretrained vs. Adapted  (b) Finetuned vs. Frozen  (c) CH Structure  (d) CH Dimension

Figure 4: Average relative MAPE on unimodal datasets with model setups (a) **Pretrained**: Moirai(ZS), **Adapted**: Moirai (UniCA). (b) **Finetuned**: Chronos-Bolt (UniCA) with fine-tuned backbone, **Frozen**:freeze backbone; Ablation on (c) structure of covariate homogenizer. (d) hidden dimension of covariate homogenizer.

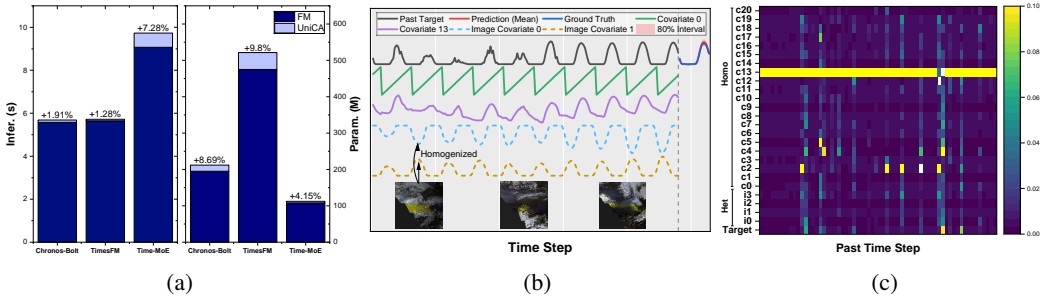

(a)  (b)  (c)

Figure 5: Analysis of UniCA. **(a) Efficiency on Time-MMD:** The adapter adds minimal overhead in inference time (left panel) and trainable parameters (right panel). **(b) Covariate Homogenization on MMSP:** Aligned heterogeneous covariates reveal meaningful patterns like seasonality and trends. **(c) Attention Maps:** The fusion module dynamically attends to different covariates over time for the sample in (b).

a lightweight yet powerful mechanism to bridge the gap between general-purpose time series representations and task-specific covariate contexts, fully utilizing pretrained knowledge while enabling covariate-aware forecasting.

**Homogenizer Architecture.** Our evaluation of the homogenizer architecture, detailed in figure 4c, shows that a simple Linear layer and a Multi-Layer Perceptron (MLP) achieve similar performance. Notably, the Linear model slightly outperforms the MLP, indicating that a more parsimonious design is not only sufficient but preferable. Therefore, we use the Linear homogenizer as the default.

**Homogenizer Dimension.** We conduct an ablation study on the homogenized dimension $d^{\text{het}}$, which controls the projection space for diverse covariates. Varying $d^{\text{het}}$ from 1 to 128, we evaluate performance using MAE, MAPE, MSE, and CRPS. As shown in Figure 4d, performance improves sharply from $d^{\text{het}} = 1$ to 4 (e.g., MSE drops from 0.15 to under 0.10), highlighting the benefit of a more expressive projection. Increasing $d^{\text{het}}$ beyond 8 yields diminishing returns and slight performance degradation, suggesting redundancy or overfitting. Metrics remain stable in the range $[4, 32]$. We set $d^{\text{het}} = 4$ by default.

**Visualization of Covariate Homogenization.** To illustrate UniCA's behavior and the effect of covariate homogenization, we visualize examples from the MMSP dataset (figure 5b). The homogenized representations of satellite images reveal meaningful temporal patterns: *Image Covariate 1* captures periodicity, while *Image Covariate 0* also reflects trends aligned with target scale. This shows that homogenization effectively transforms heterogeneous covariates into task-relevant representations, validating our alignment design in UniCA.

**Attention-based Covariate Selection.** Figure 5c shows attention maps before and after fusion for the same sample in figure 5b. The fusion module dynamically adjusts attention weights across time; notably, Covariate 13 consistently receives the highest weights, matching its rich temporal patterns and strong correlation with the target. In contrast, the target itself is not overly emphasized, suggesting the fusion module learns to complement, rather than duplicate, target signals—demonstrating its ability to identify and integrate informative covariates.

## 6 CONCLUSION

In this work, we address a critical limitation of existing Time Series Foundation Models (TSFMs): their inability to incorporate homogeneous and heterogeneous covariates in general forecasting effectively. To overcome this, we propose **UniCA**, a unified covariate adaptation framework that extends TSFMs to general covariate-aware forecasting scenarios. UniCA achieves this by transforming diverse covariates into high-order homogeneous series and integrates them via an attention-based fusion module, preserving the integrity of pretrained temporal modeling. Extensive experiments on both unimodal and multimodal datasets demonstrate UniCA's compatibility, universality, and effectiveness across diverse forecasting tasks. A discussion of its limitations and future directions is provided in Appendix I.

## 7 ETHICS STATEMENT

This work complies with the ICLR Code of Ethics . Our study focuses on methodological contributions to time series forecasting with heterogeneous covariates. All experiments are conducted on publicly available benchmark datasets (e.g., M5, Retail, MMSP, Time-MMD), which do not contain personally identifiable or sensitive information. No human subjects or private data were involved, and thus no additional ethical approval was required. We have taken care to ensure that our methods and findings do not pose foreseeable risks of harm, discrimination, or misuse. We believe this research aligns with the principles of responsible stewardship, fairness, transparency, and reproducibility.

## 8 REPRODUCIBILITY STATEMENT

We have made extensive efforts to ensure reproducibility of our results. Detailed descriptions of the model architecture, training procedures, hyperparameters, and evaluation protocols are provided in the main paper and Appendix. We also include ablation studies and additional results in the supplementary materials to validate robustness. All datasets used in our experiments are publicly accessible, with preprocessing steps clearly documented. To further support reproducibility, we provide anonymous source code and scripts, enabling verification and extension of our findings.

## ACKNOWLEDGMENTS

This research was supported by Natural Science Foundation of Jiangsu Province of China under Grant (BK20250062), NSFC (62476123, 62522605), Collaborative Innovation Center of Novel Software Technology and Industrialization, Ant Group Research Intern Program.

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

# Appendix

## Table of Contents

# A DATASETS DESCRIPTIONS

## A.1 UNI-MODAL TIME SERIES DATASETS

Our study employed 12 uni-modal datasets with covariates. Detailed descriptions of the targets, covariates, and data sources are provided in Table 1, while Table 2 outlines the dataset statistics. Specifically, a selection of electricity load forecasting datasets, including Covid19 Energy, GFC12, GFC17, PDB, Spain, BDG-2 Hog, BDG-2 Bull, and BDG-2 Cockatoo, was directly retrieved from the Lotsa repository on Hugging Face: `https://huggingface.co/datasets/Salesforce/lotsa_data`. An exception to this is the GFC14 dataset, which we acquired from its original source (Hong et al., 2016) due to the absence of covariate data in the version available on Hugging Face.

Table 1: Dataset Descriptions

| Dataset Name | Descriptions | Covariates | Source |
|---|---|---|---|
| EPF | Day-ahead electricity prices from five major power markets: Nord pool, PJM, FR, BE, and DE | load forecasts, wind generation | (Lago et al., 2021) |
| M5-daily | M5 competion using 30K hierarchical sales data from Walmart in three states CA, TX and WI to forecast the daily sales for the next 28 days. | store ID, item ID, sell prices, week day, month, year, SNAP CA, SNAP TX, SNAP WI, event type | (Makridakis et al., 2022) |
| Retail | Corporación Favorita Grocery Sales Forecasting competition hosted in Kaggle. | store no., item no., on promotion, oil prices, week day, month, year, holidays events, | (Corporación Favorita, 2018) |
| BDG-2 Hog | The Building Data Genome 2 (BDG2) dataset in the Hog region. An open dataset that includes non-residential building-level data collected from 3053 electricity meters, which covers 1636 buildings. | air temperature, drew temperature, sea level pressure, wind direction, wind speed | (Miller et al., 2020; Wang et al., 2023; Woo et al., 2024) |
| BDG-2 Bull | BDG-2 dataset collected from Univ. of Texas at Austin. | air temperature, wind speed sea level pressure | (Miller et al., 2020; Wang et al., 2023; Woo et al., 2024) |
| BDG-2 Cockatoo | BDG-2 dataset collected from Cornell University. | air temperature | (Miller et al., 2020; Wang et al., 2023; Woo et al., 2024) |
| Covid19 Energy | 3+ years of load data from the Day-Ahead Electricity Demand Forecasting Competition. The purpose is to study the impact of the Covid-19 on the power system. | air temperature | (Farrokhabadi et al., 2022; Wang et al., 2023) |
| GFC12 | 20 aggregated-level load series data from the Global Energy Forecasting Competition 2012 | Randomly selected second temperature data because there is no one-to-one correspondance between the temperature and load data | (Hong et al., 2014; Wang et al., 2023; Woo et al., 2024) |
| GFC14 | Seven years of load series data from the Global Energy Forecasting Competition 2014 | Averaged temperature from the raw 25 temperature data series. | (Hong et al., 2016) |
| GFC17 | Eight load data from year 2016 to 2017 originally from the Global Energy Forecasting Competition 2014 | air temperature | (Hong et al., 2019; Wang et al., 2023; Woo et al., 2024) |
| PDB | Two years of PDB electric power load history data from the Kaggle data competition. | air temperature | (Yeafi, 2021; Wang et al., 2023; Woo et al., 2024) |
| Spain | Hourly energy demand generation and weather in five major cities in Spain. It is a Kaggle data competition. | air temperature of Barcelona | (J., 2019; Wang et al., 2023; Woo et al., 2024) |

## A.2 MULTI-MODAL DATASETS

To evaluate our approach, we utilized two distinct multi-modal datasets: Time-MMD (Liu et al., 2024a) and the Multimodal Solar Power (MMSP) dataset (Ma et al., 2024).

Table 2: Dataset Statistics. Dynamic covariates and past dynamic covariates are covariates that are observed and unobserved in the forecasting horizon, respectively.

| Dataset Name | Domain | Num. Series | Freq. | Categorical Cov. | | Real Cov. | | Num. Obs. | Pred. Len. |
|---|---|---|---|---|---|---|---|---|---|
| | | | | Static | Dynamic | Dynamic | Past Dynamic | | |
| EPF | Energy/Price | 5 | H | 0 | 0 | 0 | 2 | 218,280 | 48 |
| M5-daily | Sales | 30,490 | D | 5 | 8 | 1 | 0 | 59,181,090 | 28 |
| Retail | Sales | 119,048 | D | 7 | 4 | 0 | 2 | 140,246,203 | 8 |
| BDG-2 Bull | Energy/Load | 41 | H | 0 | 0 | 0 | 3 | 719,304 | 48 |
| BDG-2 Cockatoo | Energy/Load | 1 | H | 0 | 0 | 0 | 1 | 17,544 | 48 |
| Covid19 Energy | Energy/Load | 1 | H | 0 | 0 | 0 | 1 | 31,912 | 48 |
| GFC12 | Energy/Load | 20 | H | 0 | 0 | 0 | 1 | 788,280 | 48 |
| GFC14 | Energy/Load | 1 | H | 0 | 0 | 0 | 1 | 60,600 | 48 |
| GFC17 | Energy/Load | 8 | H | 0 | 0 | 0 | 1 | 140,352 | 48 |
| BDG-2 Hog | Energy/Load | 24 | H | 0 | 0 | 0 | 5 | 421,056 | 48 |
| PDB | Energy/Load | 1 | H | 0 | 0 | 0 | 1 | 17,520 | 48 |
| Spain | Energy/Load | 1 | H | 0 | 0 | 0 | 1 | 35,064 | 48 |

Table 3: Multi-modal Dataset Descriptions

| Dataset Name | Domain: Target | Num. Series | Freq. | Multi-Modal Covariates | | | Num. Obs. | Pred. Len. |
|---|---|---|---|---|---|---|---|---|
| | | | | Text | Image | Time Series | | |
| Time-MMD | Agriculture: retail broiler composite | 1 | W | USDA Broiler Market News Report; Daily National Broiler Market at a Glance, etc. | N/A | N/A | 17,520 | 12 |
| | Climate: US Precipitation Index | 1 | M | Drought Report National Climate Report | N/A | N/A | 496 | 12 |
| | Economy: US trade in goods with World | 1 | M | U.S. International Trade in Goods and Services Economic Indicators Report | N/A | N/A | 423 | 12 |
| | Energy: gasoline price | 1 | M | Annual Energy Outlook fom EIA; Weekly Petroleum Status Report | N/A | N/A | 1479 | 12 |
| | Social Good: unemployment statistics in the US | 1 | M | monthly employment situations; annual labor force characteristics by race and ethnicity | N/A | N/A | 900 | 12 |
| | Public Health: InfluenzaLike Illness statistics | 1 | W | Weekly U.S. Influenza Surveillance Report Annual Flu Season Key Studies and News Reports | N/A | N/A | 1389 | 12 |
| | Environment: Outdoor air quality | 1 | D | Daily News | N/A | N/A | 11,102 | 12 |
| | Traffic: Traffic Volume Trends | 1 | M | Weekly Traffic Volume Report | N/A | N/A | 531 | 12 |
| | Security: Disaster and Emergency Grants | 1 | M | Billion-Dollar Weather and Climate Disasters; Disaster and emergency declarations | N/A | N/A | 297 | 12 |
| MMSP | Energy: Solar Power Generation | 88 | H | N/A | Satellite Images | Numerical Weather Predictions, (Latitude, Longitude) | 1,129,920 | 24 |

The Time-MMD dataset is a multi-domain resource encompassing nine diverse areas such as Agriculture, Climate, Health, and Traffic. It features paired textual and time series data, where the textual information is derived from curated reports and web search results, as detailed in (Liu et al., 2024a). Among the original 9 subsets, we exclude 2 sets (Agriculture and Economy) because we find no specialized method can outperform the Naive method, thus they may be unpredictable. We reserved 20% of each dataset as the test set. Time-MMD allows for the investigation of models capable of integrating information across different modalities and domains.

The MMSP dataset comprises one and a half years of solar power generation records collected from 88 individual plants. We select the first 10 series as in (Ma et al., 2024). Crucially, it includes temporally aligned heterogeneous covariates for each plant, consisting of satellite imagery and numerical weather predictions. This dataset provides a challenging real-world scenario for multi-modal learning, requiring the fusion of visual and numerical information to predict power output.

## B    COMPARED METHODS

In this section, we provide an overview of the baseline methods employed in our experiments, with a focus on their methodological frameworks and covariate handling strategies. Each approach is analyzed in terms of its integration of homogeneous covariates, highlighting strengths and limitations in modeling external dependencies.

### B.1    SPECIALIZED METHOD

**PatchTST (Nie et al., 2023)**    PatchTST is a model that transforms time series into patches, which are then encoded using a Transformer to produce forecasts. It involves two main components: Patching and Channel Independence. Patching divides time series into subseries-level patches, serving as input tokens to the Transformer. This preserves local temporal patterns while minimizing computational complexity for attention maps, enabling longer history modeling. Channel Independence ensures that each channel uses the same embedding and Transformer weights, treating multivariate inputs as separate but parallel sequences. PatchTST does not incorporate covariate information in forecasting.

**NBEATS (Oreshkin et al., 2019)**    NBEATS (Neural Basis Expansion Analysis for Time Series Forecasting) is a deep learning model designed for univariate time series forecasting. Its core idea is to decompose the time series into interpretable components, typically trend and seasonality, using stack-based architecture where each stack consists of multiple blocks. Each block learns to forecast a portion of the input series using basis expansion functions (approximated by fully connected layers) and subtracts its forecast from the input, passing the residual to the next block or stack. This iterative residual learning allows NBEATS to model complex patterns and achieve strong forecasting performance, often outperforming statistical ans hybrid methods, while also offering some interpretability through its decomposition into basis function.

**NBEATSx (Olivares et al., 2021)**    An extension of the purely univariate NBEATS model, NBEATSx incorporates covariates by appending them to the backcast and forecast layers in each neural block. It uses a dual-stack architecture (generic + interpretable) to model both nonlinear dependencies and explicit covariate effects. This makes it effective for scenarios where future covariates (e.g., planned events) are known.

**DeepAR (Salinas et al., 2020)**    Developed by Amazon, DeepAR is a probabilistic RNN-based model that explicitly integrates covariates. Each time step's dynamic covariates (e.g., temperature, price) are concatenated with the target variable and fed into the RNN. It assumes covariates are known for both training and prediction, making it ideal for applications like demand forecasting where external factors (e.g., marketing campaigns) drive outcomes.

**TFT (Lim et al., 2019)**    Developed by Google, the Temporal Fusion Transformer (TFT) uses a modular design to integrate static covariates (e.g., store IDs), known future inputs (e.g., holidays), and observed variables (e.g., sales). It processes past covariates in the encoder and future covariates in the decoder via gated recurrent networks and variable selection networks. This hierarchical approach ensures robustness to missing or noisy covariates while maintaining interpretability.

**TiDE (Das et al., 2023)**    TiDE (Time-series Dense Encoder) is a deep learning model for multivariate time series forecasting, distinguished by its efficient, MLP-only architecture. It processes the historical lookback window of the target series and any available past covariates through a dense encoder to learn a latent representation. A separate MLP-based decoder then uses this representation, along with linearly projected future covariate information, to generate multi-step forecasts.

**TimeXer (Wang et al., 2024b)**    TimeXer is a Transformer-based time series model that processes time series as sequences of patches. It uses a hierarchical structure with patch embedding, temporal encoding, and attention mechanisms to capture both short-term and long-term dependencies. TimeXer handles the covariates by employing the "variate-level embedding". External covariates are embedded and then integrated directly into the patch representations of the target time series. This allows the model to learn how these external factors influence the internal time series dynamics

at the patch level, enabling the model to account for the impact of these exogenous variables in its predictions.

## B.2 Pretrained Method

**Moirai (Woo et al., 2024)** [3] Moirai, a time series foundation model from Salesforce, is engineered for universal forecasting across diverse time series data. At its core, Moirai utilizes a Transformer-based architecture and is pre-trained on a massive and varied dataset called LOTSA. A key architectural component is its ability to handle any number and type of covariates, both those known in the future and those that are not. Moirai achieves this by conditioning its probabilistic forecast generation, which uses a flexible output distribution, on these covariates, allowing the model to produce forecasts that are informed by the provided exogenous variables.

**TabPFN-TS (Hoo et al., 2025)** [4] TabPFN-TS, a regression variant of the TabPFN(Hollmann et al., 2025) model to time series, is a foundation model pre-trained on pure artificial datasets, enabling few-shot time series forecasting. When incorporating covariate data, TabPFN-TS typically treats these exogenous variables as additional features. These covariate features are then concatenated with the temporal information before being processed by its Transformer-based architecture, which is adapted for tabular data. The model's pre-training on tabular data allows it to potentially learn complex relationships between all input features, including the covariates, even with limited time series-specific training.

**Time-MoE** See section C.4 for the model details.

**Chronos-Bolt** See section C.4 for the model details.

**TimesFM** See section C.4 for the model details.

## B.3 Multimodal Metehod

**Time-LLM** Time-LLM (Jin et al., 2024) is a foundation model that adapts a pre-trained Large Language Model (LLM) for general-purpose time series analysis. Instead of full fine-tuning, it employs a lightweight "reprogramming" layer. This layer transforms input time series into a text-prototype format that the frozen LLM can process. By aligning the LLM's inherent sequence modeling capabilities with statistical time series patterns, Time-LLM can perform diverse tasks like forecasting, classification, and anomaly detection through simple text prompts, leveraging the LLM's reasoning abilities while remaining computationally efficient.

**ChatTime** ChatTime (Wang et al., 2025) is a multimodal time series foundation model designed as a single, end-to-end language model that directly processes interleaved sequences of numerical time series data and natural language. It introduces key innovations, including a unified time series tokenizer that represents time series patches as discrete tokens and a temporal-aware attention mechanism to effectively capture complex temporal dependencies. By training on this mixed-modality input, ChatTime can perform diverse analysis tasks like forecasting and classification through a conversational interface, directly interpreting and responding to queries about the provided data. We used the checkpoint ChengsenWang/ChatTime-1-7B-Chat[5] released by the authors in our experiment.

## B.4 Finetuned Method

Supervised Fine-Tuning (SFT) is the process of adapting a pre-trained Time Series Foundation Model (TSFM) to a specific downstream task or dataset by further training it on target-specific labeled data. This allows the model to leverage its general time series understanding learned during

---

[3] https://huggingface.co/Salesforce/moirai-1.1-R-small
[4] https://huggingface.co/Prior-Labs/TabPFN-v2-reg
[5] https://huggingface.co/ChengsenWang/ChatTime-1-7B-Chat

pre-training and specialize its parameters for improved performance on the new, specific time series. In our implementation, we utilized the target time series without the covariates and adopted the same hyperparameters (e.g., learning rate) and data split as UniCA for consistency. During training, the model adjusts its pre-learned weights to better align with the characteristics of the target series, enhancing specialization for the task at hand.

### B.5 ADAPTER METHOD

**Linear Regression (LR) Adapter.** In our experiments, we leveraged exogenous variables using a linear regression methodology inspired by the approaches of the Chronos(Ansari et al., 2024) and TimeFM(Das et al., 2024) time series foundation models. This regressor approach involves decomposing the target series into two components: contributions from the covariates and the target itself. Initially, we perform a regression of the target variable against the known covariates. Subsequently, we subtract the predicted target values from the actual target values to compute the residuals. These residuals serve as the context for the time series foundation models, which forecast future residuals. The ultimate forecasts are obtained by summing the predicted residuals with the target forecasts derived from the covariates.

A notable limitation of the covariate regressor approach is its reliance on covariates that are known over the forecasting horizon, such as dynamic categorical and dynamic real covariates. Past covariates, including past dynamic categorical and past dynamic real covariates, are only available for the context window. To address this limitation, we employ the corresponding TSFM to forecast these past covariates into the horizon, thus extending their utility beyond the context window.

**TTM (Ekambaram et al., 2024)** [6] Tiny Time Mixers (TTM) are compact models for multivariate time series forecasting, featuring only 1 million parameters. Built on the efficient TSMixer architecture, TTMs use MLPMixer blocks with simple gated attention, offering a faster alternative to traditional Transformer self-attention mechanisms. TTMs are pre-trained on diverse, large-scale datasets from Monash and LibCity, encompassing various domains and temporal scales. TTM's architecture addresses data heterogeneity through innovations such as Adaptive Patching for adjusting patch lengths, Diverse Resolution Sampling for enhancing generalization across resolutions, and Resolution Prefix Tuning for embedding resolution info in training. This approach allows TTMs to excel in resource-limited settings by initially training models channel-independently, followed by fine-tuning to integrate target and exogenous channel correlations.

## C IMPLEMENTATION DETAILS

### C.1 CODE AVAILABILITY

Our code has been made anonymous and is available at `https://anonymous.4open. science/r/UniCA-C5E0`.

### C.2 COMPUTE RESOURCE INFORMATION

For all the experiments, we use 4 GeForce RTX 3090. For baselines, we used cpu instances with 40 virtual cpus and 384 GiB of memory. The library requirement for reproducing the results is available on the above repository.

### C.3 PREPROCESSING

We mainly follow the series preprocessing pipeline proposed in TFT (Lim et al., 2019). We impute missing values in both the target and covariate series using forward filling and add a corresponding binary indicator to mark the imputed timesteps. The time features are generated based on the time series frequency (e.g., hour, weekday, month as periodic features). Then, the time features are vertically stacked with known dynamic features to form a unified feature matrix. Missing static features are filled with a default value of zero. Finally, each time series is assigned a unique identifier

---

[6]https://huggingface.co/ibm-granite/granite-timeseries-ttm-r2

Table 4: Overview of the baseline models, grouped by type and implementation source. We utilized implementations from the popular time series libraries GluonTS and NeuralForecast, or the official author repository (marked as "Reference"). All experiments were conducted using the default hyperparameters provided by the respective implementation.

| Model | Type | Implementation |
|---|---|---|
| PatchTST | Specialized | GluonTS |
| NBEATS | Specialized | GluonTS |
| DeepAR | Specialized | GluonTS |
| TFT | Specialized | GluonTS |
| TiDE | Specialized | GluonTS |
| NBEATSx | Specialized | NeuralForecast |
| TimeXer | Specialized | NeuralForecast |
| Moirai | Pretrained | Reference |
| TimesFM | Pretrained | Reference |
| TabPFN-TS | Pretrained | Reference |
| TTM-R2 | Pretrained/Adapter | Reference |
| FusionSF | Multimodal | Reference |
| Time-LLM | Multimodal | Reference |
| ChatTime | Multimodal | Reference |

to distinguish it in multi-series forecasting. This ensures that the resulting data format meets the input requirements of deep learning models, providing a normalized representation for time series forecasting.

## C.4 ARCHITECTURE OF TSFM

In this section, we detail the tokenization, encoding, and decoding procedures of two time series foundation models: Chronos and TimesFM. These decomposition steps provide a clearer understanding of their internal mechanisms and differences.

**Chronos-Bolt (Ansari et al., 2024).** [7] Chronos-Bolt adopts an encoder-decoder architecture based on T5. During tokenization, the input time series undergoes instance normalization. It is then segmented into patches along with its mask, and the two streams are concatenated before embedding. An optional `[REG]` token can be added to support regression-style outputs. The encoder transforms the tokenized inputs via a stack of T5 encoders, generating contextualized hidden states. These are fed to the decoder, which performs sequence generation conditioned on attention masks, and yields multiple quantile forecasts. Extended prediction lengths are handled through decoding extrapolation. All outputs are rescaled back using stored normalization parameters.

**TimesFM (Das et al., 2024).** [8] TimesFM utilizes a Transformer decoder-only architecture. The tokenization stage includes preprocessing, fixed-length patching, and normalization via mean and standard deviation. Patches are augmented with mask features and projected into embedding space, optionally with positional encodings. The encoder applies multi-layer self-attention to obtain contextual representations. The decoder operates in an auto-regressive manner, iteratively generating future values. Outputs include both mean and quantile predictions, which are de-normalized to restore the original scale. TimesFM also supports frequency-based conditioning and hybrid-frequency modeling for improved multi-scale forecasting.

**Time-MoE (Shi et al., 2024)** [9] Time-MoE is the first billion-scale time-series foundation model that marries a decoder-only Transformer with a sparse mixture-of-experts (MoE) backbone to boost

---

[7]https://huggingface.co/amazon/chronos-bolt-base

[8]https://huggingface.co/google/timesfm-2.0-500m-pytorch

[9]https://github.com/time-moe/time-moe

capacity without proportional inference cost. Each Transformer block replaces the dense feed-forward layer with a shared pool of eight experts, and a learned router sparsely activates just two experts per token, while rotary positional embeddings and RMSNorm enhance stability. The authors pre-train three variants—Time-MoE-base (50 M activated / 113 M total parameters), Time-MoE-large (200 M / 453 M) and Time-MoE-ultra (1.1 B activated / 2.4 B total)—all support channel-independent forecasting for arbitrary horizons via a multi-resolution head. Training uses Huber loss and the newly curated Time-300B corpus: $\approx$ 48 M sequences and $>$ 300 billion time points drawn from nine domains.

Table 5: Comparison between Chronos-Bolt, TimesFM and Time-MoE in terms of model architecture and processing steps.

| Component | Chronos-Bolt | TimesFM | Time-MoE |
|---|---|---|---|
| Architecture | T5-based encoder-decoder | Decoder-only | Decoder-only |
| Tokenizer | Patch-based Residual MLP | Patch-based Residual MLP | Point-based Gated Linear |
| Encoder | T5 encoder stack | Custom Transformer | Custom Transformer |
| Predictor | T5 decoder | Residual MLP | Linear |
| Prediction Output | Multiple quantile predictions | Mean and quantile predictions | Point predictions |

## C.5 Algorithm of UniCA

We summarize the entire forecasting process of UniCA in Algorithm 1. The procedure begins with *Covariate Homogenization*, where heterogeneous covariates such as categories, images, or text are first encoded into dense features and then mapped into a latent homogeneous space via the Covariate Homogenizer (CH). These transformed covariates are then temporally aligned and concatenated with any observed homogeneous covariates to form the unified covariate sequence. Next, in the *Pre-Fusion Module*, we integrate historical covariate information into the tokenized target sequence using a conditional global attention mechanism followed by a gating unit. This enriched sequence is passed to the pretrained encoder of the TSFM to extract temporal patterns. After encoding, the *Post-Fusion Module* incorporates future-known covariates using another attention-based fusion mechanism, allowing the model to dynamically select complementary covariate signals. Finally, the predictor of the TSFM generates the future forecasts from the fused representation. This modular workflow enables UniCA to flexibly and effectively adapt general-purpose TSFMs to covariate-rich forecasting scenarios, while preserving the pretrained temporal modeling capabilities.

## C.6 Optimization

We train our models using the Adam optimizer. The initial learning rate is set to a fixed value and subject to adjustment via a `ReduceLROnPlateau` learning rate scheduler. The scheduler monitors validation loss (if available) or training loss and reduces the learning rate by a factor of 0.5 when the monitored metric has not improved for a specified number of epochs (`patience`). No weight decay is applied to regularize the model.

To prevent overfitting, early stopping is implemented based on validation loss. Training proceeds in mini-batches, with each epoch comprising 50 gradient steps. Model checkpoints are saved corresponding to the epoch that yields the best validation performance.

## C.7 Hyperparamters

For all experiments, we search the hyperparameters listed in table 6.

---

**Algorithm 1** UniCA: Unified Covariate Adaptation for TSFM

---

**Require:** Target series $\boldsymbol{Y}_{1:T}$, static covariates $\boldsymbol{S}$, dynamic covariates $\boldsymbol{C}_{1:T+H} = \{\boldsymbol{C}_{1:T}, \boldsymbol{C}_{T+1:T+H}\}$, pretrained TSFM $(\mathcal{T}, \mathcal{E}, \mathcal{P})$
**Ensure:** Forecast $\hat{\boldsymbol{Y}}_{T+1:T+H}$
 1: **Covariate Homogenization:**
 2: **for** each heterogeneous covariate **do**
 3:   Encode modality to dense feature $H^{(het)}$
 4:   Convert to homogeneous covariate via covariate homogenizer $\boldsymbol{C}^{(het)} = \text{CH}(H^{(het)})$
 5: **end for**
 6: Concatenate all homogeneous covariates: $\boldsymbol{C} \leftarrow \{\boldsymbol{C}, \boldsymbol{C}^{(het)}\}$
 7: **Pre-Fusion Module:**
 8: Tokenize past target: $\boldsymbol{Z} = \mathcal{T}(\boldsymbol{Y}_{1:T})$
 9: Tokenize past covariates: $\boldsymbol{E}_{C_{1:T}} = \mathcal{T}(\boldsymbol{C}_{1:T})$
10: Embed static covariates: $\boldsymbol{E}_S = \rho(\boldsymbol{S})$
11: Compute conditional attention: $\boldsymbol{Z}_{C_{1:T}} = \text{CondAttnPool}(\boldsymbol{E}_{C_{1:T}} \mid \boldsymbol{E}_S)$
12: Fuse covariates with GLU: $\tilde{\boldsymbol{Z}} = \boldsymbol{Z} + \text{GLU}(\boldsymbol{Z}_{C_{1:T}})$
13: **Temporal Encoding:**
14: Encode fused sequence: $\tilde{\boldsymbol{H}} = \mathcal{E}(\tilde{\boldsymbol{Z}})$
15: **Post-Fusion Module:**
16: Tokenize future covariates: $\boldsymbol{E}_{C_{T+1:T+H}} = \mathcal{T}(\boldsymbol{C}_{T+1:T+H})$
17: Compute conditional attention: $\boldsymbol{Z}_{C_{T+1:T+H}} = \text{CondAttnPool}(\boldsymbol{E}_{C_{T+1:T+H}} \mid \boldsymbol{E}_S)$
18: Fuse via self-attention: $[\hat{\boldsymbol{H}}, \hat{\boldsymbol{Z}}_{C_{T+1:T+H}}] = \text{SelfAttn}([\tilde{\boldsymbol{H}}, \boldsymbol{Z}_{C_{T+1:T+H}}])$
19: **Forecasting:**
20: Predict target: $\hat{\boldsymbol{Y}}_{T+1:T+H} = \mathcal{P}(\hat{\boldsymbol{H}})$

---

Table 6: Key hyperparameters, their search spaces, or fixed values used in training UniCA across all datasets.

| Hyperparameter | Value / Range | Description |
|---|---|---|
| Learning rate | {1e-3,1e-4,1e-5,1e-6} | Initial learning rate for Adam optimizer |
| Weight decay | {1e-2,1e-4,1e-6} | L2 regularization weight |
| Scheduler patience | 5 | Epochs to wait before reducing LR |
| Scheduler factor | 0.5 | Multiplicative factor for LR reduction |
| Batch size | {8,16,32,64} | Number of samples per training batch |
| Max epochs | 100 | Maximum number of training epochs |
| Early stopping patience | 10 | Epochs to wait for improvement before stopping |
| Context length | TSFM-specific | Length of input window for encoder |
| Prediction length | Dataset-specific | Length of prediction window |
| Embedding dimension | TSFM-specific | Dimension of input embeddings |
| Homogenizatoin dimension | {1,2,4,8,16} | Dimension of homogenized series of each heterogeneous covariate |

## D  EXPERIMENT DETAILS

### D.1  TRAIN-TEST SPLITTING.

The train-test follows the setups in (Aksu et al., 2024). For datasets with large number of series, *i.e.* M5 (Makridakis et al., 2022) and Retail (Lim et al., 2019), we spare the last "prediction length"

of each series for test and all the observed points before the test points are used for training. For other datasets, we partition 10% of the data as the test set and apply a sliding window approach for evaluation, with a step size of 1 (Zhou et al., 2021). Among the training points, we also split the validation points.

## D.2 EVALUATION METRICS

We use four metrics to evaluate performance of forecasters: Mean Absolute Error (MAE), Mean Square Error (MSE), Mean Absolute Percentage Error (MAPE) for point forecasting ability, and Continuous Ranked Probability Score (CRPS) for probabilistic forecasting. For all metrics, we use GluonTS library implementation to calculate final values (Alexandrov et al., 2020).

**MAE** The *Mean Absolute Error* (MAE) is a commonly used evaluation metric in time series forecasting that measures the average magnitude of errors between predicted and actual values, without considering their direction. It is defined as:

$$\text{MAE} = \frac{1}{n} \sum_{t=1}^{n} |Y_t - \hat{Y}_t|,$$

where:

- $Y_t$ is the ground truth value at time step $t$,
- $\hat{Y}_t$ is the predicted value at time step $t$,
- $n$ is the total number of observations.

MAE is scale-dependent and expresses the error in the same units as the data, making it directly interpretable. It is robust to outliers compared to squared-error metrics, but does not penalize large errors as heavily as MSE.

**MSE** The *Mean Squared Error* (MSE) quantifies the average of the squared differences between predicted and actual values. It is defined as:

$$\text{MSE} = \frac{1}{n} \sum_{t=1}^{n} (Y_t - \hat{Y}_t)^2,$$

where:

- $Y_t$ is the actual value at time $t$,
- $\hat{Y}_t$ is the forecasted value at time $t$,
- $n$ is the number of observations.

MSE penalizes larger errors more severely due to the squaring operation, which makes it particularly sensitive to outliers. Like MAE, MSE is also scale-dependent, and it is widely used in regression and forecasting tasks due to its mathematical properties that facilitate optimization.

**MAPE** MAPE is an evaluation metric used to measure the accuracy of forecasts in time series analysis. It is defined as the mean of the absolute percentage differences between the actual values $Y_t$ and the predicted values $\hat{Y}_t$. The formula for MAPE is:

$$\text{MAPE} = \frac{1}{n} \sum_{t=1}^{n} \left| \frac{Y_t - \hat{Y}_t}{Y_t} \right|,$$

where:

- $Y_t$ is the actual value at time $t$,

- $\hat{Y}_t$ is the forecasted value at time $t$,
- $n$ is the number of observations.

This metric expresses the forecast error as a percentage of the actual values, making it scale-independent and easy to interpret. However, it is sensitive to values of $Y_t$ that are zero or close to zero, as this can lead to division by zero or inflated error percentages.

**CRPS**    The *Continuous Ranked Probability Score* (CRPS) is a metric used in probabilistic forecasting to evaluate the accuracy of predicted cumulative distribution functions (CDFs) against observed values. Given a predicted distribution with CDF $F$ and a ground truth value $y$, the CRPS is defined as:

$$\mathrm{CRPS}(F, y) = \int_0^1 2\Lambda_\alpha(F^{-1}(\alpha), y)\, d\alpha,$$

where the quantile loss $\Lambda_\alpha(q, y)$ is defined as:

$$\Lambda_\alpha(q, y) = (\alpha - \mathbf{1}\{y < q\})(y - q).$$

In practice, computing the CRPS integral can be computationally intensive. To address this, we approximate the CRPS using a discrete sum over a finite set of quantile levels. This approximation, often referred to as the mean weighted quantile loss (Park et al., 2022), is given by:

$$\mathrm{CRPS} \approx \frac{1}{K} \sum_{k=1}^{K} \mathrm{wQL}[\alpha_k],$$

where $K$ is the number of quantile levels, and $\{\alpha_1, \alpha_2, \ldots, \alpha_K\}$ are the selected quantile levels (e.g., $\alpha_k = 0.1k$ for $k = 1, 2, \ldots, 9$ when $K = 9$).

The weighted quantile loss $\mathrm{wQL}[\alpha]$ for each quantile level $\alpha$ is calculated as:

$$\mathrm{wQL}[\alpha] = 2\frac{\sum_t \Lambda_\alpha(\hat{q}_t(\alpha), y_t)}{\sum_t |y_t|},$$

where:

- $\hat{q}_t(\alpha)$ is the predicted $\alpha$-quantile at time step $t$,
- $y_t$ is the actual observed value at time $t$,
- $\Lambda_\alpha(\hat{q}_t(\alpha), y_t)$ is the quantile loss at time $t$ for quantile level $\alpha$.

# E    FULL RESULTS

## E.1    ROBUSTNESS EVALUATION VIA ERROR BARS.

To evaluate the robustness of UniCA under different random seeds, we conduct experiments with seed values $\{41, 42, 43, 44, 45\}$ on all uni-modal datasets and report the average performance along with 1-sigma standard deviation (mean $\pm$ std), as shown in Table 7. Specifically, "C. (UniCA)" and "T. (UniCA)" denote the proposed UniCA framework built on top of Chronos Bolt and TimesFM, respectively.

The results demonstrate that UniCA consistently achieves strong performance with low variance across all metrics, indicating its robustness against random initialization. For example, on the GFC17 dataset, the MSE is $0.096 \pm 0.003$ for Chronos-based UniCA and $0.094 \pm 0.004$ for TimesFM-based UniCA, showcasing both accuracy and stability. This pattern holds across other datasets, supporting the statistical reliability and generalization ability of UniCA regardless of the underlying backbone.

Table 7: Error bar results of UniCA on uni-modal datasets. "C." and "T." indicate UniCA instantiated with Chronos-Bolt and TimesFM, respectively. All results are averaged over five runs with random seeds {41, 42, 43, 44, 45} and are reported as mean ± standard deviation (1-sigma).

| | | Average | Bull | Cockatoo | COVID19 | EPF | GFC12 | GFC14 | GFC17 | Hog | M5 | pdb | Retail | Spain |
|---|---|---|---|---|---|---|---|---|---|---|---|---|---|---|
| C. (UniCA) | Average | 0.457±0.001 | 0.690±0.004 | 0.822±0.000 | 0.144±0.000 | 0.434±0.002 | 0.459±0.010 | 0.319±0.002 | 0.241±0.004 | 0.716±0.005 | 0.613±0.001 | 0.190±0.002 | 0.656±0.007 | 0.196±0.002 |
| | MAE | 0.509±0.001 | 0.809±0.002 | 0.874±0.000 | 0.178±0.000 | 0.440±0.001 | 0.505±0.012 | 0.361±0.002 | 0.290±0.005 | 0.781±0.004 | 0.699±0.002 | 0.228±0.002 | 0.712±0.005 | 0.230±0.002 |
| | MAPE | 0.506±0.002 | 0.639±0.014 | 0.820±0.002 | 0.179±0.000 | 0.645±0.004 | 0.559±0.009 | 0.395±0.002 | 0.286±0.004 | 0.683±0.005 | 0.764±0.002 | 0.220±0.002 | 0.655±0.017 | 0.225±0.002 |
| | MSE | 0.383±0.002 | 0.700±0.005 | 0.746±0.000 | 0.041±0.000 | 0.356±0.002 | 0.285±0.010 | 0.166±0.002 | 0.096±0.003 | 0.701±0.006 | 0.566±0.002 | 0.078±0.001 | 0.769±0.003 | 0.089±0.001 |
| | CRPS | 0.429±0.001 | 0.610±0.002 | 0.848±0.000 | 0.178±0.000 | 0.293±0.001 | 0.487±0.010 | 0.354±0.002 | 0.292±0.005 | 0.700±0.006 | 0.424±0.001 | 0.232±0.002 | 0.489±0.004 | 0.239±0.002 |
| T. (UniCA) | Average | 0.472±0.001 | 0.739±0.001 | 0.838±0.001 | 0.143±0.000 | 0.440±0.002 | 0.468±0.002 | 0.321±0.002 | 0.237±0.005 | 0.766±0.003 | 0.613±0.001 | 0.180±0.002 | 0.655±0.005 | 0.267±0.008 |
| | MAE | 0.526±0.001 | 0.876±0.005 | 0.886±0.001 | 0.178±0.000 | 0.452±0.001 | 0.512±0.002 | 0.359±0.002 | 0.285±0.006 | 0.836±0.003 | 0.701±0.000 | 0.221±0.002 | 0.709±0.003 | 0.305±0.009 |
| | MAPE | 0.514±0.002 | 0.668±0.011 | 0.812±0.001 | 0.178±0.000 | 0.643±0.005 | 0.564±0.002 | 0.402±0.004 | 0.282±0.005 | 0.726±0.015 | 0.737±0.002 | 0.210±0.003 | 0.648±0.027 | 0.296±0.009 |
| | MSE | 0.403±0.001 | 0.757±0.006 | 0.784±0.001 | 0.040±0.000 | 0.364±0.001 | 0.299±0.002 | 0.168±0.001 | 0.094±0.004 | 0.758±0.006 | 0.587±0.001 | 0.068±0.001 | 0.776±0.004 | 0.147±0.007 |
| | CRPS | 0.445±0.001 | 0.655±0.004 | 0.870±0.001 | 0.177±0.000 | 0.302±0.001 | 0.499±0.002 | 0.354±0.001 | 0.287±0.005 | 0.746±0.005 | 0.426±0.000 | 0.222±0.002 | 0.485±0.001 | 0.318±0.009 |

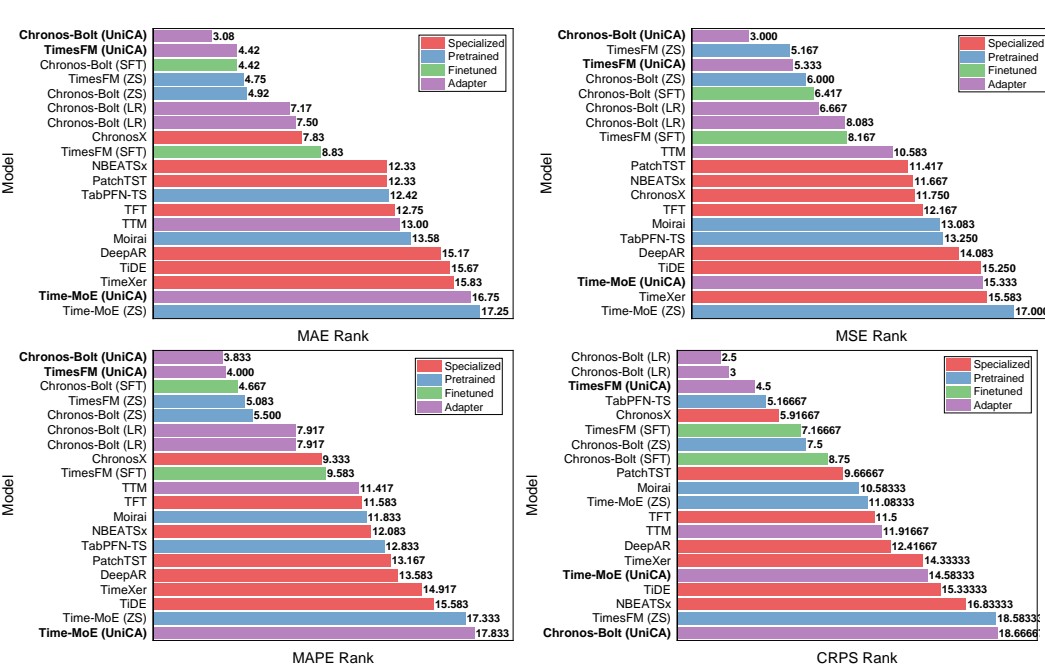

Figure 6: Metrics rank on uni-modal covariate-aware datasets.

## E.2 UNIMODAL FORECASTING

**Main results.** To provide a comprehensive comparison across all baseline and proposed methods, we report the detailed forecasting results on the 12 unimodal covariate-aware datasets in Table 8. This includes performance across four common evaluation metrics: MAE, MAPE, MSE, and CRPS. The models compared include traditional baselines (e.g., NBEATS, DeepAR, TFT), pretrained foundation models (e.g., TimesFM, Chronos), and our proposed adaptation strategies (UniCA, SFT, LR, ZS). We average each metric across all datasets and report both the raw values and metric-wise ranks.

To further illustrate the relative performance of each method, Figure 6 shows the average rank of each model across all datasets under each metric. Lower rank indicates better performance. As shown, our method **Chronos-Bolt (UniCA)** consistently outperforms all others in all four evaluation metrics. Notably, both Chronos and TimesFM, when adapted using UniCA, achieve significant improvements over their zero-shot and fine-tuned variants.

**EPF subsets.** The Electricity Price Forecasting (EPF) dataset is often treated as a collection of distinct sub-datasets representing different markets: BE, DE, FR, NP, and PJM. These subsets exhibit diverse distributional characteristics and covariate dynamics. Table 9 and figure 7 present the average performance as well as the relative performance, including MAE, MAPE, MSE, and CRPS.

Table 8: Forecasting results on 12 unimodal covariate datasets.

| | | PatchTST | DeepAR | TFT | TiDE | NBEATSx | TimeXer | TTM | Moirai | TabPFN-TS | ChronosX | Chronos (Bolt) | | | | TimesFM | | | | Time-MoE | |
|---|---|---|---|---|---|---|---|---|---|---|---|---|---|---|---|---|---|---|---|---|---|
| | | | | | | | | | | | | ZS | SFT | LR | UniCA | ZS | SFT | LR | UniCA | ZS | UniCA |
| **Average** | Avg | 0.573 | 0.743 | 0.535 | 0.588 | 0.554 | 0.732 | 0.556 | 0.539 | 0.536 | 0.518 | 0.472 | 0.480 | 0.494 | 0.457 | 0.473 | 0.539 | 0.493 | 0.472 | 0.912 | 0.808 |
| | MAE | 0.616 | 0.806 | 0.596 | 0.640 | 0.600 | 0.743 | 0.595 | 0.604 | 0.590 | 0.527 | 0.521 | 0.526 | 0.540 | 0.509 | 0.530 | 0.587 | 0.546 | 0.526 | 0.843 | 0.796 |
| | MAPE | 0.633 | 0.718 | 0.596 | 0.649 | 0.608 | 0.738 | 0.592 | 0.593 | 0.600 | 0.580 | 0.522 | 0.514 | 0.569 | 0.506 | 0.523 | 0.598 | 0.557 | 0.514 | 0.946 | 0.938 |
| | MSE | 0.531 | 0.726 | 0.449 | 0.525 | 0.482 | 0.752 | 0.440 | 0.462 | 0.467 | 0.471 | 0.403 | 0.418 | 0.413 | 0.383 | 0.402 | 0.466 | 0.405 | 0.403 | 0.971 | 0.643 |
| | CRPS | 0.510 | 0.722 | 0.499 | 0.537 | 0.525 | 0.696 | 0.598 | 0.498 | 0.488 | 0.494 | 0.441 | 0.460 | 0.456 | 0.429 | 0.437 | 0.506 | 0.463 | 0.445 | 0.887 | 0.853 |
| **bull** | Average | 0.758 | 0.796 | 0.794 | 0.795 | 0.797 | 0.849 | 0.783 | 0.701 | 0.776 | 0.782 | 0.722 | 0.719 | 0.710 | 0.690 | 0.716 | 0.723 | 0.715 | 0.739 | 0.892 | 0.848 |
| | MAE | 0.871 | 0.924 | 0.902 | 0.881 | 0.886 | 0.948 | 0.869 | 0.857 | 0.874 | 0.862 | 0.835 | 0.825 | 0.815 | 0.809 | 0.842 | 0.837 | 0.831 | 0.876 | 0.968 | 0.923 |
| | MAPE | 0.841 | 0.730 | 0.809 | 0.836 | 0.850 | 0.916 | 0.785 | 0.609 | 0.762 | 0.775 | 0.671 | 0.686 | 0.710 | 0.639 | 0.679 | 0.697 | 0.720 | 0.668 | 0.962 | 0.914 |
| | MSE | 0.693 | 0.826 | 0.809 | 0.786 | 0.775 | 0.798 | 0.701 | 0.723 | 0.835 | 0.841 | 0.753 | 0.744 | 0.713 | 0.700 | 0.730 | 0.728 | 0.696 | 0.757 | 0.823 | 0.763 |
| | CRPS | 0.629 | 0.705 | 0.656 | 0.676 | 0.677 | 0.734 | 0.775 | 0.616 | 0.634 | 0.650 | 0.628 | 0.622 | 0.603 | 0.610 | 0.611 | 0.631 | 0.613 | 0.655 | 0.814 | 0.790 |
| **cockatoo** | Average | 0.820 | 0.816 | 0.777 | 0.892 | 0.968 | 1.934 | 0.920 | 0.824 | 0.939 | 0.882 | 0.823 | 0.953 | 0.830 | 0.822 | 0.818 | 1.403 | 0.805 | 0.838 | 0.965 | 0.945 |
| | MAE | 0.869 | 0.865 | 0.827 | 0.941 | 0.972 | 1.738 | 0.926 | 0.876 | 0.971 | 0.913 | 0.876 | 0.977 | 0.871 | 0.874 | 0.875 | 1.399 | 0.852 | 0.886 | 0.972 | 0.957 |
| | MAPE | 0.836 | 0.853 | 0.800 | 0.861 | 0.906 | 1.498 | 0.815 | 0.831 | 0.936 | 0.864 | 0.803 | 0.863 | 0.795 | 0.820 | 0.803 | 1.364 | 0.782 | 0.812 | 0.891 | 0.874 |
| | MSE | 0.741 | 0.732 | 0.683 | 0.850 | 1.002 | 2.835 | 0.822 | 0.761 | 0.931 | 0.826 | 0.755 | 0.927 | 0.774 | 0.746 | 0.769 | 1.440 | 0.729 | 0.784 | 0.905 | 0.879 |
| | CRPS | 0.835 | 0.814 | 0.799 | 0.915 | 0.993 | 1.666 | 1.116 | 0.827 | 0.917 | 0.924 | 0.857 | 1.045 | 0.881 | 0.848 | 0.825 | 1.409 | 0.859 | 0.870 | 1.093 | 1.072 |
| **COVID19** | Average | 0.377 | 0.221 | 0.186 | 0.324 | 0.219 | 0.272 | 0.235 | 0.231 | 0.264 | 0.166 | 0.144 | 0.140 | 0.170 | 0.144 | 0.143 | 0.169 | 0.168 | 0.143 | 0.730 | 0.764 |
| | MAE | 0.444 | 0.271 | 0.229 | 0.388 | 0.259 | 0.319 | 0.267 | 0.277 | 0.313 | 0.204 | 0.178 | 0.171 | 0.210 | 0.178 | 0.179 | 0.207 | 0.207 | 0.178 | 0.804 | 0.834 |
| | MAPE | 0.458 | 0.277 | 0.229 | 0.394 | 0.257 | 0.315 | 0.266 | 0.272 | 0.311 | 0.205 | 0.178 | 0.172 | 0.212 | 0.179 | 0.179 | 0.207 | 0.208 | 0.178 | 0.804 | 0.843 |
| | MSE | 0.191 | 0.078 | 0.061 | 0.146 | 0.092 | 0.131 | 0.082 | 0.104 | 0.142 | 0.054 | 0.041 | 0.038 | 0.052 | 0.041 | 0.040 | 0.048 | 0.052 | 0.040 | 0.564 | 0.594 |
| | CRPS | 0.414 | 0.258 | 0.225 | 0.367 | 0.267 | 0.323 | 0.325 | 0.270 | 0.291 | 0.202 | 0.180 | 0.177 | 0.205 | 0.178 | 0.174 | 0.213 | 0.204 | 0.177 | 0.750 | 0.785 |
| **EPF** | Avg | 0.506 | 0.638 | 0.542 | 0.577 | 0.489 | 0.626 | 0.539 | 0.574 | 0.528 | 0.577 | 0.465 | 0.423 | 0.450 | 0.434 | 0.449 | 0.465 | 0.436 | 0.440 | 0.762 | 0.767 |
| | MAE | 0.526 | 0.875 | 0.574 | 0.622 | 0.514 | 0.663 | 0.546 | 0.626 | 0.541 | 0.499 | 0.461 | 0.442 | 0.469 | 0.440 | 0.459 | 0.487 | 0.468 | 0.452 | 0.826 | 0.826 |
| | MAPE | 0.740 | 0.534 | 0.750 | 0.797 | 0.688 | 0.897 | 0.779 | 0.772 | 0.808 | 0.705 | 0.725 | 0.597 | 0.654 | 0.645 | 0.657 | 0.655 | 0.589 | 0.643 | 1.007 | 1.031 |
| | MSE | 0.413 | 0.554 | 0.466 | 0.476 | 0.407 | 0.507 | 0.416 | 0.482 | 0.412 | 0.771 | 0.365 | 0.357 | 0.368 | 0.356 | 0.378 | 0.397 | 0.376 | 0.364 | 0.623 | 0.613 |
| | CRPS | 0.345 | 0.589 | 0.377 | 0.412 | 0.346 | 0.439 | 0.415 | 0.418 | 0.352 | 0.335 | 0.307 | 0.295 | 0.309 | 0.293 | 0.300 | 0.320 | 0.311 | 0.302 | 0.594 | 0.596 |
| **GFC12** | Average | 0.522 | 0.817 | 0.646 | 0.568 | 0.511 | 0.571 | 0.578 | 0.521 | 0.534 | 0.501 | 0.479 | 0.469 | 0.599 | 0.459 | 0.476 | 0.490 | 0.537 | 0.468 | 0.815 | 0.818 |
| | MAE | 0.574 | 0.880 | 0.694 | 0.620 | 0.559 | 0.625 | 0.591 | 0.571 | 0.585 | 0.546 | 0.525 | 0.510 | 0.548 | 0.505 | 0.523 | 0.535 | 0.536 | 0.512 | 0.838 | 0.841 |
| | MAPE | 0.616 | 0.751 | 0.764 | 0.661 | 0.601 | 0.649 | 0.640 | 0.624 | 0.619 | 0.595 | 0.576 | 0.560 | 0.987 | 0.559 | 0.575 | 0.601 | 0.767 | 0.564 | 0.872 | 0.881 |
| | MSE | 0.345 | 0.729 | 0.471 | 0.391 | 0.327 | 0.396 | 0.368 | 0.346 | 0.375 | 0.335 | 0.306 | 0.302 | 0.323 | 0.285 | 0.311 | 0.304 | 0.317 | 0.299 | 0.670 | 0.670 |
| | CRPS | 0.551 | 0.908 | 0.654 | 0.600 | 0.557 | 0.613 | 0.715 | 0.543 | 0.558 | 0.526 | 0.511 | 0.502 | 0.536 | 0.487 | 0.497 | 0.519 | 0.529 | 0.499 | 0.879 | 0.882 |
| **GFC14** | Average | 0.349 | 1.279 | 0.361 | 0.376 | 0.324 | 0.268 | 0.398 | 0.362 | 0.361 | 0.339 | 0.331 | 0.324 | 0.344 | 0.319 | 0.320 | 0.330 | 0.337 | 0.321 | 0.707 | 0.727 |
| | MAE | 0.391 | 1.279 | 0.405 | 0.421 | 0.364 | 0.325 | 0.438 | 0.413 | 0.403 | 0.027 | 0.368 | 0.363 | 0.385 | 0.361 | 0.361 | 0.368 | 0.375 | 0.359 | 0.059 | 0.059 |
| | MAPE | 0.446 | 1.201 | 0.449 | 0.454 | 0.405 | 0.313 | 0.497 | 0.440 | 0.449 | 0.701 | 0.401 | 0.401 | 0.428 | 0.395 | 0.401 | 0.423 | 0.425 | 0.402 | 1.463 | 1.485 |
| | MSE | 0.181 | 1.204 | 0.198 | 0.212 | 0.168 | 0.115 | 0.223 | 0.195 | 0.202 | 0.001 | 0.177 | 0.169 | 0.186 | 0.166 | 0.168 | 0.168 | 0.181 | 0.168 | 0.004 | 0.004 |
| | CRPS | 0.376 | 1.434 | 0.393 | 0.417 | 0.361 | 0.320 | 0.434 | 0.397 | 0.392 | 0.628 | 0.367 | 0.364 | 0.378 | 0.354 | 0.351 | 0.362 | 0.368 | 0.354 | 1.303 | 1.359 |
| **GFC17** | Average | 0.288 | 0.809 | 0.320 | 0.312 | 0.286 | 0.340 | 0.303 | 0.302 | 0.284 | 0.251 | 0.240 | 0.238 | 0.270 | 0.241 | 0.242 | 0.274 | 0.262 | 0.237 | 0.772 | 0.758 |
| | MAE | 0.346 | 0.856 | 0.377 | 0.369 | 0.339 | 0.402 | 0.338 | 0.363 | 0.341 | 0.299 | 0.287 | 0.283 | 0.319 | 0.290 | 0.295 | 0.328 | 0.312 | 0.285 | 0.817 | 0.801 |
| | MAPE | 0.346 | 0.820 | 0.374 | 0.372 | 0.337 | 0.392 | 0.328 | 0.351 | 0.330 | 0.297 | 0.284 | 0.280 | 0.321 | 0.286 | 0.289 | 0.324 | 0.309 | 0.282 | 0.815 | 0.813 |
| | MSE | 0.124 | 0.623 | 0.155 | 0.144 | 0.127 | 0.172 | 0.125 | 0.141 | 0.131 | 0.104 | 0.097 | 0.092 | 0.115 | 0.096 | 0.097 | 0.116 | 0.106 | 0.094 | 0.601 | 0.590 |
| | CRPS | 0.337 | 0.938 | 0.372 | 0.362 | 0.343 | 0.393 | 0.421 | 0.355 | 0.334 | 0.302 | 0.293 | 0.295 | 0.325 | 0.292 | 0.289 | 0.328 | 0.322 | 0.287 | 0.854 | 0.826 |
| **Hog** | Average | 0.790 | 1.553 | 0.839 | 0.837 | 0.880 | 1.792 | 0.916 | 0.873 | 0.835 | 0.807 | 0.771 | 0.798 | 0.820 | 0.716 | 0.760 | 0.828 | 0.802 | 0.766 | 1.241 | 0.967 |
| | MAE | 0.856 | 1.518 | 0.893 | 0.891 | 0.914 | 1.480 | 0.933 | 0.951 | 0.897 | 0.872 | 0.838 | 0.853 | 0.889 | 0.781 | 0.836 | 0.880 | 0.868 | 0.836 | 1.203 | 0.976 |
| | MAPE | 0.786 | 1.173 | 0.859 | 0.868 | 0.906 | 1.547 | 0.814 | 0.806 | 0.839 | 0.706 | 0.725 | 0.706 | 0.781 | 0.683 | 0.724 | 0.814 | 0.789 | 0.726 | 1.012 | 1.024 |
| | MSE | 0.767 | 2.209 | 0.822 | 0.807 | 0.863 | 2.145 | 0.902 | 0.920 | 0.824 | 0.864 | 0.808 | 0.853 | 0.817 | 0.701 | 0.757 | 0.814 | 0.771 | 0.758 | 1.554 | 0.890 |
| | CRPS | 0.751 | 1.311 | 0.784 | 0.782 | 0.835 | 1.994 | 1.015 | 0.816 | 0.781 | 0.785 | 0.750 | 0.782 | 0.794 | 0.700 | 0.723 | 0.803 | 0.781 | 0.746 | 1.195 | 0.980 |
| **M5** | Avg | 1.127 | 0.692 | 0.624 | 0.902 | 0.936 | 0.646 | 0.629 | 0.724 | 0.680 | 0.663 | 0.632 | 0.606 | 0.629 | 0.613 | 0.608 | 0.629 | 0.609 | 0.613 | 0.739 | 0.707 |
| | MAE | 1.033 | 0.746 | 0.728 | 0.911 | 1.000 | 0.745 | 0.776 | 0.793 | 0.759 | 0.746 | 0.710 | 0.690 | 0.714 | 0.699 | 0.705 | 0.715 | 0.714 | 0.701 | 0.873 | 0.831 |
| | MAPE | 1.085 | 0.836 | 0.713 | 0.878 | 1.000 | 0.722 | 0.654 | 0.885 | 0.801 | 0.781 | 0.788 | 0.761 | 0.787 | 0.764 | 0.731 | 0.775 | 0.719 | 0.737 | 0.694 | 0.747 |
| | MSE | 1.747 | 0.695 | 0.617 | 1.254 | 1.000 | 0.657 | 0.588 | 0.732 | 0.705 | 0.677 | 0.598 | 0.555 | 0.605 | 0.566 | 0.579 | 0.605 | 0.578 | 0.587 | 0.770 | 0.647 |
| | CRPS | 0.643 | 0.489 | 0.438 | 0.565 | 0.742 | 0.460 | 0.500 | 0.488 | 0.454 | 0.448 | 0.434 | 0.416 | 0.422 | 0.424 | 0.419 | 0.423 | 0.426 | 0.426 | 0.619 | 0.603 |
| **pdb** | Average | 0.260 | 0.242 | 0.272 | 0.278 | 0.240 | 0.314 | 0.230 | 0.238 | 0.234 | 0.198 | 0.184 | 0.229 | 0.205 | 0.190 | 0.181 | 0.227 | 0.209 | 0.180 | 0.726 | 0.742 |
| | MAE | 0.313 | 0.296 | 0.332 | 0.334 | 0.288 | 0.377 | 0.262 | 0.288 | 0.282 | 0.240 | 0.223 | 0.267 | 0.249 | 0.228 | 0.222 | 0.274 | 0.254 | 0.221 | 0.783 | 0.787 |
| | MAPE | 0.306 | 0.279 | 0.318 | 0.332 | 0.276 | 0.353 | 0.251 | 0.275 | 0.269 | 0.231 | 0.214 | 0.258 | 0.241 | 0.220 | 0.212 | 0.264 | 0.245 | 0.210 | 0.797 | 0.812 |
| | MSE | 0.112 | 0.101 | 0.120 | 0.122 | 0.102 | 0.154 | 0.084 | 0.102 | 0.103 | 0.078 | 0.072 | 0.095 | 0.083 | 0.078 | 0.070 | 0.094 | 0.084 | 0.068 | 0.540 | 0.580 |
| | CRPS | 0.310 | 0.295 | 0.319 | 0.323 | 0.293 | 0.371 | 0.324 | 0.286 | 0.281 | 0.243 | 0.227 | 0.294 | 0.246 | 0.232 | 0.219 | 0.274 | 0.254 | 0.222 | 0.783 | 0.789 |
| **Retail** | Avg | 0.689 | 0.705 | 0.685 | 0.807 | 0.715 | 0.756 | 0.759 | 0.760 | 0.668 | 0.736 | 0.663 | 0.664 | 0.666 | 0.656 | 0.653 | 0.712 | 0.714 | 0.655 | 1.789 | 0.856 |
| | MAE | 0.732 | 0.757 | 0.762 | 0.862 | 0.779 | 0.809 | 0.788 | 0.816 | 0.729 | 0.764 | 0.712 | 0.708 | 0.729 | 0.712 | 0.706 | 0.755 | 0.767 | 0.709 | 1.135 | 0.885 |
| | MAPE | 0.690 | 0.785 | 0.676 | 0.907 | 0.751 | 0.788 | 0.869 | 0.850 | 0.702 | 0.755 | 0.684 | 0.670 | 0.638 | 0.655 | 0.681 | 0.805 | 0.783 | 0.648 | 1.162 | 0.978 |
| | MSE | 0.835 | 0.769 | 0.775 | 0.883 | 0.779 | 0.865 | 0.768 | 0.826 | 0.752 | 0.904 | 0.769 | 0.788 | 0.804 | 0.769 | 0.750 | 0.775 | 0.792 | 0.776 | 3.941 | 0.852 |
| | CRPS | 0.499 | 0.510 | 0.528 | 0.574 | 0.550 | 0.560 | 0.610 | 0.549 | 0.490 | 0.519 | 0.487 | 0.490 | 0.493 | 0.489 | 0.474 | 0.514 | 0.516 | 0.485 | 0.918 | 0.707 |
| **Spain** | Average | 0.387 | 0.348 | 0.374 | 0.387 | 0.282 | 0.419 | 0.386 | 0.360 | 0.332 | 0.316 | 0.203 | 0.193 | 0.208 | 0.196 | 0.309 | 0.239 | 0.316 | 0.267 | 0.802 | 0.792 |
| | MAE | 0.443 | 0.404 | 0.432 | 0.444 | 0.325 | 0.483 | 0.413 | 0.411 | 0.384 | 0.351 | 0.239 | 0.223 | 0.282 | 0.230 | 0.355 | 0.255 | 0.362 | 0.305 | 0.840 | 0.830 |
| | MAPE | 0.445 | 0.382 | 0.410 | 0.432 | 0.319 | 0.467 | 0.406 | 0.400 | 0.374 | 0.346 | 0.223 | 0.216 | 0.273 | 0.225 | 0.345 | 0.264 | 0.350 | 0.296 | 0.875 | 0.856 |
| | MSE | 0.223 | 0.189 | 0.213 | 0.224 | 0.145 | 0.245 | 0.199 | 0.216 | 0.195 | 0.201 | 0.093 | 0.089 | 0.124 | 0.089 | 0.175 | 0.099 | 0.183 | 0.147 | 0.655 | 0.632 |
| | CRPS | 0.437 | 0.416 | 0.441 | 0.449 | 0.340 | 0.482 | 0.524 | 0.415 | 0.374 | 0.365 | 0.246 | 0.243 | 0.283 | 0.239 | 0.362 | 0.279 | 0.370 | 0.318 | 0.839 | 0.850 |

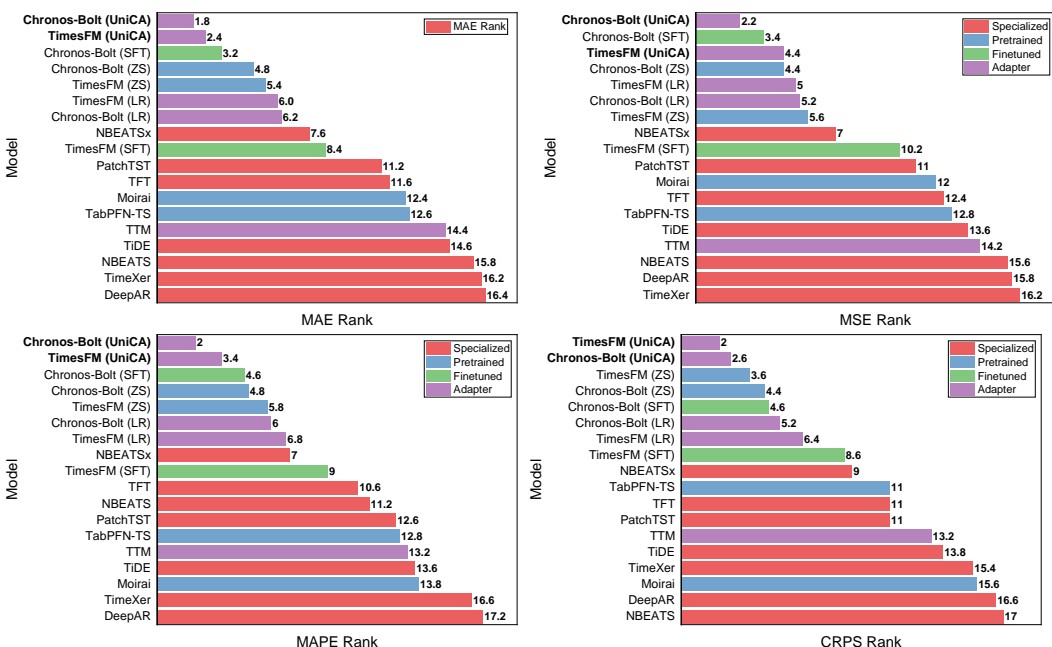

Figure 7: Metrics rank on EPF subsets.

We compare traditional deep forecasting models, pretrained time series foundation models (e.g., TimesFM and Chronos), and their adapted variants using zero-shot (ZS), supervised fine-tuning (SFT), linear adaptation (LR), and our unified covariate adapter (UniCA).

Across all EPF subsets, the UniCA adaptation strategy consistently delivers competitive or leading performance, especially when applied to pretrained models like Chronos and TimesFM. Notably, TimesFM (UniCA) and Chronos (UniCA) show particularly strong results on BE, PJM, and DE, indicating effective covariate adaptation and transferability across markets.

Table 9: Forecasting results on subsets of EPF.

| | | PatchTST | NBEATS | DeepAR | TFT | Tide | NBEATSx | TimeXer | Moirai | TTM | TabPFN-TS | Chronos-Bolt ZS | Chronos-Bolt SFT | Chronos-Bolt LR | Chronos-Bolt UniCA | TimesFM ZS | TimesFM SFT | TimesFM LR | TimesFM UniCA |
|---|---|---|---|---|---|---|---|---|---|---|---|---|---|---|---|---|---|---|---|
| **Average** | Average | 0.497 | 0.653 | 0.666 | 0.482 | 0.514 | 0.438 | 0.576 | 0.517 | 0.510 | 0.496 | 0.419 | 0.410 | 0.420 | **0.399** | 0.415 | 0.450 | 0.414 | 0.405 |
| | MAE | 0.559 | 0.759 | 0.771 | 0.543 | 0.590 | 0.496 | 0.647 | 0.596 | 0.558 | 0.555 | 0.467 | 0.456 | 0.477 | **0.450** | 0.467 | 0.509 | 0.471 | 0.458 |
| | MAPE | 0.598 | 0.564 | 0.780 | 0.563 | 0.600 | 0.519 | 0.698 | 0.594 | 0.600 | 0.591 | 0.507 | 0.495 | 0.498 | **0.476** | 0.500 | 0.532 | 0.489 | 0.487 |
| | MSE | 0.413 | 0.594 | 0.535 | 0.422 | 0.426 | 0.356 | 0.481 | 0.444 | 0.407 | 0.427 | 0.349 | 0.337 | 0.348 | **0.329** | 0.348 | 0.383 | 0.342 | 0.338 |
| | CRPS | 0.417 | 0.692 | 0.579 | 0.399 | 0.441 | 0.380 | 0.476 | 0.436 | 0.476 | 0.411 | 0.354 | 0.351 | 0.357 | 0.341 | 0.346 | 0.377 | 0.355 | **0.339** |
| **BE** | Average | 0.416 | 0.570 | 0.623 | 0.432 | 0.484 | 0.420 | 0.546 | 0.458 | 0.440 | 0.410 | 0.357 | 0.357 | 0.358 | 0.356 | 0.363 | 0.407 | 0.365 | 0.356 |
| | MAE | 0.518 | 0.704 | 0.819 | 0.536 | 0.605 | 0.526 | 0.681 | 0.574 | 0.546 | 0.519 | 0.452 | 0.451 | 0.453 | 0.451 | 0.457 | 0.513 | 0.456 | 0.450 |
| | MAPE | 0.488 | 0.640 | 0.724 | 0.461 | 0.572 | 0.482 | 0.669 | 0.520 | 0.497 | 0.479 | 0.399 | 0.399 | 0.399 | 0.391 | 0.398 | 0.430 | 0.400 | 0.391 |
| | MSE | 0.417 | 0.538 | 0.558 | 0.478 | 0.479 | 0.416 | 0.511 | 0.472 | 0.420 | 0.400 | **0.362** | 0.363 | 0.365 | 0.366 | 0.385 | 0.439 | 0.386 | 0.374 |
| | CRPS | 0.242 | 0.401 | 0.390 | 0.254 | 0.282 | 0.257 | 0.323 | 0.267 | 0.298 | 0.242 | 0.214 | 0.216 | 0.213 | 0.210 | 0.213 | 0.245 | 0.217 | 0.210 |
| **DE** | Average | 0.637 | 0.940 | 0.652 | 0.638 | 0.650 | 0.567 | 0.734 | 0.631 | 0.680 | 0.686 | 0.620 | 0.556 | 0.575 | 0.544 | 0.555 | 0.575 | **0.518** | 0.534 |
| | MAE | 0.630 | 1.124 | 0.648 | 0.649 | 0.690 | 0.589 | 0.727 | 0.669 | 0.650 | 0.679 | 0.604 | 0.557 | 0.592 | 0.556 | 0.567 | 0.587 | 0.557 | **0.553** |
| | MAPE | 0.861 | **0.276** | 0.862 | 0.836 | 0.776 | 0.706 | 0.948 | 0.754 | 0.875 | 0.916 | 0.832 | 0.735 | 0.724 | 0.702 | 0.740 | 0.751 | 0.633 | 0.701 |
| | MSE | 0.485 | 1.104 | 0.490 | 0.485 | 0.498 | 0.407 | 0.593 | 0.504 | 0.499 | 0.540 | 0.487 | 0.406 | 0.435 | 0.404 | 0.405 | 0.428 | 0.366 | 0.385 |
| | CRPS | 0.574 | 1.257 | 0.609 | 0.581 | 0.635 | 0.566 | 0.665 | 0.599 | 0.696 | 0.609 | 0.559 | 0.527 | 0.551 | 0.516 | 0.510 | 0.534 | 0.517 | **0.497** |
| **FR** | Average | 0.421 | 0.808 | 0.575 | 0.473 | 0.499 | 0.414 | 0.564 | 0.493 | 0.456 | 0.425 | **0.349** | 0.351 | 0.361 | 0.350 | 0.364 | 0.407 | 0.374 | 0.359 |
| | MAE | 0.488 | 0.937 | 0.672 | 0.539 | 0.582 | 0.472 | 0.661 | 0.576 | 0.521 | 0.493 | 0.398 | **0.392** | 0.412 | 0.394 | 0.412 | 0.464 | 0.426 | 0.408 |
| | MAPE | 0.450 | 0.853 | 0.724 | 0.485 | 0.548 | 0.440 | 0.634 | 0.533 | 0.472 | 0.468 | 0.360 | 0.356 | 0.371 | 0.352 | 0.366 | 0.409 | 0.378 | 0.362 |
| | MSE | 0.421 | 0.676 | 0.450 | 0.515 | 0.476 | 0.417 | 0.521 | 0.482 | 0.426 | 0.410 | **0.366** | 0.382 | 0.385 | 0.384 | 0.403 | 0.441 | 0.405 | 0.394 |
| | CRPS | 0.325 | 0.765 | 0.451 | 0.355 | 0.388 | 0.327 | 0.441 | 0.381 | 0.403 | 0.328 | **0.270** | 0.271 | 0.275 | 0.271 | 0.276 | 0.314 | 0.288 | 0.273 |
| **NP** | Average | 0.704 | 0.570 | 0.809 | 0.543 | 0.602 | 0.498 | 0.641 | 0.679 | 0.644 | 0.632 | 0.489 | 0.507 | 0.529 | **0.476** | 0.517 | 0.564 | 0.531 | 0.501 |
| | MAE | 0.776 | 0.588 | 0.912 | 0.581 | 0.650 | 0.528 | 0.682 | 0.749 | 0.670 | 0.673 | 0.523 | 0.529 | 0.555 | **0.506** | 0.546 | 0.604 | 0.555 | 0.528 |
| | MAPE | 0.828 | 0.653 | 0.892 | 0.661 | 0.709 | 0.624 | 0.781 | 0.790 | 0.771 | 0.716 | **0.612** | 0.653 | 0.662 | 0.614 | 0.665 | 0.707 | 0.689 | 0.648 |
| | MSE | 0.556 | 0.418 | 0.642 | 0.430 | 0.465 | 0.367 | 0.514 | 0.552 | 0.496 | 0.569 | 0.364 | 0.374 | 0.398 | **0.341** | 0.387 | 0.431 | 0.397 | 0.375 |
| | CRPS | 0.656 | 0.622 | 0.792 | 0.500 | 0.586 | 0.474 | 0.587 | 0.625 | 0.638 | 0.570 | 0.455 | 0.472 | 0.482 | **0.445** | 0.469 | 0.513 | 0.482 | 0.453 |
| **PJM** | Average | 0.305 | 0.374 | 0.672 | 0.322 | 0.335 | 0.289 | 0.393 | 0.325 | 0.331 | 0.326 | 0.281 | 0.277 | 0.278 | **0.268** | 0.277 | 0.299 | 0.283 | 0.277 |
| | MAE | 0.386 | 0.444 | 0.804 | 0.409 | 0.421 | 0.363 | 0.485 | 0.412 | 0.401 | 0.410 | 0.355 | 0.353 | 0.354 | **0.340** | 0.352 | 0.378 | 0.360 | 0.352 |
| | MAPE | 0.363 | 0.401 | 0.697 | 0.371 | 0.397 | 0.344 | 0.459 | 0.373 | 0.385 | 0.373 | 0.332 | 0.330 | 0.337 | **0.321** | 0.333 | 0.363 | 0.345 | 0.333 |
| | MSE | 0.184 | 0.235 | 0.537 | 0.204 | 0.210 | 0.172 | 0.265 | 0.210 | 0.195 | 0.215 | 0.166 | 0.156 | 0.157 | **0.150** | 0.162 | 0.175 | 0.156 | 0.162 |
| | CRPS | 0.286 | 0.415 | 0.651 | 0.305 | 0.313 | 0.278 | 0.363 | 0.306 | 0.345 | 0.305 | 0.271 | 0.268 | 0.265 | **0.260** | 0.262 | 0.279 | 0.272 | 0.262 |

## E.3 MULTI-MODAL FORECASTING

**Time-MMD (TS-Text).** Table 10 reports the full forecasting results on the Time-MMD benchmark across seven domains: *Climate*, *Energy*, *Environment*, *Health*, *Security*, *SocialGood*, and *Traf-*

*fic*. We compare UniCA with strong baselines, including classic models (e.g., DeepAR, TFT), recent pretrained models (e.g., Chronos-Bolt, TimesFM), and domain-specific models (e.g., Time-LLM, TTM, Moirai). Metrics include Mean Absolute Error (MAE), Mean Absolute Percentage Error (MAPE), Mean Squared Error (MSE), and Continuous Ranked Probability Score (CRPS).

UniCA consistently achieves the best or competitive performance across most domains and metrics, demonstrating its versatility and scalability in handling heterogeneous covariates under diverse multimodal forecasting scenarios.

Table 10: Full forecasting results on Time-MMD.

| | NBEATSx | PatchTST | DeepAR | TFT | TFT(+CH) | TiDE | TiDE(+CH) | Time-LLM | TTM | Moirai | TabPFN-TS | MM-TSF | ChatTime | Chronos (Bolt) ZS | SFT | UniCA | TimesFM ZS | SFT | UniCA | Time-MOE ZS | UniCA |
|---|---|---|---|---|---|---|---|---|---|---|---|---|---|---|---|---|---|---|---|---|---|
| **Average** Average | 0.840 | 0.894 | 1.270 | 0.971 | 0.934 | 0.920 | 0.889 | 0.818 | 0.778 | 0.742 | 0.801 | 0.691 | 1.213 | 0.682 | 0.738 | 0.671 | 0.652 | 0.804 | 0.638 | 0.821 | 0.815 |
| MAE | 0.884 | 1.009 | 1.219 | 0.958 | 0.928 | 0.976 | 0.952 | 0.847 | 0.866 | 0.821 | 0.837 | 0.777 | 0.647 | 0.737 | 0.794 | 0.742 | 0.716 | 0.869 | 0.703 | 0.918 | 0.918 |
| MAPE | 0.717 | 0.778 | 0.998 | 1.042 | 1.035 | 0.900 | 0.856 | 0.766 | 0.651 | 0.715 | 0.819 | 0.655 | 2.227 | 0.674 | 0.721 | 0.634 | 0.646 | 0.776 | 0.601 | 0.648 | 0.634 |
| MSE | 0.782 | 0.793 | 1.605 | 0.992 | 0.885 | 0.869 | 0.813 | 0.723 | 0.685 | 0.696 | 0.787 | 0.618 | 1.361 | 0.637 | 0.647 | 0.614 | 0.608 | 0.732 | 0.604 | 0.781 | 0.769 |
| CRPS | 0.980 | 0.996 | 1.260 | 0.891 | 0.886 | 0.937 | 0.932 | 0.935 | 0.909 | 0.735 | 0.762 | 0.727 | 0.618 | 0.681 | 0.791 | 0.693 | 0.640 | 0.841 | 0.645 | 0.937 | 0.938 |
| **Climate** Average | 0.628 | 0.720 | 0.815 | 0.708 | 0.800 | 0.573 | 0.578 | 0.599 | 0.536 | 0.545 | 0.536 | 0.509 | 1.799 | 0.505 | 0.626 | 0.515 | 0.529 | 0.634 | 0.533 | 0.555 | 0.559 |
| MAE | 0.712 | 0.788 | 0.779 | 0.768 | 0.885 | 0.685 | 0.682 | 0.687 | 0.635 | 0.706 | 0.638 | 0.635 | 1.000 | 0.622 | 0.685 | 0.624 | 0.635 | 0.700 | 0.635 | 0.714 | 0.714 |
| MAPE | 0.510 | 0.710 | 1.048 | 0.744 | 0.718 | 0.569 | 0.598 | 0.498 | 0.566 | 0.393 | 0.568 | 0.406 | 3.610 | 0.462 | 0.682 | 0.466 | 0.541 | 0.661 | 0.542 | 0.255 | 0.259 |
| MSE | 0.519 | 0.640 | 0.623 | 0.599 | 0.819 | 0.465 | 0.461 | 0.468 | 0.408 | 0.488 | 0.407 | 0.418 | 1.576 | 0.397 | 0.481 | 0.412 | 0.403 | 0.500 | 0.402 | 0.511 | 0.511 |
| CRPS | 0.773 | 0.743 | 0.809 | 0.719 | 0.778 | 0.574 | 0.571 | 0.746 | 0.535 | 0.593 | 0.529 | 0.571 | 1.010 | 0.537 | 0.656 | 0.557 | 0.535 | 0.676 | 0.552 | 0.739 | 0.753 |
| **Energy** Average | 1.475 | 1.144 | 2.981 | 0.995 | 0.952 | 1.209 | 0.923 | 1.204 | 1.120 | 1.039 | 1.167 | 0.896 | 0.790 | 0.925 | 0.894 | 0.932 | 0.918 | 1.330 | 0.904 | 1.162 | 1.109 |
| MAE | 1.429 | 1.252 | 2.368 | 1.004 | 0.987 | 1.138 | 0.974 | 1.161 | 1.042 | 1.035 | 1.163 | 0.944 | 0.237 | 0.938 | 0.957 | 0.947 | 0.946 | 1.302 | 0.957 | 1.199 | 1.146 |
| MAPE | 1.068 | 0.753 | 0.621 | 0.926 | 0.883 | 0.928 | 0.467 | 1.056 | 0.834 | 1.123 | 0.970 | 0.849 | 2.543 | 0.963 | 0.730 | 0.968 | 0.950 | 1.292 | 0.795 | 0.847 | 0.834 |
| MSE | 1.706 | 1.305 | 6.328 | 1.047 | 0.947 | 1.391 | 0.924 | 1.217 | 1.019 | 1.024 | 1.370 | 0.872 | 0.111 | 0.889 | 0.859 | 0.886 | 0.886 | 1.454 | 0.934 | 1.283 | 1.203 |
| CRPS | 1.699 | 1.266 | 2.607 | 1.004 | 0.989 | 1.379 | 1.326 | 1.380 | 1.587 | 0.975 | 1.167 | 0.920 | 0.268 | 0.911 | 1.031 | 0.929 | 0.888 | 1.271 | 0.928 | 1.320 | 1.251 |
| **Environment** Average | 0.721 | 0.665 | 0.721 | 0.670 | 0.676 | 0.640 | 0.644 | 0.707 | 0.630 | 0.658 | 0.667 | 0.655 | 0.928 | 0.626 | 0.690 | 0.612 | 0.643 | 0.707 | 0.642 | 0.717 | 0.712 |
| MAE | 0.809 | 0.785 | 0.822 | 0.763 | 0.766 | 0.778 | 0.773 | 0.774 | 0.777 | 0.756 | 0.772 | 0.742 | 0.528 | 0.740 | 0.779 | 0.728 | 0.753 | 0.799 | 0.756 | 0.830 | 0.826 |
| MAPE | 0.706 | 0.731 | 0.819 | 0.766 | 0.776 | 0.646 | 0.667 | 0.733 | 0.586 | 0.710 | 0.735 | 0.597 | 3.650 | 0.674 | 0.821 | 0.656 | 0.605 | 0.627 | 0.599 | 0.615 | 0.610 |
| MSE | 0.628 | 0.589 | 0.648 | 0.601 | 0.612 | 0.572 | 0.574 | 0.617 | 0.546 | 0.623 | 0.611 | 0.597 | 0.583 | 0.562 | 0.615 | 0.533 | 0.605 | 0.627 | 0.599 | 0.615 | 0.610 |
| CRPS | 0.739 | 0.558 | 0.596 | 0.550 | 0.551 | 0.564 | 0.562 | 0.707 | 0.609 | 0.543 | 0.550 | 0.539 | 0.623 | 0.538 | 0.574 | 0.525 | 0.540 | 0.582 | 0.556 | 0.734 | 0.732 |
| **Health** Average | 0.834 | 0.844 | 1.070 | 0.984 | 0.867 | 0.913 | 0.938 | 0.746 | 0.894 | 0.760 | 1.089 | 0.642 | 1.582 | 0.616 | 0.828 | 0.624 | 0.692 | 0.821 | 0.692 | 0.934 | 0.953 |
| MAE | 0.860 | 0.928 | 1.118 | 1.004 | 0.934 | 0.992 | 1.011 | 0.846 | 0.989 | 0.821 | 1.008 | 0.718 | 0.706 | 0.690 | 0.826 | 0.708 | 0.753 | 0.886 | 0.753 | 1.001 | 1.019 |
| MAPE | 0.715 | 0.586 | 0.886 | 0.895 | 0.701 | 0.733 | 0.766 | 0.398 | 0.678 | 0.710 | 1.448 | 0.612 | 3.650 | 0.674 | 0.873 | 0.684 | 0.711 | 0.712 | 0.684 | 0.750 | 0.782 |
| MSE | 0.739 | 0.874 | 1.023 | 1.059 | 0.883 | 0.916 | 0.943 | 0.735 | 0.906 | 0.722 | 0.964 | 0.511 | 1.164 | 0.475 | 0.699 | 0.487 | 0.603 | 0.742 | 0.600 | 0.939 | 0.941 |
| CRPS | 1.020 | 0.989 | 1.251 | 0.979 | 0.948 | 1.010 | 1.031 | 1.004 | 1.002 | 0.786 | 0.936 | 0.727 | 0.808 | 0.686 | 0.915 | 0.707 | 0.699 | 0.945 | 0.733 | 1.045 | 1.070 |
| **Security** Average | 0.835 | 1.208 | 1.448 | 1.469 | 1.256 | 1.555 | 1.540 | 0.878 | 0.739 | 0.718 | 0.649 | 0.819 | 1.933 | 0.697 | 0.823 | 0.694 | 0.694 | 0.669 | 0.593 | 0.685 | 0.686 |
| MAE | 0.927 | 1.332 | 1.607 | 1.409 | 1.142 | 1.767 | 1.752 | 0.951 | 0.880 | 0.856 | 0.764 | 0.987 | 1.077 | 0.822 | 0.966 | 0.788 | 0.826 | 0.790 | 0.707 | 0.815 | 0.814 |
| MAPE | 0.799 | 1.324 | 1.535 | 1.679 | 1.965 | 1.657 | 1.639 | 0.926 | 0.666 | 0.633 | 0.565 | 0.706 | 1.141 | 0.619 | 0.744 | 0.662 | 0.606 | 0.544 | 0.467 | 0.500 | 0.505 |
| MSE | 0.692 | 0.882 | 1.078 | 1.614 | 0.839 | 1.260 | 1.241 | 0.690 | 0.676 | 0.669 | 0.612 | 0.712 | 4.723 | 0.640 | 0.708 | 0.634 | 0.646 | 0.632 | 0.578 | 0.648 | 0.649 |
| CRPS | 0.922 | 1.295 | 1.571 | 1.175 | 1.078 | 1.535 | 1.528 | 0.946 | 0.732 | 0.714 | 0.657 | 0.872 | 0.791 | 0.707 | 0.876 | 0.693 | 0.697 | 0.710 | 0.621 | 0.778 | 0.777 |
| **SocialGood** Average | 0.823 | 1.118 | 1.429 | 1.378 | 1.380 | 0.998 | 1.044 | 1.069 | 0.898 | 0.725 | 0.882 | 0.778 | 0.936 | 0.838 | 0.781 | 0.818 | 0.653 | 1.024 | 0.664 | 0.856 | 0.900 |
| MAE | 0.843 | 1.347 | 1.403 | 1.172 | 1.166 | 0.943 | 0.947 | 1.036 | 1.062 | 0.803 | 0.912 | 0.840 | 0.530 | 0.804 | 0.798 | 0.806 | 0.686 | 1.154 | 0.699 | 0.969 | 1.077 |
| MAPE | 0.701 | 0.815 | 1.558 | 1.721 | 1.651 | 1.137 | 1.227 | 1.120 | 0.652 | 0.558 | 0.820 | 0.635 | 1.784 | 0.803 | 0.708 | 0.782 | 0.527 | 0.907 | 0.547 | 0.670 | 0.595 |
| MSE | 0.780 | 0.877 | 1.231 | 1.469 | 1.512 | 0.973 | 1.046 | 0.932 | 0.816 | 0.735 | 0.917 | 0.781 | 0.980 | 0.932 | 0.762 | 0.883 | 0.715 | 0.831 | 0.717 | 0.770 | 0.818 |
| CRPS | 0.967 | 1.434 | 1.523 | 1.150 | 1.189 | 0.941 | 0.958 | 1.188 | 1.061 | 0.804 | 0.881 | 0.855 | 0.450 | 0.813 | 0.855 | 0.803 | 0.685 | 1.204 | 0.694 | 1.016 | 1.110 |
| **Traffic** Average | 0.568 | 0.559 | 0.430 | 0.589 | 0.605 | 0.554 | 0.553 | 0.521 | 0.630 | 0.749 | 0.619 | 0.539 | 0.526 | 0.567 | 0.525 | 0.502 | 0.438 | 0.444 | 0.438 | 0.838 | 0.786 |
| MAE | 0.608 | 0.632 | 0.435 | 0.589 | 0.618 | 0.528 | 0.528 | 0.475 | 0.679 | 0.772 | 0.605 | 0.572 | 0.453 | 0.539 | 0.547 | 0.595 | 0.412 | 0.448 | 0.413 | 0.901 | 0.831 |
| MAPE | 0.518 | 0.529 | 0.517 | 0.563 | 0.548 | 0.630 | 0.627 | 0.630 | 0.577 | 0.881 | 0.629 | 0.545 | 0.885 | 0.596 | 0.520 | 0.307 | 0.510 | 0.493 | 0.512 | 0.825 | 0.780 |
| MSE | 0.408 | 0.385 | 0.305 | 0.552 | 0.583 | 0.506 | 0.504 | 0.401 | 0.428 | 0.610 | 0.631 | 0.437 | 0.388 | 0.561 | 0.402 | 0.466 | 0.398 | 0.336 | 0.399 | 0.700 | 0.653 |
| CRPS | 0.737 | 0.689 | 0.462 | 0.657 | 0.671 | 0.553 | 0.552 | 0.576 | 0.834 | 0.731 | 0.611 | 0.604 | 0.376 | 0.571 | 0.632 | 0.639 | 0.431 | 0.499 | 0.428 | 0.926 | 0.877 |

## E.4 IMPUTATION

UniCA is a general covariate adaptation framework for time-series foundation models (TSFMs). Its applicability is therefore not limited to forecasting—UniCA can be attached to any downstream task that the underlying TSFM supports, provided the task is covariate-aware.

To demonstrate this, we evaluate UniCA on the *imputation* setting of the MOMENT TSFM Goswami et al. (2024), which natively supports forecasting, classification, anomaly detection, and imputation. Among these tasks, only imputation naturally involves multivariate inputs or covariates, making it the most appropriate benchmark for UniCA.

For each dataset (ETTh1, ETTh2, ETTm1, ETTm2, Electricity, Weather), we treat the `OT` column as the target variable and use all remaining variables as past dynamic real covariates. Each dataset is split with a 6:2:2 ratio. Following the MOMENT imputation setup, on the test split we extract sliding windows of length 512 and randomly mask patches of length 8 using MOMENT's patch-based masking module at mask ratios $12.5\%, 25\%, 37.5\%, 50\%$. MOMENT is loaded in "reconstruction" mode and is never fine-tuned; its RevIN normalizer and reconstruction head remain fixed throughout.

UniCA operates on top of the frozen MOMENT encoder: the target context is tokenized by MOMENT; all covariates are homogenized and fused through UniCA's gated residual and attention modules; and the fused representation is passed directly into MOMENT's frozen reconstruction head. During training, we optimize *only* UniCA's parameters to minimize squared error on the masked target positions. MOMENT's weights remain completely frozen. At evaluation time, we compare zero-shot MOMENT and MOMENT+UniCA using the same patch masks and report MSE and MAE over the masked entries only. Results are shown in Table 12.

Across all six datasets and all four mask ratios, UniCA consistently improves imputation over zero-shot MOMENT model. Averaged over mask ratios, UniCA reduces MSE by roughly 33–35% on ETTh1/ETTh2, about 60–85% on ETTm1/ETTm2, 45% on electricity, and 40% on weather, with corresponding MAE reductions of about 20–25% (ETTh), 40–65% (ETTm), 25% (electric-

Table 11: Forecasting performance on MMSP (TS-Image multimodal) dataset.

| | | P.TST | NB.S | D.AR | TFT | TFT (+CH) | TiDE | TiDE (+CH) | TTM | Moirai | PFN | F.SF | MM-TSF | Chronos (Bolt) | | | TimesFM | | | Time-MOE | |
|---|---|---|---|---|---|---|---|---|---|---|---|---|---|---|---|---|---|---|---|---|---|
| | | | | | | | | | | | | | | ZS | SFT | UniCA | ZS | SFT | UniCA | ZS | UniCA |
| MMSP | Average | 0.485 | 0.152 | 0.137 | 0.112 | **0.103** | 0.292 | 0.135 | 0.408 | 0.238 | 0.555 | 0.378 | 0.127 | 0.120 | 0.140 | 0.121 | 0.158 | 0.167 | 0.147 | 0.703 | 0.607 |
| | MAE | 0.682 | 0.219 | 0.216 | 0.177 | **0.168** | 0.438 | 0.206 | 0.263 | 0.378 | 0.711 | 0.566 | 0.200 | 0.193 | 0.225 | 0.193 | 0.245 | 0.258 | 0.229 | 0.814 | 0.765 |
| | MAPE | 0.020 | 0.030 | 0.019 | 0.034 | 0.021 | 0.022 | **0.014** | 0.090 | 0.038 | 0.187 | 0.016 | 0.037 | 0.019 | 0.023 | 0.019 | 0.049 | 0.080 | 0.046 | 0.755 | 0.337 |
| | MSE | 0.662 | 0.107 | 0.097 | 0.067 | **0.067** | 0.297 | 0.100 | 0.095 | 0.206 | 0.633 | 0.478 | 0.090 | 0.090 | 0.099 | 0.090 | 0.113 | 0.100 | 0.098 | 0.538 | 0.653 |
| | CRPS | 0.576 | 0.252 | 0.214 | 0.168 | **0.157** | 0.410 | 0.220 | 1.183 | 0.329 | 0.691 | 0.452 | 0.183 | 0.180 | 0.212 | 0.180 | 0.227 | 0.231 | 0.215 | 0.707 | 0.672 |

Table 12: Imputation performance of zero-shot MOMENT and UniCA-adapted MOMENT on six ETT/electricity/weather benchmarks under four random patch-masking ratios (12.5%, 25%, 37.5%, 50%). Reported metrics are MSE and MAE on the masked target entries, together with percentage-point improvements of UniCA over the MOMENT baseline (negative values indicate lower error with UniCA).

| Dataset | Mask Ratio | MSE | | | MAE | | |
|---|---|---|---|---|---|---|---|
| | | MOMENT | UniCA | $\Delta(\%)$ | MOMENT | UniCA | $\Delta(\%)$ |
| ETTh1 | 12.5% | 0.016 | **0.012** | -23.9% | 0.093 | **0.079** | -14.5% |
| | 25.0% | 0.022 | **0.016** | -27.9% | 0.110 | **0.090** | -18.4% |
| | 37.5% | 0.030 | **0.019** | -35.4% | 0.128 | **0.100** | -21.6% |
| | 50.0% | 0.041 | **0.025** | -38.3% | 0.152 | **0.115** | -24.4% |
| | Average | 0.027 | **0.018** | -33.3% | 0.121 | **0.096** | -20.4% |
| ETTh2 | 12.5% | 0.042 | **0.026** | -37.8% | 0.150 | **0.108** | -27.6% |
| | 25.0% | 0.059 | **0.039** | -33.8% | 0.176 | **0.128** | -27.3% |
| | 37.5% | 0.086 | **0.059** | -31.8% | 0.212 | **0.154** | -27.0% |
| | 50.0% | 0.131 | **0.084** | -35.5% | 0.269 | **0.188** | -30.2% |
| | Average | 0.079 | **0.052** | -34.5% | 0.202 | **0.145** | -28.2% |
| ETTm1 | 12.5% | 0.007 | 0.003 | -48.4% | 0.057 | **0.039** | -31.3% |
| | 25.0% | 0.010 | **0.004** | -58.0% | 0.068 | **0.043** | -37.5% |
| | 37.5% | 0.015 | **0.005** | -62.9% | 0.083 | **0.048** | -42.1% |
| | 50.0% | 0.021 | **0.007** | -65.4% | 0.102 | **0.056** | -45.4% |
| | Average | 0.013 | **0.005** | -61.1% | 0.077 | **0.046** | -40.2% |
| ETTm2 | 12.5% | 0.015 | **0.002** | -83.1% | 0.080 | **0.032** | -59.6% |
| | 25.0% | 0.028 | **0.004** | -85.0% | 0.108 | **0.040** | -63.0% |
| | 37.5% | 0.051 | **0.007** | -86.2% | 0.148 | **0.051** | -65.6% |
| | 50.0% | 0.082 | **0.014** | -83.3% | 0.195 | **0.068** | -65.0% |
| | Average | 0.044 | **0.007** | -84.4% | 0.133 | **0.048** | -63.9% |
| electricity | 12.5% | 0.124 | **0.098** | -20.6% | 0.258 | **0.234** | -9.3% |
| | 25.0% | 0.159 | **0.110** | -30.4% | 0.290 | **0.248** | -14.4% |
| | 37.5% | 0.232 | **0.127** | -45.2% | 0.351 | **0.266** | -24.2% |
| | 50.0% | 0.376 | **0.151** | -59.9% | 0.458 | **0.288** | -37.0% |
| | Average | 0.223 | **0.122** | -45.4% | 0.339 | **0.259** | -23.6% |
| weather | 12.5% | 0.000 | **0.000** | -19.5% | 0.008 | **0.007** | -10.6% |
| | 25.0% | 0.000 | **0.000** | -35.0% | 0.009 | **0.008** | -19.4% |
| | 37.5% | 0.000 | **0.000** | -43.9% | 0.012 | **0.008** | -26.8% |
| | 50.0% | 0.000 | **0.000** | -45.2% | 0.014 | **0.010** | -30.2% |
| | Average | 0.000 | **0.000** | -39.5% | 0.011 | **0.008** | -23.3% |

ity), and 23% (weather). The gains are monotonic in the mask ratio: UniCA's relative improvement is smallest at 12.5% masking and largest at 50% masking on every dataset, showing that covariate-aware adaptation becomes increasingly beneficial as the imputation problem becomes harder.

## F  SHOWCASES

In this section, we present a detailed case study to demonstrate the effectiveness of our proposed UniCA framework. We analyze the feature attention affinity, important covariates identified by our model, and compare the prediction performance with and without UniCA adaptation.

Figure 8, 9, 10, 11 illustrate our analysis on two time series samples for each dataset, labeled as (a) and (b). The visualization is organized into three components for each sample: Important Features Visualization (top), Prediction Comparison (middle), and Feature Weights Visualization (bottom). The Important Features Visualization reveals how our model identifies and leverages key covariates during prediction. In figure 8 (a), covariate 18 demonstrates high importance (0.8323) while covariate 7 shows minimal contribution (0.0125). Similarly, in sample (b), covariate 18 maintains high importance (0.8586) while covariate 24 has low importance (0.0077). This selective attention mechanism enables UniCA to focus on the most relevant covariates for each specific forecasting task, effectively filtering out noise from less informative features.

The Prediction Comparison clearly demonstrates the superior performance of UniCA-adapted models compared to their non-adapted counterparts. The middle rows show predictions without UniCA adaptation, while the lower rows display predictions with UniCA adaptation. Both models generate prediction intervals (10%-90%), but the UniCA-adapted model produces forecasts that align more closely with the ground truth values. Notably, the predictions with UniCA show better alignment with the temporal patterns and magnitude of the ground truth, particularly in the forecast horizon (the shaded area after the vertical dotted line).

The Feature Weights Visualization at the bottom provides insight into how attention is distributed across different feature dimensions and sequence positions. The heatmaps reveal that certain feature dimensions consistently receive higher attention weights (shown in brighter yellow), indicating their greater influence on the final prediction. These patterns vary between samples (a) and (b), highlighting UniCA's ability to adapt dynamically to different time series characteristics.

Our case study demonstrates that UniCA effectively identifies important covariates through its attention mechanism and significantly improves prediction accuracy by incorporating this covariate information. The comparison between adapted and non-adapted models confirms that UniCA successfully bridges TSFMs with covariate-aware forecasting while preserving the foundation model's generalization capabilities. Furthermore, the feature weight visualizations provide interpretability insights, showing which dimensions and temporal positions are most influential for specific forecasting tasks.

## G  MORE ABLATION

### G.1  FUSION POSITION.

To further understand the impact of fusion positions in integrating covariate information, we conduct an ablation study by varying where the past and future covariates are fused within the TSFM pipeline. Specifically, we compare four fusion strategies: `Pre-Pre`, `Post-Pre`, `Post-Post`, and `Pre-Post` (our default setting). These denote whether past and future covariates are fused before (`Pre`) or after (`Post`) the temporal encoder.

As shown in Figure 12, the results indicate that the choice of fusion position has a relatively minor impact on the overall forecasting performance. On TimesFM, all variants achieve similar performance, with the aggregated error ranging from 0.472 to 0.476. Interestingly, although `Post-Post` slightly underperforms the others, the differences remain marginal. On Chronos-Bolt, all configurations perform nearly identically, with `Post-Post` achieving the lowest error (0.455), reinforcing the robustness of our fusion design. These findings suggest that while the timing of fusion can affect the model's attention mechanism and how it contextualizes covariate information, UniCA re-

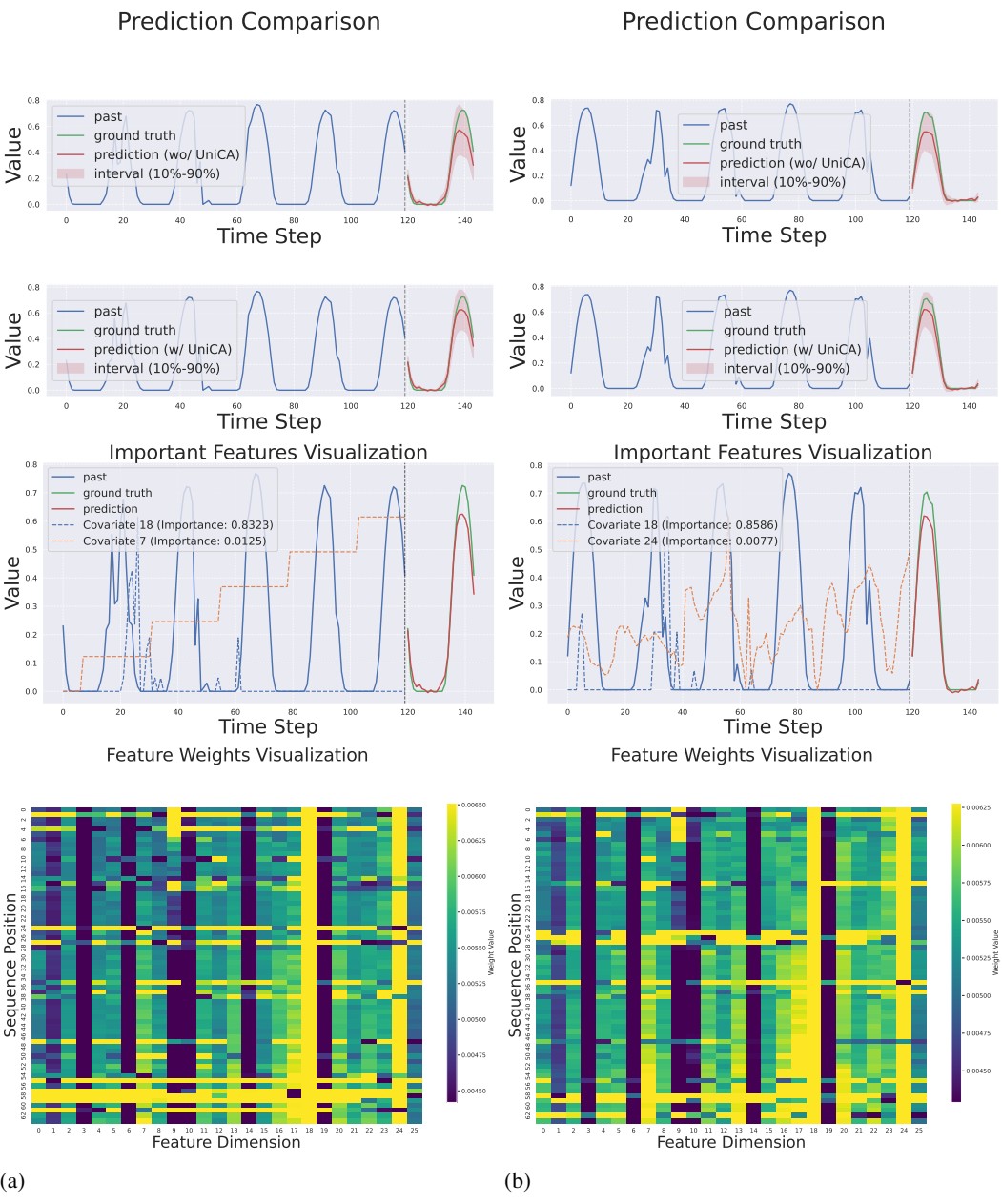

Figure 8: Visualization of prediction comparison, important covariates and attention map on MMSP

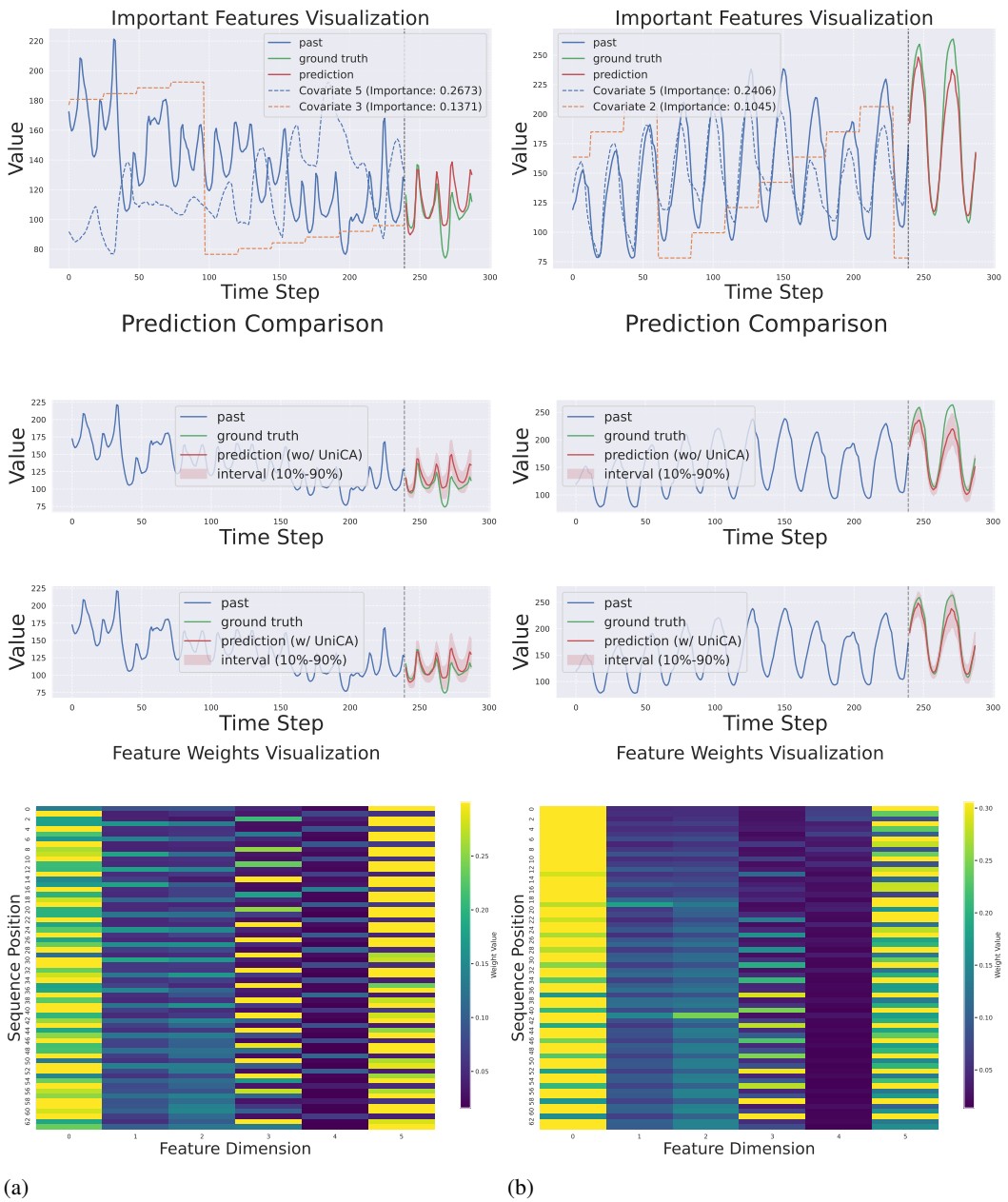

(a)                                             (b)

Figure 9: Visualization of prediction comparison, important covariates and attention map on GFC14

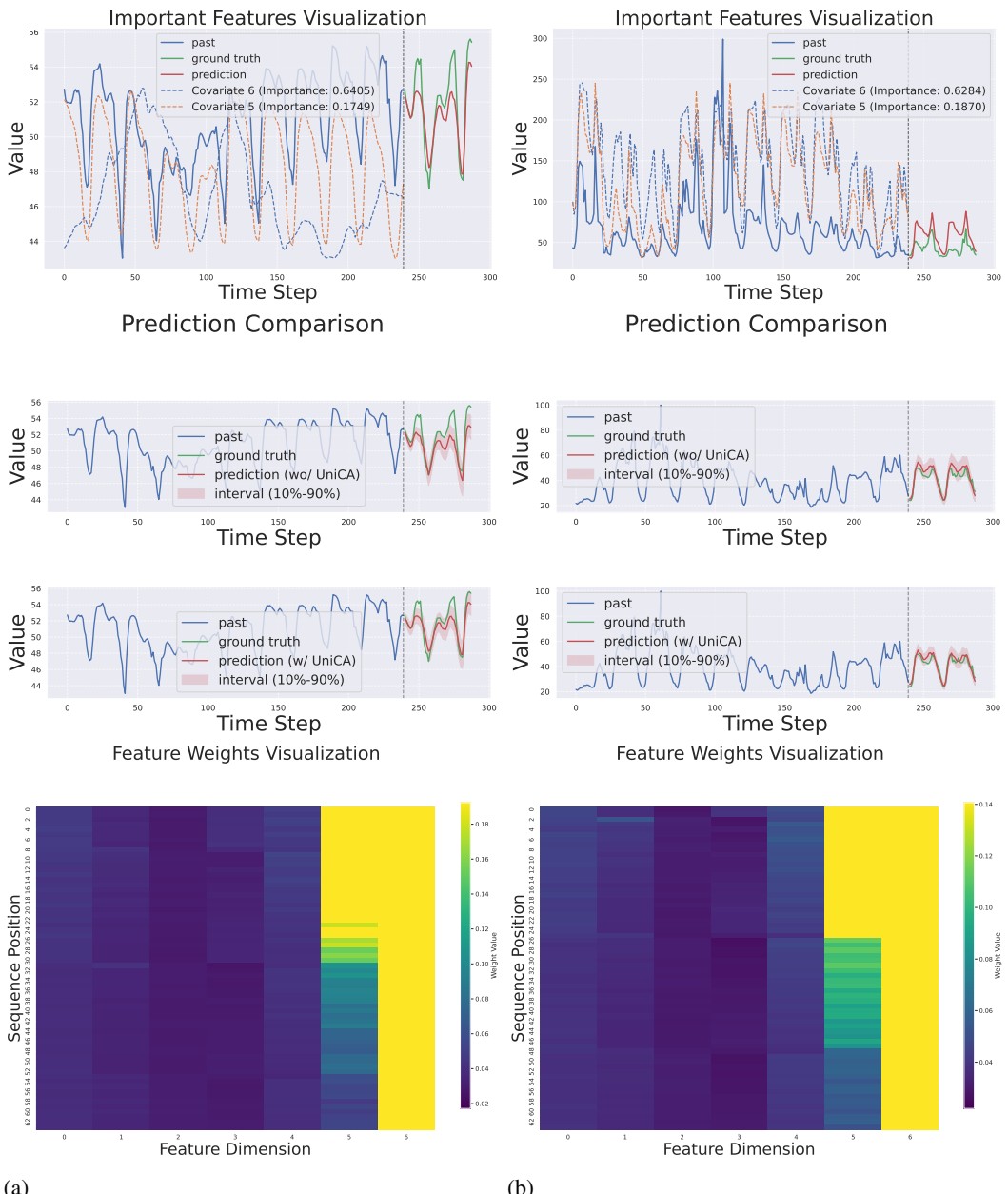

Figure 10: Visualization of prediction comparison, important covariates and attention map on EPF

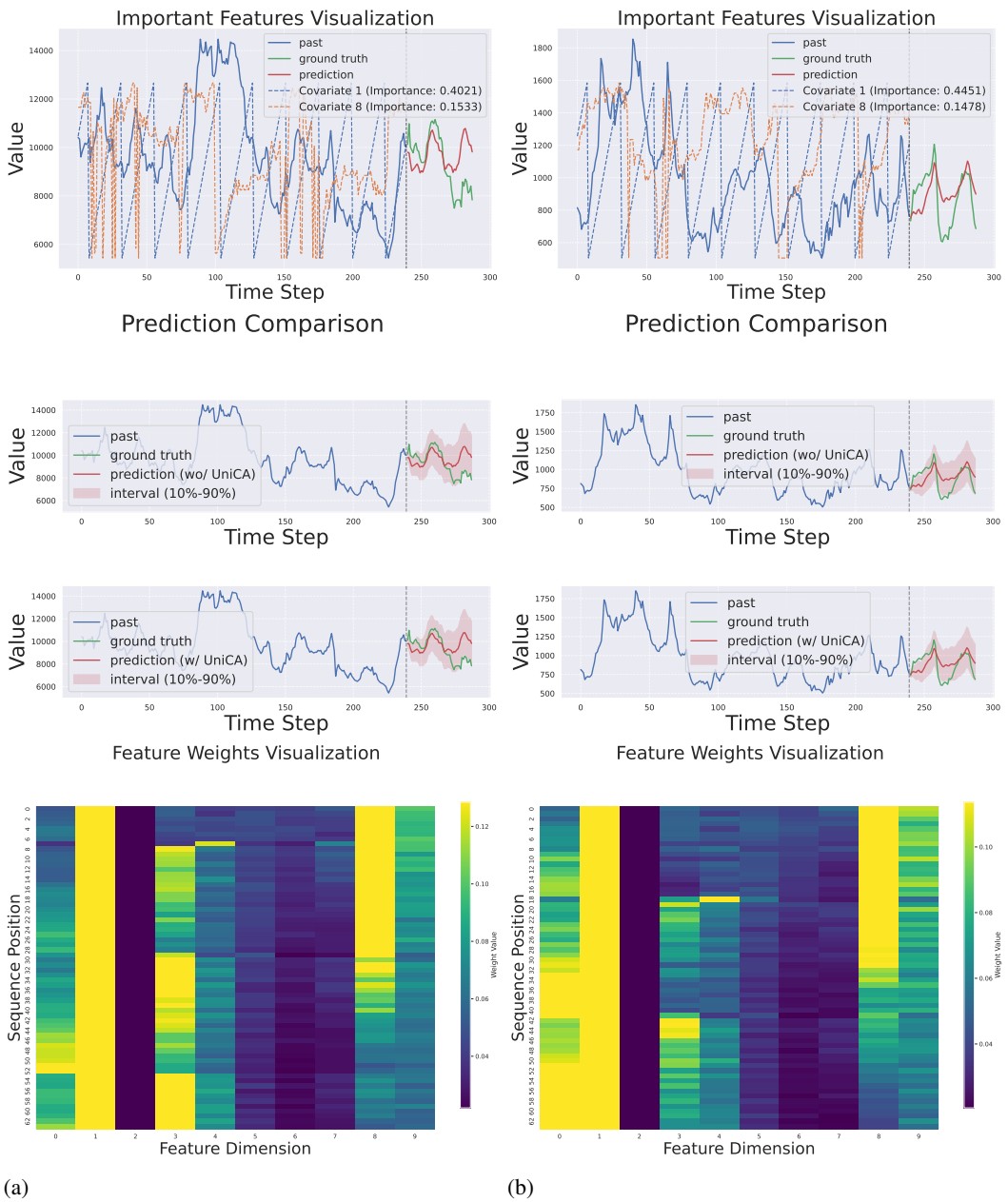

Figure 11: Visualization of prediction comparison, important covariates and attention map on Hog

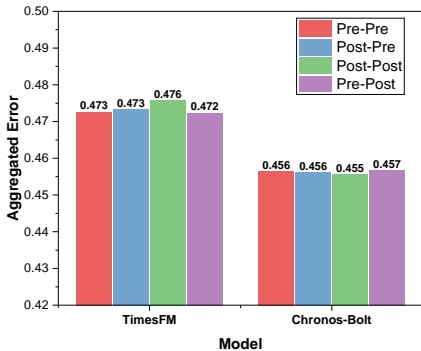

Figure 12: Ablation study on fusion position. `Pre-Pre`, `Post-Pre`, `Post-Post`, and `Pre-Post` indicate the fusion positions of past and future covariates (before or after the temporal encoder). Results are shown on TimesFM and Chronos-Bolt. Performance remains stable across all configurations, demonstrating the fusion position does not affect the performance much.

mains stable and effective regardless of specific fusion positions. This reflects the flexibility of our attention-based fusion modules and their adaptability across model architectures.

## G.2 INFLUENCE OF MODALITY ENCODER

To investigate the effect of different text modality encoders on forecasting performance, we conduct a comprehensive evaluation on the Time-MMD benchmark across six domains: *Climate*, *Energy*, *Environment*, *Health*, *Security*, and *SocialGood*. We embed the same textual covariates using four representative pretrained language models—GIST (Solatorio, 2024) [10] (a text embedding model), BERT (Devlin et al., 2019) [11], GPT-2 (Radford et al., 2019) [12], and LLAMA-2 (Touvron et al., 2023)[13]—and report forecasting results under two forecasting backbones: Chronos-Bolt and TimesFM, both implemented via the UniCA framework. The results are summarized in Table 13. Chronos consistently benefits from text embeddings, with minor variation across encoder types. In contrast, TimesFM exhibits significantly larger fluctuations in performance depending on the encoder. For instance, under the *Environment* domain, the MAE of TimesFM ranges from 0.756 (GIST) to 1.181 (BERT), whereas Chronos maintains stable performance across all encoders (MAE = 0.738 for all). Among all text encoders, GIST demonstrates the most consistent performance across both Chronos and TimesFM, achieving the lowest average forecasting errors in domains such as *Security* (MAE = 0.707) and *SocialGood* (MAE = 0.860) for TimesFM. On the other hand, large language models such as LLAMA do not necessarily outperform smaller encoders. In several cases, GPT and BERT lead to degraded performance in TimesFM, especially for domains with noisier text (e.g., *Environment* and *Security*). The optimal encoder choice appears domain-dependent. For example, in the *Health* domain, GPT yields the best MAE (0.609) under Chronos, while GIST performs best under TimesFM (MAE = 0.692). In the *Security* domain, TimesFM benefits the most from GIST (MAE = 0.707), which is substantially better than other encoders.

These findings suggest that the selection of the modality encoder impacts model performance, depending on the dataset and base model. However, we also warn that the number of observed points in Time-MMD may not be enough to draw a consistent conclusion. This experiment only shows preliminary results.

## G.3 ARCHITECTURE OF COVARIATE HOMOGENIZER

We examine how the architecture of the covariate homogenizer affects forecasting performance in our unified framework. Specifically, we compare two designs: a simple Linear layer and a two-layer

---

[10]https://huggingface.co/avsolatorio/GIST-small-Embedding-v0

[11]https://huggingface.co/google-bert/bert-base-uncased

[12]https://huggingface.co/openai-community/gpt2

[13]https://huggingface.co/meta-llama/Llama-2-7b-hf

Table 13: Forecasting error on Time-MMD subsets with different text encoder.

| | | Chronos-Bolt (UniCA) | | | | TimesFM (UniCA) | | | |
|---|---|---|---|---|---|---|---|---|---|
| | | GIST | Bert | GPT | LLAMA | GIST | Bert | GPT | LLAMA |
| Climate | Average | 0.507 | 0.507 | 0.507 | **0.507** | 0.533 | 0.541 | 0.542 | 0.533 |
| | MAE | 0.628 | 0.628 | 0.628 | **0.628** | 0.635 | 0.630 | 0.631 | 0.635 |
| | MAPE | 0.421 | 0.421 | 0.423 | **0.421** | 0.543 | 0.577 | 0.578 | 0.543 |
| | MSE | 0.411 | 0.411 | 0.411 | 0.411 | 0.402 | **0.401** | 0.402 | 0.402 |
| | CRPS | 0.568 | 0.568 | 0.567 | 0.567 | **0.552** | 0.558 | 0.556 | 0.552 |
| Energy | Average | **0.879** | 0.890 | 0.893 | 0.903 | 0.904 | 0.915 | 0.907 | 0.954 |
| | MAE | **0.921** | 0.929 | 0.928 | 0.937 | 0.957 | 0.972 | 0.934 | 0.968 |
| | MAPE | 0.831 | 0.850 | 0.866 | 0.876 | **0.795** | 0.811 | 0.925 | 0.971 |
| | MSE | **0.865** | 0.874 | 0.871 | 0.882 | 0.934 | 0.935 | 0.866 | 0.934 |
| | CRPS | **0.899** | 0.907 | 0.906 | 0.916 | 0.928 | 0.944 | 0.903 | 0.944 |
| Environment | Average | 0.625 | 0.625 | 0.625 | **0.625** | 0.642 | 1.103 | 0.661 | 1.024 |
| | MAE | 0.738 | 0.738 | 0.738 | **0.738** | 0.756 | 1.181 | 0.754 | 1.095 |
| | MAPE | 0.666 | 0.667 | 0.666 | 0.666 | **0.656** | 1.189 | 0.716 | 1.145 |
| | MSE | 0.560 | 0.561 | 0.560 | **0.560** | 0.599 | 1.114 | 0.620 | 0.982 |
| | CRPS | 0.536 | 0.536 | 0.536 | **0.536** | 0.556 | 0.930 | 0.554 | 0.875 |
| Health | Average | 0.612 | 0.609 | **0.609** | 0.617 | 0.692 | 0.693 | 0.710 | 0.709 |
| | MAE | 0.697 | **0.694** | 0.694 | 0.702 | 0.753 | 0.754 | 0.769 | 0.764 |
| | MAPE | 0.565 | 0.569 | **0.565** | 0.573 | 0.684 | 0.679 | 0.698 | 0.718 |
| | MSE | 0.484 | **0.479** | 0.481 | 0.490 | 0.600 | 0.604 | 0.623 | 0.612 |
| | CRPS | 0.699 | 0.696 | **0.695** | 0.703 | 0.733 | 0.735 | 0.750 | 0.742 |
| Security | Average | 0.705 | 0.706 | 0.702 | 0.698 | **0.593** | 0.663 | 0.879 | 0.642 |
| | MAE | 0.860 | 0.862 | 0.858 | 0.854 | **0.707** | 0.785 | 1.009 | 0.762 |
| | MAPE | 0.536 | 0.537 | 0.532 | 0.526 | **0.467** | 0.553 | 0.854 | 0.526 |
| | MSE | 0.660 | 0.660 | 0.658 | 0.656 | **0.578** | 0.624 | 0.747 | 0.613 |
| | CRPS | 0.764 | 0.765 | 0.761 | 0.757 | **0.621** | 0.689 | 0.907 | 0.668 |
| SocialGood | Average | 0.790 | 0.790 | 0.781 | 0.790 | 0.664 | 0.656 | 0.661 | **0.654** |
| | MAE | 0.860 | 0.860 | 0.834 | 0.860 | 0.699 | 0.689 | 0.695 | **0.687** |
| | MAPE | 0.666 | 0.666 | 0.673 | 0.666 | 0.547 | 0.530 | 0.541 | **0.529** |
| | MSE | 0.784 | 0.784 | 0.791 | 0.784 | 0.717 | **0.711** | 0.717 | 0.716 |
| | CRPS | 0.850 | 0.850 | 0.827 | 0.850 | 0.694 | 0.692 | 0.690 | **0.683** |

MLP, implemented under two different model backbones—Chronos-Bolt and TimesFM, both with UniCA. Results are reported on the MMSP and MMSP$^{\dagger}$ benchmarks, as shown in Table 14.

From the results, we observe that for Chronos, the homogenizer design has negligible impact on performance: both Linear and MLP yield nearly identical errors across all metrics. In contrast, TimesFM shows a consistent preference for the Linear homogenizer. For instance, in MMSP, the Average error increases from 0.147 to 0.154 when switching from Linear to MLP, with similar degradation observed in MAE, MSE, and CRPS. This trend holds across both MMSP and MMSP$^{\dagger}$.

These results suggest that more complex homogenizer structures like MLP may introduce unnecessary parameterization and lead to overfitting in already expressive models such as TimesFM, while simpler designs suffice for both backbones. Therefore, we adopt the Linear homogenizer as the default in all main experiments.

Table 14: Forecasting error on MMSP with different designs of covariate homogenizer.

| | | Chronos-Bolt (UniCA) | | TimesFM (UniCA) | |
|---|---|---|---|---|---|
| | | Linear | MLP | Linear | MLP |
| MMSP | Average | 0.121 | **0.120** | **0.147** | 0.154 |
| | MAE | 0.193 | **0.193** | **0.229** | 0.236 |
| | MAPE | 0.019 | **0.019** | **0.046** | 0.050 |
| | MSE | 0.090 | **0.090** | **0.098** | 0.106 |
| | CRPS | 0.180 | **0.180** | **0.215** | 0.222 |

## G.4 INFLUENCE OF STATIC COVARIATES

Our UniCA model incorporates the static covariate that most of the datasets do not include. To isolate and quantify the benefit of this feature, we conduct an ablation study on two datasets with static

covariates: M5 and Retail[1]. [14] The results, presented in table 15, indicate that incorporating static covariates generally improves performance. The performance gains are most pronounced on the highly diverse Retail dataset, where UniCA improves all metrics for both base models (e.g., reducing the Average metric for TimesFM from 0.672 to 0.655). While on the M5 dataset, the improvements are modest but still noticeable in several metrics (e.g., Chronos-Bolt's MAPE drops from 0.779 to 0.764, MSE from 0.581 to 0.566). These findings suggest that static variables provide useful categorical and contextual information that enhances model generalization, especially in datasets with diverse item-level characteristics like Retail.

Table 15: Ablation study on the impact of static covariates, conducted on the M5 and Retail datasets. Results show that adding static variables generally improves performance, especially on the Retail dataset, indicating their importance for capturing categorical and contextual information.

| Dataset | Metric | Chronos-Bolt | | TimesFM | |
|---|---|---|---|---|---|
| | | wo/ static | w/ static | wo/ static | w/ static |
| M5 | Average | 0.625 | **0.613** | **0.609** | 0.613 |
| | MAE | 0.706 | **0.699** | **0.699** | 0.701 |
| | MAPE | 0.779 | **0.764** | **0.731** | 0.737 |
| | MSE | 0.581 | **0.566** | **0.581** | 0.587 |
| | CRPS | 0.432 | **0.424** | **0.425** | 0.426 |
| Retail | Average | 0.680 | **0.656** | 0.672 | **0.655** |
| | MAE | 0.720 | **0.712** | 0.712 | **0.709** |
| | MAPE | 0.702 | **0.655** | 0.707 | **0.648** |
| | MSE | 0.799 | **0.769** | 0.781 | **0.776** |
| | CRPS | 0.499 | **0.489** | 0.486 | **0.485** |

## G.5 IMPACT OF COVARIATES

To see whether UniCA's performance gains simply stem from the introduction of additional trainable parameters rather than the effective integration of covariates, we designed a crucial ablation study. We created an ablated model, denoted as UniCA `wo/ Cov`, which retains the exact same model structure and the same number of trainable parameters as the full UniCA model. However, in this ablated version, we intentionally **set the gating mechanism for all covariates to zero throughout the training and inference processes**. This ensures that while the extra parameters are present and trained, the adapted covariate features are prevented from influencing the TSFM backbone. We compare the full UniCA model against this UniCA `wo/ Cov` variant using both Chronos-Bolt and TimesFM backbones across all metrics.

The results of this ablation study are presented in Figure 13. For nearly all metrics and both backbones, the full UniCA model (red bars) significantly outperforms the ablated UniCA `wo/ Cov` model (blue bars). For instance, with the Chronos-Bolt backbone, UniCA achieves 0.509 MAE and 0.383 MSE, substantially better than the `wo/ Cov` model's 0.517 MAE and 0.396 MSE. A similar pattern holds for TimesFM, particularly for the MAE and MAPE metrics. The consistent degradation in performance for the `wo/ Cov` configuration clearly demonstrates that the performance gain is not merely due to the additional parameters introduced by the adaptation layer. Instead, the gains are primarily attributable to the **effective and meaningful integration of external covariate information** enabled by the learned adaptation and gating mechanism, thereby validating the core design of UniCA.

## G.6 ROBUSTNESS TO NOISY COVARIATES

To verify UniCA's stability against irrelevant external information, we conducted a robustness experiment by introducing a purely noisy covariate to all benchmark datasets. This synthetic feature was generated as white noise, sampled from $\mathcal{N}(0, 1)$, and is entirely uninformative and future-unknowable, simulating a low-quality input that a robust model should ignore. The UniCA model in its `w/ Noise` configuration processes this noise alongside any existing covariates. We compared the performance of UniCA (`w/ Noise`) against the Zero-Shot (ZS) baseline and the original

---

[14]The M5 dataset includes static covariates 'item id", "depeartment id", "category id", "store id", and "state id". The Retail dataset includes "city", "state", "type", "cluster", "family", "class", and "perishable"

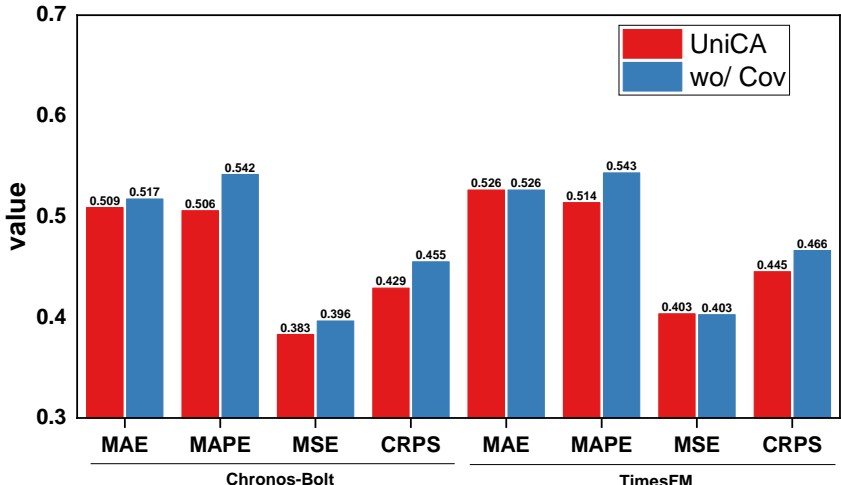

Figure 13: Comparison between the full UniCA model (red) and an ablated version (UniCA wo/ Cov, blue) where the covariate adaptation parameters are introduced but the covariate influence is intentionally gated to zero. A lower score is better.

UniCA configuration, testing both Chronos-Bolt and TimesFM backbones, with the goal of ensuring the adaptation mechanism does not catastrophically integrate non-predictive features.

Table 16: Performance comparison of UniCA and the Zero-Shot (ZS) baseline when UniCA is provided with a purely random, future-unknowable noisy covariate (w/ Noise). Results are averaged across all benchmark datasets, demonstrating UniCA's resilience to irrelevant noise inputs.

| | Chornos-Bolt | | | TimesFM | | |
|---|---|---|---|---|---|---|
| | ZS | UniCA | w/ Noise | ZS | UniCA | w/ Noise |
| **Avg** | 0.472 | 0.457 | 0.468 | 0.473 | 0.472 | 0.475 |
| **MAE** | 0.521 | 0.509 | 0.518 | 0.530 | 0.526 | 0.526 |
| **MAPE** | 0.522 | 0.506 | 0.516 | 0.523 | 0.514 | 0.523 |
| **MSE** | 0.403 | 0.383 | 0.397 | 0.402 | 0.403 | 0.403 |
| **CRPS** | 0.441 | 0.429 | 0.441 | 0.437 | 0.445 | 0.446 |

The results, summarized in Table 16, confirm UniCA's strong resilience to noisy covariates. With the Chronos-Bolt backbone, UniCA (w/ Noise) shows only a marginal performance degradation (Avg Metric $0.457 \rightarrow 0.468$) and still outperforms the ZS baseline ($0.468$ vs $0.472$). This suggests the model's feature integration module effectively suppresses the disturbance from the pure noise. The stability is even clearer with the TimesFM backbone, where the performance is nearly identical ($0.472 \rightarrow 0.475$). This high degree of stability validates the design of the UniCA architecture: its adaptation layer is highly effective at identifying and mitigating the impact of uninformative or noisy input features, ensuring that the integration of external data does not compromise the model's core forecasting accuracy.

## G.7 Fusion Architecture

To justify the choice of Gated Residual Network (GRN) and Gated Linear Unit (GLU) for covariate adaptation and fusion, we conducted an ablation study against a simpler alternative. The reviewer questioned the necessity of these specific non-linear structures. We thus implemented a baseline fusion mechanism, termed **Weight Fusion**, where the complex GRN/GLU adaptation module is replaced by a simple, weight-learnable linear combination (a single fully connected layer followed by a linear output) before fusion with the TSFM backbone's representations. This alternative maintains a similar parameter count but removes the complex gating and non-linear residual structure. We compare the performance of the full UniCA model against this simpler Weight Fusion approach,

Table 17: Performance comparison between the full UniCA model (using GRN/GLU for adaptation) and an alternative "Weight Fusion" mechanism where the complex adaptation module is replaced by a simple trainable weighted summation. A lower score is better, validating the superiority of the non-linear GRN/GLU structure.

|  | Chornos-Bolt | | | TimesFM | | |
|---|---|---|---|---|---|---|
|  | ZS | UniCA | Weight Fusion | ZS | UniCA | Weight Fusion |
| **Avg** | 0.472 | 0.457 | 0.471 | 0.473 | 0.472 | 0.478 |
| **MAE** | 0.521 | 0.509 | 0.519 | 0.530 | 0.526 | 0.532 |
| **MAPE** | 0.522 | 0.506 | 0.529 | 0.523 | 0.514 | 0.523 |
| **MSE** | 0.403 | 0.383 | 0.396 | 0.402 | 0.403 | 0.408 |
| **CRPS** | 0.441 | 0.429 | 0.439 | 0.437 | 0.445 | 0.450 |

as well as the Zero-Shot (ZS) baseline, across all metrics and both Chronos-Bolt and TimesFM backbones.

The results of the fusion mechanism ablation are presented in the table. Across all metrics and both backbones, the full UniCA model consistently achieves the best performance. Specifically, the simple Weight Fusion mechanism performs noticeably worse than the full UniCA model and, in many cases (e.g., Chronos-Bolt Avg: 0.471, TimesFM Avg: 0.478), its performance degrades close to or even below the Zero-Shot (ZS) baseline (TimesFM Avg: 0.473). For example, with Chronos-Bolt, UniCA's MAE is 0.509, while Weight Fusion's is 0.519. This significant performance gap strongly validates our architectural choice. The GRN and GLU structures are critical because their **gated, non-linear, and residual design** allows the model to selectively adapt and dynamically weigh the contribution of each covariate feature, which is essential for effective fusion with the general-purpose TSFM representations. Simple linear fusion, in contrast, fails to capture the necessary complexities for optimal TSFM adaptation.

## H    CORRELATION ANALYSIS OF HOMOGENIZED COVARIATE EMBEDDINGS

To address the reviewer's concern regarding the interpretability and meaningfulness of the homogenized covariate embeddings, we conducted an analysis of the feature space. The core idea of homogenization is to transform the diverse covariate inputs into a unified, rich representation space suitable for fusion with the TSFM's temporal embeddings. To demonstrate that these features capture meaningful temporal structure, we calculated the **Pearson Correlation Coefficient** between the first four dimensions of the final homogenized covariate embedding (just before it enters the TSFM backbone) and the target time series for a diverse set of benchmark datasets.

Table 18: Pearson Correlation of Homogenized Embedding Dimensions with the Target Series. The table presents the Pearson correlation coefficient between the first four dimensions of the homogenized embeddings (before fusion) and the corresponding target time series across various datasets. The results show that different embedding dimensions capture distinct and meaningful temporal structure related to the target.

|  | MMSP | Climate | Energy | Environment | Health | Security | SocialGood | Traffic |
|---|---|---|---|---|---|---|---|---|
| Feature 1 | -0.149 | -0.058 | 0.068 | -0.055 | 0.033 | 0.052 | 0.434 | 0.271 |
| Feature 2 | 0.415 | 0.058 | -0.419 | -0.001 | -0.114 | 0.063 | -0.114 | 0.127 |
| Feature 3 | 0.310 | -0.117 | 0.113 | 0.042 | 0.087 | 0.105 | 0.459 | 0.289 |
| Feature 4 | 0.222 | 0.024 | 0.224 | 0.037 | -0.023 | -0.084 | -0.371 | -0.104 |

The correlation results are presented in the table. While the absolute value of the correlation varies significantly by dataset and feature dimension, the patterns strongly support our claim that the homogenized embeddings capture meaningful, yet diverse, temporal structures. For instance, in the `SocialGood` dataset, Feature 3 exhibits a strong positive correlation ($0.4593$), whereas Feature 4 shows a moderate negative correlation ($-0.3713$), indicating that different dimensions of the embed-

ding are learning to capture distinct aspects of the underlying temporal dynamics of the target series. The general non-zero correlations across datasets (e.g., strong correlation in `MMSP` and `Traffic`) confirms that the adaptation process successfully transforms the raw, disparate covariate information into a fixed-length embedding that is temporally structured and relevant to the forecasting task, which is a necessary condition for effective fusion.

## I  DISCUSSION OF LIMITATION

UniCA assumes temporal alignment between covariates and the target series, which we approximate using imputation and missing-value indicators. However, more effective alignment strategies may exist. Additionally, noisy or conflicting covariates can degrade performance. Future work may incorporate uncertainty-aware fusion, handle non-aligned or partially observed covariates, and embed task-specific inductive biases to enhance the robustness and generalizability of TSFM adaptation.

## J  THE USE OF LLMS

We utilized a Large Language Model (LLM) to assist in the writing process of this paper. The primary use of the LLM was for improving the language, style, and readability of the text. This included refining sentence structure, correcting grammatical errors, and ensuring consistency in terminology. All intellectual contributions, including the research ideas, methodology, and conclusions, are solely the work of the human authors. The authors have reviewed and take full responsibility for the entire content of this paper, ensuring its originality and scientific accuracy.

