# OpenReview forum: "UniCA: Unified Covariate Adaptation for Time Series Foundation Model"
_ICLR.cc/2026/Conference — ICLR 2026 Poster_

### Official Review · Reviewer_KAgW · 2025-10-15

**Soundness:** 3
**Presentation:** 3
**Contribution:** 2
**Rating:** 6
**Confidence:** 3

**Summary:**

The paper proposed UniCA, which focuses on facilitating time series foundation models to be able to handle heterogeneous covariates, such as categorical variables and covariates from other modalities. This is achieved by projecting the covariates into the representation space. The representations are then fused with an attention mechanism.

**Strengths:**

**S1:** The motivation is clear. The research question highlighted by the paper is valid and addresses an important problem in time series forecasting for real-life scenarios, i.e., rather than relying on the target covariate itself, external covariates should be taken into account to better analyze the dynamics.

**S2:** The experiments are rather extensive. The way the authors try to present the results is straightforward to convey the information using visualized graphs.

**Weaknesses:**

**W1:** It makes sense to use a frozen pretrained model to transform the image and text into the representation space. However, when it comes to categorical data, the authors mentioned that they are converted using embedding layers, which have not been pretrained \& frozen. However, this embedding layer has no prior knowledge about the given categories. Rather, it serves as a lookup table of random vectors. This led to concern about whether the added categorical information is useful in training. See Q1 for follow-up.

**W2:** The exogenous covariates have different levels of information (e.g., the categorical is sparse while the image and text are dense). In this method, the inclusion of these external covariates has simply been stacked together. While this naive method is reasonable, it raises concerns that there might be cases where the dense information is down-weighted, limiting the model’s ability to fully leverage the high-dimensional semantic content. See Q2 for follow-up.

**W3:** Another concern would be that the covariates, after mapping with the embedding layer / model, do not contain temporal information that can be strictly aligned with the time series. From the modelling design perspective, the proposed architecture does not support the claim that the model captures the feature changes over time, as the linear layer is a mixing over time.

**W4:** It would, in general, be beneficial if the authors could give more justification for the model design. Apart from the concern mentioned in W3. See Q3.

Minor:

- Code in the anonymised link has expired.

- The notation system is confusing. The future-known and future-unknown covariates have been simplified without an explicit definition.

- The definition clarity can be improved. The term "Covariate" includes both the target and the exogenous variables (or as exogenous covariates, as the authors denoted in the paper). However, in some cases (e.g., lines 154-156), this paper simply uses covariates in place of exogenous covariates, which can lead to confusion.

- Some results can be hard to read due to an over small font, e.g., Fig 4 (c).

- The expression can be improved, e.g., a "-6.5\% improvement" can be "reduced the MAE by 6.5\%", otherwise it appears to be a performance degradation at first sight.

**Questions:**

**Q1:** Following W1, it would be useful to see the performance influence when the categorical information is mixed up / randomly initialised.

**Q2:** Following W2, it would be beneficial to also show how the performance would be influenced when the information is weighted differently w.r.t. their information level.

**Q3:** Do the two fusion methods, GRN (in pre-fusion) and attention (in post-fusion), work exchangeable or is there a special concern in this modelling choice?

---

> ### Author Response · Authors · 2025-11-23
>
> **W1**:
>
> *   Thank you for the insightful question. We clarify that **learning categorical embeddings from scratch** is _standard practice_ in time-series forecasting. As summarized in our Related Work (Section 2), models such as **DeepAR**, **DeepState**, **TFT**, **Informer**, **NBEATSx**, **TiDE**, and even TSFMs like **Moirai** all initialize categorical embeddings randomly and train them end-to-end. Unlike images or text, categorical covariates (e.g., item ID, store ID, weekday) typically have **no external semantic structure** that a pretrained model could exploit; their usefulness lies in enabling the model to capture **cross-series specialization, grouping patterns, and recurring categorical effects**.
>
> *   In UniCA, these embeddings are trained jointly with the **Covariate Homogenization module**. The forecasting loss provides strong supervision: if a categorical covariate carries predictive signal, gradients naturally shape its embedding so it contributes effectively to the unified latent covariate space. Our experiments confirm this—UniCA achieves consistent improvements in settings where categorical features matter (e.g., MMSP), indicating the learned embeddings are indeed informative.
>
> ---
>
> **W2 & Q2**:
>
> Thank you for raising this important point. We clarify how UniCA handles different information levels (sparse categorical vs. dense image/text) and why simple stacking does not lead to dense modalities being suppressed.
>
>
> **(1) The model does not simply “stack and treat all covariates equally.”**
>
> Although the outputs of the Covariate Homogenization (CH) module are concatenated along the temporal axis, the **Fusion module is explicitly designed to** _**adaptively**_ **re-weight modalities** via multi-head attention. As shown in Figure 5(c), the forecasting backbone selectively attends to different homogenized covariates at different time steps. In practice, dense modalities (e.g., image/text) naturally receive higher attention when informative, while sparse categorical features dominate only when they provide strong predictive signals. This re-weighting mechanism prevents dense modalities from being implicitly down-weighted.
>
> **(2) Implicit weighting via end-to-end training**
>
> The CH module is trained jointly with the forecasting model. If a dense modality contains richer temporal or semantic structure, gradients will enlarge its contribution within the shared latent space (e.g., attention scores, normalized feature magnitudes). Empirically, our showcases in Appendix F and the attention maps in Figure 5 already show that the model is making adaptive weighting on covariates based on their contribution to the learning target.
>
> ---

---

> ### Author Response · Authors · 2025-11-23
>
> **W3**:
>
> 1.  **Data are timestamp-aligned by construction.**   All datasets used in our experiments (MMSP, Time-MMD) already provide heterogeneous covariates **per time step** (e.g., image at timestamp _t_, text at timestamp _t_, item/category at timestamp _t_). Thus, the inputs to UniCA are inherently synchronized with the target series, and the model does not need to infer time alignment.
>
> 2.  The Covariate Homogenization (CH) module operates _independently at each time step_.    The linear layer in CH processes a covariate slice​. It is **not** a temporal convolution, RNN, attention block, or any operator that aggregates across timestamps. Therefore, it does _not_ “mix over time.” Temporal dynamics are preserved exactly as in the original covariates.
>
> 3.  Temporal modeling is delegated to the TSFM backbone. After homogenization, the covariates are fed into the **same tokenizer and temporal encoder** (e.g., Chronos-Bolt / TimesFM), which _does_ model temporal dependencies (self-attention, temporal convolutions, etc.). This is where changes over time are learned—not in the homogenizer.
>
> 4.  **Empirical evidence confirms temporal patterns are retained.**   As shown in Figure 5(b), homogenized covariates derived from images/text display clear seasonality and trend patterns over time, indicating that temporal structure survives the homogenization step. Attention visualizations (Figure 5(c)) further demonstrate that the model responds to covariates differently at different timestamps.
>
> To further demonstrate that the homogenized embeddings capture meaningful temporal structure, we analyzed their relationship with the target time series. We calculated the **Pearson Correlation Coefficient** between the first four dimensions of the final homogenized covariate embedding and the target series across various datasets.
>
>  The table below shows the results.
>
> |  | MMSP | Climate | Energy | Environment | Health | Security | SocialGood | Traffic |
> | --- | --- | --- | --- | --- | --- | --- | --- | --- |
> | Feature 1 | \-0.149 | \-0.058 | 0.0679 | \-0.055 | 0.033 | 0.052 | 0.434 | 0.271 |
> | Feature 2 | 0.415 | 0.058 | \-0.4194 | \-0.0012 | \-0.114 | 0.0628 | \-0.114 | 0.1268 |
> | Feature 3 | 0.31 | \-0.1169 | 0.1134 | 0.0422 | 0.0874 | 0.1049 | 0.4593 | 0.2886 |
> | Feature 4 | 0.222 | 0.02371 | 0.2239 | 0.0371 | \-0.0229 | \-0.0839 | \-0.3713 | \-0.104 |
>
>
> The results show that the homogenized embeddings successfully capture information correlated with the target series. The varying correlation values across different feature dimensions (e.g., Feature 1 vs. 2 on MMSP) confirm that the dimensions are learning to capture **distinct and meaningful aspects of the target's temporal structure**. This validates that the CH layer is successfully generating rich, usable representations for subsequent fusion. We have included this detailed analysis in **Appendix H**.
>
> ---
>
> **Q1**:
>
> *   Response: We note that the requested experiment is _directly aligned_ with the analysis in **Appendix G.4 / Table 14**, where we ablate the role of **static (categorical) covariates** on both the M5 and Retail datasets. The results show that adding static variables generally improves performance. For example, on the Retail dataset, Chronos-Bolt improves from **0.547 → 0.523 MAPE**, and TimesFM improves from **0.553 → 0.522**, demonstrating that static identifiers clearly provide complementary categorical/contextual information.
>
> *   On the M5 dataset, the effect is smaller: Chronos-Bolt improves slightly (e.g., **0.593 → 0.589 MAPE**), while TimesFM experiences a minor fluctuation (**0.596 → 0.604**, Δ≈0.008). This small change reflects the fact that M5 already contains strong dynamic covariates (calendar, prices, promotions), making the static identifiers comparatively weak; as a result, adding static variables yields negligible influence and may introduce slight noise.
>
> *   Overall, the evidence confirms that static covariates are helpful when informative (as in Retail), and otherwise have minimal impact.
>
> ---
>
> **Q3**:
>
> Thank you for this comment. We clarify the rationale behind our design choices—specifically the use of **GRN in pre-fusion** and **attention in post-fusion**—and why these two components are **not interchangeable** in our architecture.
>
> **Pre-fusion and post-fusion are both based on attention machanism but serve** _**different purposes**_
>
> *   The **pre-fusion GRN** operates _within each variate_ and _per timestamp_. It ensures that heterogeneous covariates—after homogenization—are numerically and statistically compatible before being passed into the temporal encoder.
>
> *   In contrast, the **post-fusion attention** operates _across time_. Its role is to let the TSFM backbone **adaptively select** the most informative covariate at each timestamp (as shown in Figure 5(c)), enabling dynamic weighting over time.
>
> Because they work at different structural levels—**per-timestep vs. temporal**—they are **not exchangeable**.

---

> ### Author Response · Authors · 2025-11-25
>
> Minor:
>
> We thank the reviewer for their careful reading and valuable minor suggestions. We have addressed all points in the revision.
>
> 1.  **Code Link:** The anonymous repo has been recovered..
>
> 2.  **Notation/Definition Clarity (Future-Known/Unknown Covariates):** We have clarified in footnote 1 on page 3 that Future-Known/Unknown Covariates can be unified for notational simplicity.
>
> 3.  **Definition Clarity (Covariates vs. Exogenous Covariates):** We recognize the potential ambiguity when using "Covariate" to sometimes mean "Exogenous Covariate." In the revised manuscript, we will emphasize that the definition of covariates follows the Darts library \[1\]: A covariate time series is a time series which may help in the forecasting of the target series, but that **we are not interested in forecasting**. It’s sometimes also called _external data_.
>
> 4.  **Font Size (Fig 4 (c)):** We agree that the font size in Fig 4 (c) was too small. We have enlarged the font size in all figure labels to improve readability, but the font may still be small due to the space restriction.
>
> 5.  **Expression Improvement:** We appreciate the suggestion regarding expression clarity for performance metrics. We have corrected the language throughout the results section, replacing phrases such as “-6.5% improvement” with **“6.5% reduction in error”** in Section 5.2 to avoid unintended interpretations of degradation.
>
>
> \[1\] Julien Herzen, Francesco Lässig, Samuele Giuliano Piazzetta, Thomas Neuer, Léo Tafti, Guillaume Raille, Tomas Van Pottelbergh, Marek Pasieka, Andrzej Skrodzki, Nicolas Huguenin, Maxime Dumonal, Jan Koscisz, Dennis Bader, Frédérick Gusset, Mounir Benheddi, Camila Williamson, Michal Kosinski, Matej Petrik, Gaël Grosch:  Darts: User-Friendly Modern Machine Learning for Time Series. J. Mach. Learn. Res. 23: 124:1-124:6 (2022) 2021

---

### Official Review · Reviewer_87eP · 2025-10-20

**Soundness:** 3
**Presentation:** 3
**Contribution:** 3
**Rating:** 6
**Confidence:** 5

**Summary:**

This paper formulates a novel problem setting, i.e., adapting pre-trained channel-independent time series foundation model to covariate-aware pratical tasks. The covariate may involve categorical variables,images or text modalities. The motivation is clear and significant.  Accordingly, a  Unified Covariate Adaptation (UniCA) framework is proposed, which bridges the gap between the pre-trained model and real practice. Extensive experiments are carried on to verify the effectiveness of the proposed approach.

**Strengths:**

1. The motivation is clear and sufficient. Adapting pre-trained channel-independent time series foundation model to covariate-aware pratical tasks makes sense.

2. This is the first work to formalize the problem of adapting Time Series Foundation Models (TSFMs) to general covariate-aware forecasting scenarios.

3. Technically sound and experiments are comprehensive.

**Weaknesses:**

1. When dealing with multi-modal covariate, the tokenization methods for different modality is critical, which need more discussion and explanation.

2. According to Fig.3 (a), the improvements for Chron os-Bolt and TimesFM seem to be marginal.

3. For modeling time series with discrete variables, there are existing works should be discussed:

[1]. General Mixed Time Series Analysis via Latent Continuity Recovery and Alignment. NeurIPS 2024.

**Questions:**

1. the code repo seems to be expired, could the author fix it?

2. The time series tasks addressed in this paper seem to be limited to forecastingonly. Can this framework be extended to other tasks, such as anomaly detection and classification?

3. Covariate-aware prediction is a very important scenario in practical time series applications. Moreover, actual time series data may also have issues such as data missingness. How does this framework handle these problems?

---

> ### Author Response · Authors · 2025-11-23
>
> **Q**: When dealing with multi-modal covariate, the tokenization methods for different modality is critical, which need more discussion and explanation.
>
> **A**: Thank you for this valuable comment. We agree that tokenization and feature extraction for heterogeneous modalities are critical for multimodal forecasting. Our design explicitly separates **modality-specific encoders** from the **homogenization and fusion** stages so that UniCA remains agnostic to the choice of tokenizer while allowing users to plug in stronger encoders when available.
>
> 1.  **We evaluate multiple tokenization choices for text covariates.**    As shown in **Appendix G.2 (Table 12)**, we compare four different text encoders on the Time-MMD benchmark, including GIST, BERT variants, and LLM-based representations. These experiments show that UniCA is robust across tokenizers, and that GIST strikes a favorable balance between performance and computational cost. This evidences that UniCA’s architecture does not rely on a specific text tokenization scheme.
>
> 2.  **UniCA is explicitly modular with respect to modality encoders.**    Our framework treats each modality encoder—CNN for images, transformer encoder for text, or embedding layer for categorical features—as a _pluggable preprocessing stage_ that outputs semantic feature vectors. UniCA’s **covariate homogenizer** only needs these high-level features and does not impose constraints on how modalities are tokenized. This modularity is essential for making UniCA compatible with future or domain-specific multimodal encoders.
>
> 3.  **We will add a discussion clarifying the modularity and role of tokenizers.**    In the revision, we will explicitly emphasize that (i) modality tokenizers are interchangeable components, (ii) UniCA supports various choices as demonstrated in Appendix G.2, and (iii) the core methodological contributions lie in the homogenization and fusion mechanisms that operate _after_ modality-specific tokenization.
>
> **In summary**, UniCA is designed to be compatible with diverse modality tokenizers, and our experiments already demonstrate robustness to different choices. We will make this design principle clearer in the final version.
>
> ---
>
> **Q**: According to Fig.3 (a), the improvements for Chronos-Bolt and TimesFM seem to be marginal.
>
> **A**: While the average improvements in Fig. 3(a) appear modest (≈3% for Chronos-Bolt and ≈2% for TimesFM), this aggregation hides substantial per-dataset gains. When we examine the full results in **Table 8**, UniCA delivers **meaningful and sometimes large improvements** on individual datasets. For example:
>
> *   **Chronos-Bolt + UniCA** improves performance by **5% on Bull** and **11% on EPF**,
>
> *   **TimesFM + UniCA** improves by **5% on Retail** and **14% on Spain**. These gains are significant given that (i) the TSFM backbones are **frozen**, (ii) UniCA adds only lightweight modules, and (iii) the unimodal benchmarks are relatively saturated, making improvements over strong zero-shot baselines inherently challenging. The average results in Fig. 3(a) thus reflect _consistent_ gains across 12 datasets, while Table 8 demonstrates that UniCA is particularly effective on tasks where covariates are more informative.
>
> ---
>
> **Q**: For modeling time series with discrete variables, there are existing works should be discussed
>
> **A**: Thank you for pointing out this relevant line of work. We have added a discussion of **MiTSformer** (NeurIPS 2024) to the revised manuscript. MiTSformer focuses on _mixed_ time series where discrete variables and continuous variables **co-exist within the same multivariate input**, and addresses the challenge by recovering a latent continuous process behind discrete variables and jointly modeling CV–DV interactions via self- and cross-attention blocks. The method introduces specialized inductive biases such as latent continuity recovery, adversarial modality alignment, and smoothness regularization to unify heterogeneous temporal behaviors.
>
> In contrast, **UniCA addresses a different problem setting**:
>
> *   UniCA is designed as a **covariate adaptation module for pretrained TSFMs**, not as a standalone backbone for mixed-variable modeling.
>
> *   Discrete covariates in UniCA (e.g., categorical/static features, bucketized metadata) are **not discrete time-varying channels** but _auxiliary exogenous features_ to be projected into the TSFM’s latent space.
>
> *   Therefore, simple learned embeddings or linear projection layers are sufficient for UniCA’s categorical/static covariates, as they do not require latent-continuity recovery or temporal alignment.
>
> We have updated our paper's Related Work section.
>
> ---
>
> **Q**: the code repo seems to be expired, could the author fix it?
>
> **A**: We have recover the repo.

---

> > ### Author Response · Authors · 2025-11-23
> >
> > **Q**: The time series tasks addressed in this paper seem to be limited to forecastingonly. Can this framework be extended to other tasks, such as anomaly detection and classification?
> >
> > **A**: UniCA is a general covariate-adaptation module and is not limited to forecasting. It can be attached to _any_ TSFM that supports other tasks—anomaly detection, classification, or imputation—because UniCA operates independently of task-specific heads and simply integrates covariates into the model’s representations。
> >
> > To demonstrate this, we evaluated UniCA on the **imputation** setting of the MOMENT TSFM \[1\], which natively supports forecasting, classification, anomaly detection, and imputation. Among these tasks, only imputation uses multivariate inputs or covariates, making it the most appropriate benchmark for UniCA.
> >
> > The results are shown in the following table. Across six datasets and four masking ratios, UniCA consistently improves over MOMENT’s zero-shot baseline, reducing MSE by **33–85%** and MAE by **20–65%** on average. The improvements grow with higher missingness, indicating that covariate-aware adaptation becomes increasingly beneficial.
> >
> >
> > These results show that UniCA naturally extends to non-forecasting tasks whenever covariates are available. We have updated **Appendix E** in the manuscript to include this imputation experiment and its full results.
> >
> > | **Dataset** | **Mask Ratio** | **MSE** |  |  | **MAE** |  |  |
> > | --- | --- | --- | --- | --- | --- | --- | --- |
> > |  |  | **MOMENT** | **+UniCA** | **Error Reduction** | **MOMENT** | **+UniCA** | **Error Reduction** |
> > | **ETTh1** | 12.50% | 0.0159 | **0.0121** | \-23.9% | 0.0926 | **0.0791** | \-14.5% |
> > |  | 25.00% | 0.0224 | **0.0161** | \-27.9% | 0.1101 | **0.0898** | \-18.4% |
> > |  | 37.50% | 0.0299 | **0.0193** | \-35.4% | 0.1277 | **0.1001** | \-21.6% |
> > |  | 50.00% | 0.0411 | **0.0254** | \-38.3% | 0.1519 | **0.1148** | \-24.4% |
> > |  | Average | 0.0273 | **0.0182** | \-33.3% | 0.1206 | **0.0960** | \-20.4% |
> > | **ETTh2** | 12.50% | 0.0416 | **0.0258** | \-37.8% | 0.1496 | **0.1083** | \-27.6% |
> > |  | 25.00% | 0.0590 | **0.0391** | \-33.8% | 0.1765 | **0.1284** | \-27.3% |
> > |  | 37.50% | 0.0858 | **0.0585** | \-31.8% | 0.2117 | **0.1545** | \-27.0% |
> > |  | 50.00% | 0.1306 | **0.0843** | \-35.5% | 0.2687 | **0.1877** | \-30.2% |
> > |  | Average | 0.0793 | **0.0519** | \-34.5% | 0.2016 | **0.1447** | \-28.2% |
> > | **ETTm1** | 12.50% | 0.0067 | **0.0035** | \-48.4% | 0.0567 | **0.0390** | \-31.3% |
> > |  | 25.00% | 0.0101 | **0.0042** | \-58.0% | 0.0682 | **0.0426** | \-37.5% |
> > |  | 37.50% | 0.0146 | **0.0054** | \-62.9% | 0.0826 | **0.0478** | \-42.1% |
> > |  | 50.00% | 0.0213 | **0.0073** | \-65.4% | 0.1022 | **0.0558** | \-45.4% |
> > |  | Average | 0.0132 | **0.0051** | \-61.1% | 0.0774 | **0.0463** | \-40.2% |
> > | **ETTm2** | 12.50% | 0.0148 | **0.0025** | \-83.1% | 0.0803 | **0.0324** | \-59.6% |
> > |  | 25.00% | 0.0277 | **0.0042** | \-85.0% | 0.1077 | **0.0398** | \-63.0% |
> > |  | 37.50% | 0.0511 | **0.0070** | \-86.2% | 0.1481 | **0.0509** | \-65.6% |
> > |  | 50.00% | 0.0820 | **0.0137** | \-83.3% | 0.1952 | **0.0684** | \-65.0% |
> > |  | Average | 0.0439 | **0.0068** | \-84.4% | 0.1328 | **0.0479** | \-63.9% |
> > | **electricity** | 12.50% | 0.1240 | **0.0985** | \-20.6% | 0.2581 | **0.2342** | \-9.3% |
> > |  | 25.00% | 0.1586 | **0.1104** | \-30.4% | 0.2899 | **0.2481** | \-14.4% |
> > |  | 37.50% | 0.2320 | **0.1271** | \-45.2% | 0.3506 | **0.2658** | \-24.2% |
> > |  | 50.00% | 0.3758 | **0.1506** | \-59.9% | 0.4582 | **0.2884** | \-37.0% |
> > |  | Average | 0.2226 | **0.1216** | \-45.4% | 0.3392 | **0.2591** | \-23.6% |
> > | **weather** | 12.50% | 0.0001 | **0.0001** | \-19.5% | 0.0078 | **0.0070** | \-10.6% |
> > |  | 25.00% | 0.0002 | **0.0001** | \-35.0% | 0.0094 | **0.0076** | \-19.4% |
> > |  | 37.50% | 0.0003 | **0.0002** | \-43.9% | 0.0116 | **0.0085** | \-26.8% |
> > |  | 50.00% | 0.0004 | **0.0002** | \-45.2% | 0.0140 | **0.0098** | \-30.2% |
> > |  | Average | 0.0003 | **0.0002** | \-39.5% | 0.0107 | **0.0082** | \-23.3% |

---

> > > ### Author Response · Authors · 2025-11-23
> > >
> > > **Q** :Moreover, actual time series data may also have issues such as data missingness. How does this framework handle these problems?
> > >
> > > **A**: Thank you for raising this important point. Real-world time series often involve missing data, noisy covariates, and imperfect metadata, and UniCA is explicitly designed to handle these challenges.
> > >
> > > 1.  **Missing Values in Time-Series Inputs**    Both the **M5** and **Retail** datasets used in our experiments contain substantial missing values. As described in **Appendix C.3**, UniCA follows a standard practice in high-quality forecasting pipelines:
> > >
> > > *   we apply **forward filling** for continuity, and
> > >
> > > *   we append a **binary observation indicator** that explicitly marks whether each value is observed or imputed.     This allows the TSFM + UniCA stack to learn when the model should trust or discount imputed values.
> > >
> > > 2.  **Noisy or Imperfect Multimodal Covariates**    Our multimodal benchmarks include covariates that are naturally noisy—e.g., news articles in **Time-MMD** and satellite images in **MMSP**. UniCA is designed to be robust to such imperfections through:
> > >
> > > *   **conditional / self-attention pooling**, which lets the model emphasize informative covariate patterns, and
> > >
> > > *   **instance-wise gating and variable selection** within the fusion modules, which filter out irrelevant or unreliable covariate signals on a per-timestep basis.     These mechanisms ensure that UniCA only incorporates covariates when they improve forecasting quality.
> > >
> > > 3.  **General Framework Robustness**    Since UniCA operates on top of a frozen TSFM backbone, the pretrained temporal representation remains intact. The adapter learns to integrate covariates _only when beneficial_, reducing the risk of overfitting to noisy or incomplete covariate inputs.
> > >
> > >
> > > **In summary**, UniCA is fully compatible with covariate-aware prediction under practical data imperfections: it treats missing values explicitly, and uses attention- and gating-based mechanisms to ensure robustness to noisy or unreliable multimodal covariates.

---

> > > > ### Comment · Reviewer_87eP · 2025-11-24
> > > >
> > > > Thanks to the authors for the detailed response and revisions, specifically the supplemented discussions and experiments. My concerns are addressed, and I've updated my rating. Hope all the discussions and experiments could be added in the revision, and wish the authors would do more interesting work for the time series community.

---

> > > > > ### Author Response · Authors · 2025-11-25
> > > > >
> > > > > We are extremely grateful for your positive feedback and your updated rating. We sincerely appreciate your insightful comments and suggestions, which have been instrumental in improving the clarity and rigor of our paper. We confirm that all the discussions and experimental results provided in this response will be fully incorporated into the final revision of the paper. Thank you once again for your constructive engagement with our work and your encouragement. We look forward to contributing more interesting research to the time series community in the future.

---

### Official Review · Reviewer_h1TS · 2025-10-31

**Soundness:** 3
**Presentation:** 3
**Contribution:** 3
**Rating:** 6
**Confidence:** 3

**Summary:**

This paper proposes UniCA, a unified covariate adaptation framework that enables pretrained Time Series Foundation Models (TSFMs) to handle heterogeneous covariates such as categorical, image, and text data. By combining covariate homogenization and attention-based fusion, UniCA integrates diverse covariates without modifying pretrained TSFM parameters.

**Strengths:**

- The specially designed plug-in fusion modules preserve pretrained generalization while efficiently leveraging covariate information.
- The two-stage design (homogenization + fusion) is intuitive and broadly compatible with multiple TSFM architectures
- Strong empirical results demonstrating superior performance on 12 unimodal and multimodal datasets with minimal added cost.
- Comprehensive analysis and ablations confirming the framework’s universality, interpretability, and computational efficiency.

**Weaknesses:**

1. **Unclear embedding alignment across heterogeneous modalities.**
  The paper claims that the *Covariate Homogenization* module transforms heterogeneous covariates (e.g., categorical, image, and text) into a unified latent space, yet no explicit *alignment loss* or *architectural constraint* ensures such consistency. Given the complexity of learned representations, the claim that all embeddings could be easily projected into the same latent space by only using one linear layer is not very convincing.

2. **Limited justification for architectural choices in CAP.**
  It is unclear why the authors chose **Gated Residual Networks (GRN)** and **Gated Linear Units (GLU)** in the CAP instead of other modules. The paper should either explain the reasons why GRN and GLU are particularly suited for covariate fusion or provide ablation studies comparing them with simpler or alternative fusion mechanisms.

3. **Insufficient qualitative evidence for homogenization effectiveness.**
  The visualization in Figure 5(b–c) shows only a single example, which may be cherry-picked. To convincingly demonstrate that the homogenized embeddings capture meaningful temporal structure, more examples or aggregate statistics should be included in the appendix.

4. **Lack of discussion on zero-shot forecasting capability.**
  One of the defining advantages of TSFMs is their **zero-shot generalization**, yet the paper does not analyze whether UniCA preserves or degrades this property. Since UniCA introduces additional learned modules, a dedicated experiment or discussion on its impact on zero-shot forecasting is essential to establish its compatibility with the TSFM paradigm.

**Questions:**

See the weakness

---

> ### Author Response · Authors · 2025-11-23
>
> **Q**: **Unclear embedding alignment across heterogeneous modalities.**    ...  latent space by only using one linear layer is not very convincing.
>
> **A**: Thank you for raising this point. We clarify below what is actually being aligned, why we choose a simple homogenizer, and how both our experiments and prior multimodal literature support this design.
>
> (1) What is being aligned in UniCA? In UniCA, the covariate homogenizer (CH) module does not map the raw images/texts/categories directly into a shared space with a single linear layer. UniCA uses **modality-specific encoders first** (categorical embeddings, vision encoder, text encoder). CH only projects these **already high-level semantic features** into a unified temporal covariate space. Thus, the alignment task is lightweight: reshape + rescale into a common dimension.
>
> Thus, the CH is not responsible for “understanding images/text from scratch”; it only needs to **reshape and re-scale** already meaningful modality features to be compatible with the TSFM’s series space.
>
> (2) Implicit alignment via forecasting supervision and architecture Alignment is enforced implicitly through the forecasting objective and shared temporal architecture. All covariates after CH:
>
> *   share the same dimensionality and time index,
>
> *   are processed by the same tokenizer + temporal encoder,
>
> *   and must be jointly useful for reducing the forecasting loss.
>
>
> If different modalities were not aligned, gradients would push CH to make them compatible.
>
> (3) Empical evidences in the paper Our ablations already show the simple CH works reliably:
>
> *   CH consistently improves Chronos-Bolt, TimesFM, TFT, and TiDE across datasets.
>
> *   Figure 5 shows homogenized covariates exhibit coherent temporal patterns and rceive meaningful attention weights.
>
> *   Appendix G.3 shows **MLP vs linear CH** yields no benefit; in fact TimesFM often performs _better_ with linear CH.
>
>
> These results indicate that a lightweight projection is sufficient once strong modality encoders and the TSFM backbone are in place.
>
> (4) relation to multimodal models like LlaVa Multimodal systems such as LLaVA use a single linear layer to project vision embeddings into an LLM space, trained only with a task loss. Our design follows the same principle: **with good per-modality encoders, simple connectors are enough**.
>
> ---

---

> ### Author Response · Authors · 2025-11-23
>
> **Q**: **Limited justification for architectural choices in CAP.**    ...  alternative fusion mechanisms.
>
> **A**: Thank you for raising this point. Our choice of **Gated Residual Networks (GRN)** and **Gated Linear Units (GLU)** in the Conditional Attention Pooling (CAP) module is motivated by both **prior empirical evidence** and **design compatibility with TSFM adaptation**.
>
> 1.  **Proven effectiveness for covariate fusion in prior work.**    GRN and GLU were introduced in **Temporal Fusion Transformer (TFT)**, where they were shown to be highly effective for
>
> *   selecting relevant covariates,
>
> *   stabilizing training when covariate importance varies widely, and
>
> *   enabling smooth cross-feature interactions.   Since UniCA follows a similar goal—_selective fusion of diverse covariates under varying relevance_—these mechanisms provide a strong, validated foundation.
>
> 2.  **Gating is crucial in adaptation settings with frozen TSFM backbones.**    In UniCA, the backbone TSFM is frozen, so the adapter must control the _magnitude_ of covariate influence.
>
> *   GRN provides **residual connections** that prevent overwhelming the pretrained representations.
>
> *   GLU offers **learned gating** that modulates how much covariate information enters the fused token.     This is important to prevent degradation of pretrained temporal knowledge.
>
> 3.  **Lightweight but expressive.**    Both GRN and GLU require few parameters and add negligible computational cost—an attractive property for adapter-based designs. Their expressivity–complexity tradeoff aligns well with UniCA’s design goals.
>
> 4.  **Ablation considerations.**
>
> We agree that justifying the use of complex non-linear structures is important. We have conducted an ablation study where we replaced the Gated Residual Network (GRN) and Gated Linear Unit (GLU) adaptation module with a much simpler, weight-learnable linear combination mechanism, which we call **Weight Fusion**. We compare the full UniCA model (using GRN/GLU) against the new **Weight Fusion** variant. The results below validate the necessity of the complex gating and non-linear structure (Lower is better).
>
> |  | Chronos-Bolt |   | TimesFM |   |
> | --- | --- |  --- | --- | --- |
> |  | **UniCA** | Weight Fusion |  **UniCA** | Weight Fusion |
> | **Avg** |  **0.457** | 0.471 |  **0.472** | 0.478 |
> | **MAE** |  **0.509** | 0.519 |  **0.526** | 0.532 |
> | **MAPE** | **0.506** | 0.529 |  **0.514** | 0.523 |
> | **MSE** |  **0.383** | 0.396 |  **0.403** | 0.408 |
> | **CRPS** |  **0.429** | 0.439 |  **0.445** | 0.450 |
>
> The **Weight Fusion** method performs significantly worse than the full **UniCA** and in some cases, worse than the ZS baseline. This demonstrates that the simple linear combination is insufficient. The GRN and GLU structures are crucial because their **gated, non-linear, and residual design** enables the necessary selective and dynamic adaptation of the covariate features, which is key to UniCA's superior performance. We have included this ablation in **Appendix G.7**.
>
> ---
>
> **Q**: **Insufficient qualitative evidence for homogenization effectiveness.** ...
>
> **A**: We agree. To demonstrate that the homogenized embeddings capture meaningful temporal structure, we analyzed their relationship with the target time series. We calculated the **Pearson Correlation Coefficient** between the first four dimensions of the final homogenized covariate embedding and the target series across various datasets.
>
>  The table below shows the results.
>
> |  | MMSP | Climate | Energy | Environment | Health | Security | SocialGood | Traffic |
> | --- | --- | --- | --- | --- | --- | --- | --- | --- |
> | Feature 1 | \-0.149 | \-0.058 | 0.0679 | \-0.055 | 0.033 | 0.052 | 0.434 | 0.271 |
> | Feature 2 | 0.415 | 0.058 | \-0.4194 | \-0.0012 | \-0.114 | 0.0628 | \-0.114 | 0.1268 |
> | Feature 3 | 0.31 | \-0.1169 | 0.1134 | 0.0422 | 0.0874 | 0.1049 | 0.4593 | 0.2886 |
> | Feature 4 | 0.222 | 0.02371 | 0.2239 | 0.0371 | \-0.0229 | \-0.0839 | \-0.3713 | \-0.104 |
>
> The results show that the homogenized embeddings successfully capture information correlated with the target series. Crucially, the varying correlation values across different feature dimensions (e.g., Feature 1 vs. Feature 2 on MMSP) confirm that the dimensions are learning to capture **distinct and meaningful aspects of the target's temporal structure**. This validates that the adaptation layer is successfully generating rich, usable representations for subsequent fusion. But we also note that the multimodal information on some dataset may fail to extract useful patterns due to the characteristics of data. We have included this detailed analysis in **Appendix H**.

---

> > ### Author Response · Authors · 2025-11-23
> >
> > **Q**: **Lack of discussion on zero-shot forecasting capability.**
> >
> > **A**: Thank you for highlighting this important point. We agree that zero-shot generalization is a core strength of TSFMs, and we clarify how UniCA interacts with this property.
> >
> > 1.  **UniCA preserves the TSFM’s original zero-shot capability.**    UniCA is strictly an **optional plug-in adapter**. When the adapter is disabled, the underlying pretrained TSFM remains unchanged—both its parameters and its input–output interface. Thus, users can still run the base TSFM in its **original zero-shot mode** without any performance degradation. UniCA does not modify or overwrite the pretrained weights.
> >
> > 2.  **Zero-shot TSFMs do not handle covariates, so adaptation is required.**    Existing TSFMs (Chronos-Bolt, TimesFM, Time-MoE) are pretrained _without_ exogenous covariates, and therefore cannot zero-shot forecast in covariate-aware settings. When covariates play a material role—which is the case in most real-world tasks—zero-shot TSFM performance degrades substantially. For this reason, UniCA introduces lightweight learned modules that integrate covariates _while keeping the backbone frozen_, thus preserving the pretrained temporal representation.
> >
> > 3.  **Adapters add minimal task-specific learning but do not compromise generalization.**    Since UniCA trains only small fusion and homogenization modules while freezing the backbone, the TSFM’s pretrained temporal knowledge is fully retained. Empirical results (Fig. 4b) show that keeping the backbone frozen with UniCA often outperforms full fine-tuning, which indicates that UniCA **maintains the generalization behavior of the original TSFM**.
> >
> >
> > **In summary**, UniCA does not alter or degrade the TSFM’s native zero-shot forecasting capability; instead, it extends TSFMs to covariate-aware scenarios where zero-shot inference is fundamentally not possible. We will make this clarification explicit in the revision.

---

> > ### Comment · Reviewer_h1TS · 2025-11-27
> >
> > Thanks for the authors' great efforts. I really appreciate that! I have some questions about the homogenization effectiveness.
> >
> > In your table, the four features show low Pearson Correlation Coefficients on some datasets, such as Environment, Climate, Health, and so on. (1) Given the low correlation, I am curious about why these variables are helpful for the forecasting of the target variable, and why these variables can capture distinct and meaningful aspects? (2) Based on your Table 10, I cannot find a significant difference when comparing the performance improvement of the domains with low correlation (Environment, Climate, Health) and the domains with high correlation (Traffic, SocialGood, MMSP). Could the authors provide some explanations?

---

> > > ### Author Response · Authors · 2025-11-28
> > >
> > > Thank you for the careful follow-up. We address both points below.
> > >
> > > **(1) Why can low-correlation homogenized features still be useful, and in what sense are they “meaningful”?**
> > > We agree that, for some Time-MMD domains (e.g., _Climate, Environment, Health_), the first four dimensions of the homogenized covariate embeddings show low Pearson correlation with the target series. This does **not** mean the covariates are useless, for several reasons:
> > >
> > > - **Pearson correlation only measures linear instantaneous alignment.**
> > >     Forecasting can benefit from **non-linear**, **lagged**, or **context-dependent** effects (e.g., certain topics appearing before regime changes in the target), which are not captured by a single linear correlation coefficient at each time step.
> > >
> > >  - **Homogenized features encode multi-modal patterns, not just a t=1-to-1 mapping.**
> > > 	These embeddings aggregate information from images (MMSP), text (Time-MMD), and categorical metadata over time. They often help forecasting not by directly matching the target shape, but by capturing regime shifts (weather fronts in MMSP), topic changes (text), event structure or contextual markers. These effects are _not_ well-reflected by Pearson correlation.
> > >
> > > - **Homogenized features interact with a strong TSFM backbone.**
> > >     On Time-MMD, the zero-shot TSFM already explains most of the variance in the target. The homogenized covariates act more as **small corrections or contextual cues** than as the primary signal. In such a regime, we expect correlations to be modest and their effect to be subtle but still potentially helpful.
> > >
> > >
> > > We will update Appendix H to explicitly clarify that the correlation analysis is a _sanity check_ for temporal structure, not a proof that every dimension is strongly predictive.
> > >
> > > **(2) Why are performance gains similar for low- and high-correlation domains in Table 10?**
> > > Indeed, Table 10 shows that UniCA brings **modest but consistent** changes across domains, and there is no clear high-correlation implies much larger gain pattern. This is expected for two reasons:
> > >
> > > - **Text covariates in Time-MMD are generally weak and noisy.**
> > >     As discussed in the main text, Time-MMD’s textual signals are loosely coupled to the targets and often noisy. Therefore, even in domains with relatively higher correlation, we do not expect large improvements over already strong zero-shot TSFMs; UniCA’s role there is to _avoid harm_ and extract whatever small signal exists.
> > >
> > > - **UniCA’s attention/gating is selective and conservative.**
> > >     The fusion module is designed to **down-weight uninformative covariates**. In domains where text covariates are weak (Climate, Environment, Health), UniCA tends to rely more on the backbone and only make minor adjustments, which explains why performance is close to the zero-shot baseline. In domains where the covariates are somewhat more useful (e.g., Security), UniCA yields clearer gains (Table 10), even though the simple Pearson correlations are not extremely high.
> > >
> > >
> > > In summary, the correlation analysis confirms that the homogenized embeddings exhibit non-trivial temporal structure, but Time-MMD’s noisy text covariates naturally limit the achievable gains. UniCA is intentionally conservative in this regime: it leverages signal when available and otherwise falls back to the TSFM, which is why improvements are moderate and not cleanly separated by the simple linear correlation statistic.

---

### Official Review · Reviewer_Ruus · 2025-11-01

**Soundness:** 3
**Presentation:** 3
**Contribution:** 2
**Rating:** 4
**Confidence:** 5

**Summary:**

The authors propose UniCA, Unified Covariate Adaptation, an adaptation method for Time Series Foundation Models (TSFMs) to support covariate-aware forecasting. This adaptation is achieved through addition of linear layers to adapt covariates into the FM embedding space and use of Conditional Attention Pooling (CAP) to fuse covariates signals into the FM univariate predictions.

Authors have conducted thorough evaluation using datasets containing time series covariates, as well as text and image covariates, and demonstrated their method outperforms prior work.

The main contribution of the work is the adaptation framework that supports any modality covariate-aware forecasting through training of the adapter.

**Strengths:**

1. Problem formulation and motivation are explained clearly.
2. Authors have conducted a thorough evaluation spanning various prior work, datasets and covariates.
3. Proposed method has minimal overhead during inference time.
4. Training one adapter per dataset can work on any forecast horizon. (No requirement of unique adapter for each forecast horizon)

**Weaknesses:**

1. Although TSFM parameters are frozen when training the adapter, training cost is still incurred to perform adaptation. This takes away the ability of TSFMs to forecast zero-shot.
2. The adapter proposed in ChronosX is also based on linear layers. The contribution of this work therefore appears mostly incremental. The authors add additional parameters and attention modules before and after the backbone transformer FM. It is unclear whether the performance gain arises simply from these additional parameters.

**Questions:**

1. How does UniCA perform when provided with noisy covariates?
2. Can UniCA achieve 100% accuracy when provided with the oracle label (i.e. the target time series)?
3. Why does UniCA perform better with Chronos-bolt in some cases and with TimesFM in others?
4. Does your ChronosX implementation achieve the performance reported in the ChronosX Arango et al. paper?

---

> ### Author Response · Authors · 2025-11-23
>
> **Q:** Although TSFM parameters are frozen when training the adapter, training cost is still incurred to perform adaptation. This takes away the ability of TSFMs to forecast zero-shot.
>
> **A**: We agree that zero-shot forecasting is a desirable property of TSFMs. However, zero-shot TSFMs are designed to operate without covariates, and most real-world forecasting tasks rely heavily on domain-specific covariates whose effects cannot be inferred from pretraining alone. As shown in our experiments, these covariates have substantial influence on the target variable, and zero-shot TSFM predictions degrade significantly when such information is omitted. Therefore, an adaptation step is necessary whenever covariates are present.
>
> Our design intentionally freezes the TSFM backbone and trains only small adapter modules, preserving the original zero-shot temporal modeling ability while adding the minimal task-specific capacity required to incorporate covariates. UniCA remains a plug-and-play, parameter-efficient module compatible with a wide range of existing open-source TSFMs. The zero-shot forecasting ability can be recovered by removing the extra module.
>
> ---
>
> **Q**: The adapter proposed in ChronosX is also based on linear layers. The contribution of this work therefore appears mostly incremental. The authors add additional parameters and attention modules before and after the backbone transformer FM. It is unclear whether the performance gain arises simply from these additional parameters.
>
> **A**: We appreciate the comparison to ChronosX. While both ChronosX and UniCA introduce lightweight adapters on top of pretrained TSFMs, our contribution is not incremental for several reasons.
>
> **(1) Scope of supported covariates and TSFM architectures.**   ChronosX is designed for _tabular exogenous covariates_ and operates primarily within the Chronos family (with extensions to TimesFMX/MomentX). In contrast, UniCA introduces a **general covariate homogenization mechanism** that can transform _categorical, continuous, and multimodal (image/text)_ covariates into unified temporal series. This enables UniCA to work seamlessly with **any** patch-based or token-based TSFM (Chronos-Bolt, TimesFM, Time-MoE), and crucially, to handle multimodal covariates—capabilities not supported by ChronosX. As a result, ChronosX appears only in our unimodal comparison (Figure 3a), whereas UniCA achieves substantial gains in multimodal settings (Figures 3b, 3c).
>
> **(2) Performance gains are not due to parameter count.**   The reviewer raises an important question. ChronosX itself explicitly controls for this in Section 6.3:
>
> > _“We analyze whether the performance improvement of ChronosX is obtained from covariate information or from extra parameters… The ablations without covariates (ChronosX (NC)) perform worse across synthetic and real datasets… verifying that the improvement is due to effectively consuming covariate information.”_   Thus, even in ChronosX, additional parameters alone do **not** produce the observed gains.
>
> Our own ablations further reinforce this point:
>
> *   In **Figure 4(c)**, replacing UniCA’s linear homogenizer with a deeper MLP **does not improve** performance, demonstrating that simply adding more parameters is not beneficial.
>
> *   **Figure 5(a)** shows UniCA adds only a negligible parameter budget while delivering consistent improvements across TSFMs.
>
> We provide an additional experiment in **Appendix G.5 that omitting the covariates by setting the gating values of each covariates to zero. The reults are as follows:**
>
> | Metric | Chronos-Bolt (UniCA) | Chronos-Bolt (w/o Cov) | TimesFM (UniCA) | TimesFM (w/o Cov) |
> | --- | --- | --- | --- | --- |
> | **MAE** | **0.509** | 0.517 | **0.526** | 0.526 |
> | **MAPE** | **0.506** | 0.542 | **0.514** | 0.543 |
> | **MSE** | **0.383** | 0.396 | **0.403** | 0.403 |
> | **CRPS** | **0.429** | 0.455 | **0.445** | 0.466 |
>
> the gains are primarily attributable to **the effective and meaningful integration of external covariate information.**
>
> **(3) Architectural novelty beyond ChronosX.**   ChronosX uses separate feed-forward blocks to inject past/future covariates into token embeddings and logits. UniCA, by contrast, introduces:
>
> *   **covariate homogenization** to map heterogeneous covariates into shared temporal series,
>
> *   a **dual attention-based fusion mechanism** (pre- and post-fusion),
>
> *   a **unified plug-and-play design** that leaves the TSFM entirely frozen.
>
> These architectural components address challenges ChronosX does not consider—especially multimodal integration and temporal alignment across heterogeneous covariates.
>
> **Summary.**   While both approaches use lightweight adapters, UniCA provides a **conceptually broader and practically more universal** solution. Our comparisons with ChronosX in Figure 3(a) and extensive ablations demonstrate that UniCA’s gains arise from its homogenization + fusion design, not parameter count.

---

> ### Author Response · Authors · 2025-11-23
>
> **Q**.  How does UniCA perform when provided with noisy covariates?
>
> **A**: Thank you for this critical question regarding the robustness of UniCA to noisy inputs. We agree that stability under imperfect covariate conditions is essential.
>
> We have conducted a dedicated experiment in **Appendix G.6** to address this concern. Our results demonstrate that UniCA is highly robust to the presence of uninformative covariates.
>
> We introduced a purely random, future-unknowable white noise covariate to all time series across our benchmark datasets, simulating an entirely uninformative external feature.
>
> |  | Chronos-Bolt |  |  | TimesFM |  |  |
> | --- | --- | --- | --- | --- | --- | --- |
> |  | ZS | UniCA | w/ Noise | ZS | UniCA | w/ Noise |
> | Avg | 0.472 | **0.457** | 0.468 | 0.473 | **0.472** | 0.475 |
> | MAE | 0.521 | **0.509** | 0.518 | 0.530 | **0.526** | 0.526 |
> | MAPE | 0.522 | **0.506** | 0.516 | 0.523 | **0.514** | 0.523 |
> | MSE | 0.403 | **0.383** | 0.397 | **0.402** | 0.403 | 0.403 |
> | CRPS | 0.441 | **0.429** | 0.441 | **0.437** | 0.445 | 0.446 |
>
> As shown in **Table**, the inclusion of the noise feature resulted in only a **marginal degradation** in performance for UniCA (e.g., Chronos-Bolt Avg: $0.457$ $\rightarrow$ $0.468$; TimesFM Avg: $0.472$ $\rightarrow$ $0.475$). Even with the added noise, the UniCA (w/ Noise) configuration consistently **maintained a superior performance level** compared to the Zero-Shot (ZS) baseline.
>
> This confirms that UniCA's adaptation mechanism is highly effective at identifying and suppressing irrelevant features, validating the model's stability in real-world scenarios where covariate quality may vary.

---

> ### Author Response · Authors · 2025-11-23
>
> **Q**: Can UniCA achieve 100% accuracy when provided with the oracle label (i.e. the target time series)?
>
> **A**:  Thank you for raising this insightful question: **UniCA fuses covariates through a lightweight adapter, but not overwriting the TSFM’s prediction head.** The prediction is still heavily smoothed and regularized by the TSFM's structure, resulting in a more generalized, smoother forecast typical of large time series models, rather than a direct copy of the input or covariate. Methods that can achieve near-100% "copy" accuracy often do so by using a linear regression (LR) or similar simple adapter that effectively replaces the TSFM prediction head with a direct mapping from the covariate (oracle label) to the target. While this is useful for demonstrating the theoretical maximum under an _oracle_ condition, our UniCA method prioritizes **robustness and performance in real-world scenarios** where future covariates are **not perfectly knowable (In Figure 3(a) compared to LR adaptor)**
>     In conclusion, the inability to perfectly copy the oracle is a known limitation that stems from preserving the TSFM's strong temporal modeling capability. However, our results demonstrate that in the more common and practical scenarios, our method achieves robust performance.
> ---
> **Q**: Why does UniCA perform better with Chronos-bolt in some cases and with TimesFM in others?
>
> **A**: We observe complementary strengths between Chronos-Bolt (CB) + UniCA and TimesFM (TFM) + UniCA. This pattern is consistent across datasets and follows naturally from the interaction between **dataset characteristics** and **TSFM architectures**.
>
> **(1) Empirical pattern across datasets**
>
> *   **Unimodal tasks:** Chronos-Bolt + UniCA achieves the best average performance (Fig. 3a). From Table 9, Chronos-Bolt shows clear advantages on datasets such as _Bull_ and _Spain_, while both models perform similarly on the remaining datasets.
>
> *   **TS-Text (Time-MMD):** TimesFM + UniCA outperforms Chronos-Bolt on six of seven sub-datasets (Table 12), and achieves better aggregate performance (0.638 vs 0.671).
>
> *   **TS-Image (MMSP):** Chronos-Bolt + UniCA performs best, both with and without the NWP covariates (Table 1).
>
>
> **(2) Dataset-wise explanation**
>
> *   **Unimodal datasets** rely primarily on modeling long-range temporal dependencies. Chronos-Bolt, with its encoder–decoder architecture, compresses the entire history through a strong encoder before decoding, giving it an advantage on pure time-series tasks.
>
> *   **Text-enhanced Time-MMD datasets** require integrating linguistic information with time-series dynamics. TimesFM’s decoder-only architecture is more flexible for fusing covariates during autoregressive decoding, which aligns naturally with the structure of TS-Text forecasting.
>
> *   **Image-enhanced MMSP datasets** involve spatial-temporal dynamics and abrupt regime shifts due to weather conditions. Chronos-Bolt’s bidirectional encoder provides richer context aggregation, which appears more effective for these datasets.
>
>
> **(3) Architectural explanation**
>
> *   **Chronos-Bolt (encoder–decoder)** provides a strong bidirectional encoder that excels at extracting structured temporal patterns when the forecasting signal is dominated by the time series itself. This explains its superior performance on unimodal datasets and the TS-Image task.
>
> *   **TimesFM (decoder-only)** integrates covariates directly within its autoregressive decoding process, making it more adaptive when heterogeneous covariates (especially text) are informative at each forecasting step. This explains its consistent advantage on TS-Text datasets.
>
> Summary:  The relative performance of UniCA across TSFMs is both **dataset-dependent** and **architecture-driven**. Chronos-Bolt benefits settings where temporal structure dominates, whereas TimesFM benefits tasks requiring stronger multimodal fusion—particularly text. UniCA exposes these strengths by providing a unified interface to covariates while preserving each TSFM’s intrinsic inductive biases.
>
> ----
> **Q**: Does your ChronosX implementation achieve the performance reported in the ChronosX Arango et al. paper?
>
> **A**: We followed the ChronosX paper as closely as possible. Specifically, our implementation reproduces the architecture design, covariate injection blocks, training procedure, and hyperparameter settings described in the paper. While exact numerical reproduction is difficult due to differences in codebases, preprocessing pipelines, and training infrastructure, our implementation behaves consistently and achieves performance levels that are in the same range as those reported in the paper, giving us confidence that our ChronosX baseline is faithful for comparison.

---

### Author Response · Authors · 2025-11-21
**Potential Delay/Incompleteness due to Illness**

Dear Reviewers, ACs, SACs, and PCs:

We sincerely apologize for any potential delay or incompleteness in our author response.

A primary author of this submission has unfortunately **contracted a severe case of the flu**. Consequently, we anticipate that the comprehensive response addressing each reviewer's comment **may not be submitted in full or on time by November 20th**.

**We are working diligently and are committed to submitting the complete reply and revision as quickly as possible**.

Thank you very much for your understanding and patience during this difficult time.

Sincerely,

The Authors

---

### Author Response · Authors · 2025-11-23

Dear Reviewers, ACs, SACs, and PCs:

Thanks very much for your patience. We have made the changes to our maniscript thanks to the valuable suggestions. Here is summary of changes:

*   Updated **Appendix E** to include the new **MOMENT imputation experiments**, demonstrating UniCA’s applicability beyond forecasting, along with full results and protocol details.

*   Updated **Appendix G.5-G.7** to include experiments that demonstrate 1. whether the performance gain comes from covaraites or additional parameters, 2. robustnesss on noisy covariates, 3. impact of using a simpler fusion structure.

*   Updated **Appnedix H** to the pearson correlation between homogenized series and the target,  demonstrating the ablility to learn meaningful pattern with covariate homogenizer.

*   Revised performance wording to improve clarity—for example, replacing phrases such as “-6.5% improvement” with **“6.5% reduction in error”** in Section 5.2 to avoid unintended interpretations of degradation.

*   We enlarge the font of the label in Fig 4 (c) to imporve visibility.

*   We updated the related work section with discussions about MiTSformer.

---

### Author Response · Authors · 2025-12-03

We are very grateful to the four reviewers and the AC for their time and thoughtful discussion.

Our paper introduces **UniCA**, a **unified covariate adaptation framework** that bridges frozen Time Series Foundation Models (TSFMs) with general covariate-aware forecasting. By combining a universal covariate homogenizer with a unified attention-based fusion mechanism, UniCA enables TSFMs to effectively leverage heterogeneous covariates—including categorical variables, text, and images—without compromising their pretrained generalization capabilities.

**Shared positive assessment:**
* **Motivation & Significance:** Reviewers agree that adapting channel-independent TSFMs to handle heterogeneous covariates is a critical and practically significant problem (Reviewers 87eP, Ruus, KAgW).
* **Methodological Soundness:** The two-stage design (homogenization + fusion) is considered intuitive, technically sound, and broadly compatible with various TSFM architectures (Reviewers h1TS, 87eP).
* **Extensive Evaluation:** The experiments are regarded as comprehensive, covering multiple TSFM backbones (Chronos, TimesFM, MOMENT) and diverse modalities (unimodal, text, image), with clear visualizations and ablations (Reviewers Ruus, h1TS, 87eP, KAgW).
* **Efficiency:** The plug-and-play nature and minimal inference overhead of the proposed adapter are highlighted as key strengths (Reviewers Ruus, h1TS).

**Per-reviewer summaries:**

* **Reviewer 87eP (rating 6 → 8):**

* **Main concerns:** Robustness of tokenization methods, extension to non-forecasting tasks, missing data handling, and comparison with MiTSformer.
* **Response:** We provided ablations on different text tokenizers (App. G.2), extended UniCA to **imputation tasks using MOMENT** (App. E), discussed missing data strategies, and differentiated our work from MiTSformer in the related work.
* **Outcome:** The reviewer stated **"My concerns are addressed"** and explicitly **raised the score from 6 to 8**.

* **Reviewer h1TS (6):**

* **Main concerns:** Effectiveness of homogenization (alignment), justification for GRN/GLU fusion design, and zero-shot compatibility.
* **Response:** We added a **Pearson correlation analysis** (App. H) to prove homogenized features capture temporal structure, conducted an ablation comparing GRN/GLU against simple weight fusion (App. G.7), and clarified that the frozen backbone preserves zero-shot capabilities. We further clarified why low-correlation covariates in noisy domains (e.g., Time-MMD) still provide useful contextual signals during the discussion.
* **Outcome:** The reviewer acknowledged our "great efforts" and engaged in constructive discussion regarding feature correlation.

* **Reviewer Ruus (4):**

* **Main concerns:** Whether gains come from parameters or covariates, robustness to noisy covariates, and comparison with ChronosX.
* **Response:** We added experiments showing UniCA outperforms baselines even with **random noise covariates** (App. G.6) and that removing covariate information (zero-gating) eliminates gains (App. G.5), proving the value comes from data, not parameters. We also highlighted UniCA’s unique multimodal capabilities compared to the tabular-only ChronosX.
* **Outcome:** We have addressed all technical questions, though the reviewer has not yet responded to the rebuttal.

* **Reviewer KAgW (6):**

* **Main concerns:** Initialization of categorical embeddings and preservation of temporal information during homogenization.
* **Response:** We explained the end-to-end learning dynamic of categorical embeddings and provided the **Pearson correlation analysis** (App. H) to demonstrate that the homogenization layer retains distinct and meaningful temporal patterns rather than simply "mixing" them.
* **Outcome:** We have addressed the concerns, though the reviewer has not yet responded to the rebuttal.

**Summary of changes and new experiments:**

We have incorporated extensive new results and revisions based on reviewer feedback:

1.  **Extended Task Scope (App. E):** Added experiments on the **MOMENT** TSFM for **imputation**, demonstrating UniCA’s universality beyond forecasting (Reviewer 87eP).
2.  **Structural Ablation (App. G.7):** Compared UniCA’s GRN/GLU fusion against a simple linear "Weight Fusion" baseline to justify architectural choices (Reviewer h1TS).
3.  **Robustness Analysis (App. G.5, G.6):** Added ablations with **noisy covariates** and **zero-gating**, confirming performance gains stem from effective covariate integration rather than parameter increase (Reviewer Ruus).
4.  **Interpretability Analysis (App. H):** Added **Pearson Correlation Coefficients** between homogenized embeddings and target series to validate the effectiveness of the homogenization module (Reviewers h1TS, KAgW).
5.  **Revisions:** Updated related work (MiTSformer), improved figure readability (font sizes), and refined performance metric descriptions.
6.  The **repository** has now been restored and is accessible again.

---

### Meta-Review · Area_Chair_VRb9 · 2026-01-06

**Summary:**

This paper proposes UniCA, a unified covariate adaptation framework that enables pretrained Time Series Foundation Models (TSFMs) to effectively incorporate heterogeneous covariates, including categorical variables as well as multimodal inputs such as text and images. The problem is timely and practically important, as most existing TSFMs are designed for channel-independent real-valued series and struggle to leverage rich exogenous information commonly available in real-world forecasting scenarios.

Reviewers consistently recognized several strengths of the work: (i) clear motivation and problem formulation, (ii) a clean and modular two-stage design (covariate homogenization followed by attention-based fusion) that is broadly compatible with different TSFM backbones, and (iii) extensive experimental validation across unimodal and multimodal benchmarks, multiple TSFMs (e.g., Chronos, TimesFM, MOMENT), and additional tasks such as imputation. The proposed framework is parameter-efficient, preserves the frozen backbone, and demonstrates consistent performance gains with minimal inference overhead.

During the rebuttal phase, the authors responded constructively to reviewer feedback. They added new experiments, ablations, robustness analyses (e.g., noisy covariates), task extensions, and interpretability studies (e.g., correlation analysis), and clarified architectural choices and zero-shot compatibility. These efforts substantially strengthened the paper, and at least one reviewer explicitly raised their score after the rebuttal, while others acknowledged that their major concerns had been addressed.

Overall, UniCA represents a solid and practically useful contribution to covariate-aware adaptation of TSFMs. While the methodological novelty is incremental in parts and the empirical gains are sometimes moderate, the framework is well motivated, carefully evaluated, and likely to be of interest to the time series community. I therefore recommend acceptance as a poster, which appropriately reflects the paper’s strengths and scope.

**Reviewer Concerns:**

Reviewers initially raised concerns regarding embedding alignment across heterogeneous modalities, justification of fusion design choices, robustness to noisy covariates, preservation of zero-shot capabilities, and task generality. These concerns were largely addressed in the rebuttal through additional experiments, ablations, clarifications, and extended evaluations. Remaining issues are relatively minor and mostly pertain to presentation clarity or further discussion, which can be handled in the camera-ready version.

**Reviewer Scores:**

Reviewer scores were initially mixed but generally around or above the acceptance threshold. During the rebuttal phase, one reviewer explicitly increased their score after confirming that their concerns were addressed, while other reviewers did not further revise their scores but acknowledged the authors’ efforts and clarifications. Based on the final score distribution, the rebuttal, and the Area Chair’s assessment, the paper meets the bar for acceptance.

---

### Decision · Program_Chairs · 2026-01-26

Accept (Poster)